# How does Inverse RL Scale to Large State Spaces?
# A Provably Efficient Approach

**Filippo Lazzati**
Politecnico di Milano
Milan, Italy
filippo.lazzati@polimi.it

**Mirco Mutti**
Technion
Haifa, Israel

**Alberto Maria Metelli**
Politecnico di Milano
Milan, Italy

## Abstract

In online Inverse Reinforcement Learning (IRL), the learner can collect samples about the dynamics of the environment to improve its estimate of the reward function. Since IRL suffers from identifiability issues, many theoretical works on online IRL focus on estimating the entire set of rewards that explain the demonstrations, named the *feasible reward set*. However, none of the algorithms available in the literature can scale to problems with large state spaces. In this paper, we focus on the online IRL problem in Linear Markov Decision Processes (MDPs). We show that the structure offered by Linear MDPs is not sufficient for efficiently estimating the feasible set when the state space is large. As a consequence, we introduce the novel framework of *rewards compatibility*, which generalizes the notion of feasible set, and we develop `CATY-IRL`, a sample efficient algorithm whose complexity is independent of the cardinality of the state space in Linear MDPs. When restricted to the tabular setting, we demonstrate that `CATY-IRL` is minimax optimal up to logarithmic factors. As a by-product, we show that Reward-Free Exploration (RFE) enjoys the same worst-case rate, improving over the state-of-the-art lower bound. Finally, we devise a unifying framework for IRL and RFE that may be of independent interest.

## 1 Introduction

Inverse Reinforcement Learning (IRL) is the problem of inferring the reward function given demonstrations of an optimal behavior, i.e., from an *expert* agent. [49, 42]. Since its formulation, much of the research effort has been put into the design of efficient algorithms for solving the IRL problem [6, 4]. Indeed, the solution of the IRL problem opens the door to a variety of interesting applications, including Apprenticeship Learning (AL) [2, 1], reward design [16], interpretability of the expert's behavior [17], and transferability to new environments [15].

Nowadays, the factor that most negatively impacts the adoption of IRL solutions in real-world applications is the intrinsic *ill-posedness* of its formulation. The IRL problem has been historically defined as the problem of recovering *the* reward function underlying the demonstrations [49, 42], even though mere demonstrations can be equivalently explained by a *variety* of rewards. In other words, the IRL problem is underconstrained, even in the limit of infinite demonstrations [42, 39].

To overcome this weakness and to come up with a *single* reward function, three main approaches are commonly adopted in the literature. ($i$) The first approach consists of the use of a *heuristic* to select a specific reward function from the set of all the rewards that explain the demonstrations. Implicitly, these works re-define IRL as the problem of recovering *the* reward function explaining the demonstrations *and* complying with the heuristic. As an example, [42, 48] select the reward that maximizes some notion of margin, and [70] implicitly chooses the reward returned by the optimization algorithm among those that maximize the likelihood. However, these approaches may generate issues in applications [56, 15]. ($ii$) In the second approach, additional *constraints* beyond

mere demonstrations are enforced to guarantee the uniqueness of the reward function to recover. In "reward identifiability" works, the additional information commonly concerns some structure of the environment [25], or multiple demonstrations across various environments [5, 10]. In Reward Learning (ReL) works [19], demonstrations of optimal behavior are combined with other kinds of expert feedback, like comparisons [65]. $(iii)$ As a third approach, recently, [39, 38] proposed the alternative formulation of IRL as the problem of recovering *all* the reward functions compatible with the demonstrations, i.e., the *feasible reward set*. In this manner, we are not subject to the limitations of the first approach, and we do not depend on additional information like in the second approach.

In practical applications, the chosen IRL formulation has to be tackled by algorithms that use a *finite* number of demonstrations and a limited knowledge of the dynamics of the environment. In the common *online* IRL scenario, the learner explores the (unknown) environment, and exploits this additional information to improve its performance on the IRL task [e.g., 39, 33, 38, 68, 31]. On this basis, the IRL approach $(iii)$ based on the *feasible set* [39, 38] displays desirable properties since "postpones" the choice of the heuristic and/or enforcement of additional constraints, with the advantage of analyzing the intrinsic complexity of the IRL problem only, without being obfuscated by other factors. In other words, this recent formulation of the IRL problem paves the way for the design and analysis of provably efficient IRL algorithms, endowed with solid theoretical guarantees.

However, the algorithms designed for learning the feasible set currently available in the literature [e.g., 39, 33, 38, 68, 31] struggle when attempting to scale them to IRL problems with *large state spaces*. This is apparent because their sample complexity exhibits an explicit dependence on the cardinality of the state space. This inevitably represents a major limitation since most real-world scenarios concern problems with large, or even continuous, state spaces [e.g., 15, 7, 40, 14].

In this context, function approximation represents an essential tool to tackle the curse of dimensionality and enforce generalization [54, 41]. Linear Markov Decision Processes (MDPs) [23, 67] offer a simple but powerful structure, in which we assume the reward function and the transition model can be expressed as linear combinations of known features, that permits theoretical analysis of the sample complexity. Even though many extensions have been developed [64, 22, 13], the Linear MDPs framework typically represents one of the first function approximation settings to analyze when focusing on a novel problem, before moving to more complex settings [e.g., 63, 61].

In this paper, we aim to shed light on the challenges of scaling the feasible reward set to large-scale problems. Motivated by its limitations when dealing with large state spaces, we introduce the novel *Rewards Compatibility* framework. Being a generalization of the notion of feasible set, it allows us to define the new *IRL Classification Problem*, a fourth approach to cope with the ill-posedness of the IRL formulation. This permits the development of `CATY-IRL` (Comp**AT**ibilit**Y** for `IRL`), a provably efficient IRL algorithm for Linear MDPs characterized by large or even continuous state spaces.

**Original Contributions.** The main contributions of the current work can be summarized as follows:

- We prove that the notion of feasible set can *not* be learned efficiently in MDPs with large/continuous state spaces, even under the structure enforced by Linear MDPs. Nevertheless, we show that this problem disappears under the *assumption* that the expert's policy is known, by providing a sample efficient algorithm for such setting (Section 3).
- To overcome the need for knowing the expert's policy exactly, we propose *Rewards Compatibility*, a novel framework that formalizes the intuitive notion of *compatibility* of a reward function with expert demonstrations. It generalizes the feasible set and allows us to define an original learning setting, *IRL classification*, based on a new formulation of IRL *classification* task (Section 4).
- For the newly-devised framework, we develop `CATY-IRL` (Comp**AT**ibilit**Y** for `IRL`), a new sample and computationally efficient IRL algorithm for both tabular and Linear MDPs. Remarkably, this `CATY-IRL` does not require the additional assumption that the expert's policy is known (Section 5).
- In the tabular setting, we prove a tight minimax lower bound to the sample complexity of the IRL classification problem of $\Omega\big(\frac{H^3 SA}{\epsilon^2}(S + \log \frac{1}{\delta})\big)$ episodes, where $S$ and $A$ are the cardinalities of the state and action spaces, $H$ is the horizon, $\epsilon$ the accuracy and $\delta$ the failure probability. This bound is *matched* by `CATY-IRL`, up to logarithmic factors. Exploiting a similar construction, we show that a lower bound with the same rate holds also for the Reward-Free Exploration (RFE) problem, improving by an $H$ factor over the RFE state-of-the-art lower bound [21] (Section 6.1).
- Finally, we formulate a novel *Objective-Free Exploration* (OFE) setting that isolates the challenges of exploration beyond Reinforcement Learning (RL), by unifying RFE and IRL (Section 6.2).

Additional related works and the proofs of all the results are reported in Appendix A and B -E.

## 2 Preliminaries

**Notation.** Given an integer $N \in \mathbb{N}$, we define $[\![N]\!] := \{1, \ldots, N\}$. Given sets $\mathcal{X}$ and $\mathcal{Y}$, we denote $\mathcal{H}_d(\mathcal{X}, \mathcal{Y}) := \max\{\sup_{x \in \mathcal{X}} \inf_{y \in \mathcal{Y}} d(x, y), \sup_{y \in \mathcal{Y}} \inf_{x \in \mathcal{X}} d(y, x)\}$ their Hausdorff distance with inner distance $d$. We denote by $\Delta^{\mathcal{X}}$ the probability simplex over $\mathcal{X}$, and by $\Delta^{\mathcal{X}}_{\mathcal{Y}}$ the set of functions from $\mathcal{Y}$ to $\Delta^{\mathcal{X}}$. Sometimes, we denote the dot product between vectors $x, y$ as $\langle x, y \rangle := x^\mathsf{T} y$. We employ $\mathcal{O}, \Omega, \Theta$ for the common asymptotic notation and $\widetilde{\mathcal{O}}, \widetilde{\Omega}, \widetilde{\Theta}$ to omit logarithmic terms.

**Markov Decision Processes.** A finite-horizon Markov Decision Process (MDP) without reward [45] is defined as a tuple $\mathcal{M} := (\mathcal{S}, \mathcal{A}, H, d_0, p)$, where $\mathcal{S}$ and $\mathcal{A}$ are the measurable state and action spaces, $H \in \mathbb{N}$ is the horizon, $d_0 \in \Delta^{\mathcal{S}}$ is the initial-state distribution, and $p \in \mathcal{P} := \Delta^{\mathcal{S}}_{\mathcal{S} \times \mathcal{A} \times [\![H]\!]}$ is the transition model. Given a (deterministic) reward function $r \in \mathfrak{R} := [-1, 1]^{\mathcal{S} \times \mathcal{A} \times [\![H]\!]}$, we denote by $\overline{\mathcal{M}} := \mathcal{M} \cup \{r\}$ the MDP obtained by pairing $\mathcal{M}$ and $r$. Each policy $\pi \in \Pi := \Delta^{\mathcal{A}}_{\mathcal{S} \times [\![H]\!]}$ induces in $\overline{\mathcal{M}}$ a state-action probability distribution $d^{p,\pi} := \{d^{p,\pi}_h\}_{h \in [\![H]\!]}$ (we omit $d_0$ for simplicity) that assigns, to each subset $\mathcal{Z} \subseteq \mathcal{S} \times \mathcal{A}$, the probability of being in $\mathcal{Z}$ at stage $h \in [\![H]\!]$ when playing $\pi$ in $\overline{\mathcal{M}}$. We denote with $\mathcal{S}^{p,\pi}_h$ the set of states supported by $d^{p,\pi}_h$ for any action at stage $h$, and with $\mathcal{S}^{p,\pi}$ the disjoint union of sets $\{\mathcal{S}^{p,\pi}_h\}_{h \in [\![H]\!]}$. The $Q$-function of policy $\pi$ in MDP $\overline{\mathcal{M}}$ is defined at every $(s, a, h) \in \mathcal{S} \times \mathcal{A} \times [\![H]\!]$ as $Q^\pi_h(s, a; p, r) := \mathbb{E}_{p,\pi}[\sum_{t=h}^{H} r_t(s_t, a_t) | s_h = s, a_h = a]$, and the optimal $Q$-function as $Q^*_h(s, a; p, r) := \sup_{\pi \in \Pi} Q^\pi_h(s, a; p, r)$, where the expectation $\mathbb{E}_{p,\pi}$ is computed over the stochastic process generated by playing policy $\pi$ in the MDP $\overline{\mathcal{M}}$. Similarly, we define the $V$-function of policy $\pi$ at $(s, h)$ as $V^\pi_h(s; p, r) := \mathbb{E}_{p,\pi}[\sum_{t=h}^{H} r_t(s_t, a_t) | s_h = s]$, and the optimal $V$-function as $V^*_h(s; p, r) := \sup_{\pi \in \Pi} V^\pi_h(s; p, r)$. We define the utility of $\pi$ as $J^\pi(r; p) := \mathbb{E}_{s \sim d_0}[V^\pi_1(s; p, r)]$, and the optimal utility as $J^*(r; p) := \mathbb{E}_{s \sim d_0}[V^*_1(s; p, r)]$. A forward (sampling) model of the environment permits to collect samples starting from $s \sim d_0$ and following some policy. A generative (sampling) model consists in an oracle that, given an arbitrary state-action-stage triple $s, a, h$ in input, returns a sampled next state $s' \sim p_h(\cdot | s, a)$.

**Linear MDPs.** Based on [23], we say that an MDP $\overline{\mathcal{M}} = (\mathcal{S}, \mathcal{A}, H, d_0, p, r)$ is a *Linear MDP* with a (known) feature map $\phi : \mathcal{S} \times \mathcal{A} \to \mathbb{R}^d$, if for every $h \in [\![H]\!]$, there exist $d \in \mathbb{N}$ unknown (signed) measures $\mu_h = [\mu^1_h, \ldots, \mu^d_h]^\mathsf{T}$ over $\mathcal{S}$ and an unknown vector $\theta_h \in \mathbb{R}^d$, such that for every $(s, a) \in \mathcal{S} \times \mathcal{A}$, we have $p_h(\cdot | s, a) = \langle \phi(s, a), \mu_h(\cdot) \rangle$ and $r_h(s, a) = \langle \phi(s, a), \theta_h \rangle$. Without loss of generality, we assume $\|\phi(s, a)\|_2 \leq 1$ for all $(s, a) \in \mathcal{S} \times \mathcal{A}$, and $\max\{\|\theta_h\|_2, \||\mu_h|(\mathcal{S})\|_2\} \leq \sqrt{d}$.[1] $\mathcal{M}$ is a *Linear MDP without reward* if its transition model satisfies the assumption described above.

**BPI and RFE.** In both Best-Policy Identification (BPI) [37] and Reward-Free Exploration (RFE) [21], the learner has to explore the *unknown* MDP to optimize a certain reward function. In BPI, the learner observes the reward function $r$ during exploration, and its goal is to output a policy $\widehat{\pi}$ such that, in the true MDP with transition model $p$ we have $\mathbb{P}(J^*(r; p) - J^{\widehat{\pi}}(r; p) \leq \epsilon) \geq 1 - \delta$ for every $\epsilon, \delta \in (0, 1)$. RFE considers the setting in which the reward to optimize is revealed *a posteriori* of the exploration phase. Thus the goal of the agent in RFE is to compute an estimate $\widehat{p}$ of the true dynamics $p$ so that $\mathbb{P}(\sup_{r \in \mathfrak{R}} \{J^*(r; p) - J^{\widehat{\pi}_r}(r; p)\} \leq \epsilon) \geq 1 - \delta$ for every $\epsilon, \delta \in (0, 1)$, where $\widehat{\pi}_r$ is the optimal policy in the MDP with $\widehat{p}$ as transition model and $r$ as reward function.

**Online IRL.** We consider the online[2] IRL setting [39, 33, 68, 66, 53] in which, similarly to the online AL setting [53, 66], we are given a dataset $\mathcal{D}^E = \{(s^i_1, a^i_1, \ldots, s^i_{H-1}, a^i_{H-1}, s^i_H)\}_{i \in [\![\tau^E]\!]}$ of $\tau^E \in \mathbb{N}$ trajectories collected by executing the expert's policy $\pi^E$ in a certain (unknown) MDP $\overline{\mathcal{M}} = \mathcal{M} \cup \{r^E\}$. We make the assumption that $\pi^E$ is optimal under the true (unknown) reward $r^E$ in $\overline{\mathcal{M}}$. Since the dynamics of $\overline{\mathcal{M}}$ is unknown, we are allowed to actively explore the environment through a *forward* model to collect a new state-action dataset $\mathcal{D}$. The goal is to use the latter and demonstrations in $\mathcal{D}^E$ to estimate a reward function that makes the expert's policy $\pi^E$ optimal. Sometimes, we will denote an IRL instance as $\mathcal{M} \cup \{\pi^E\}$, and a Linear IRL instance with recovered reward $r$ as an IRL instance in which $\mathcal{M} \cup \{r\}$ is a Linear MDP.

---

[1] $|\mu_h|(\mathcal{B})$ denotes the vector containing the variation of each measure $\mu^i_h$ over the measurable set $\mathcal{B}$.

[2] "Online" refers to how we estimate the transition model $p$, not to the expert's policy $\pi^E$, for which we assume to have access to a batch dataset. This is justified by the fact that most of IRL real-world applications involve the presence of a fixed dataset of expert demonstrations previously collected and the agent can explore the environment in order to reconstruct one (or more) reward functions compatible with those demonstrations.

# 3 Limitations of the Feasible Set

In this section, after having characterized the feasible set formulation in Linear MDPs, we show that it suffers from *statistical* (and *computational*) inefficiency in problems with large state spaces, even under the Linear MDP assumption. We will provide a solution to these issues in Section 4.

**The Feasible Set.** According to the standard definition [e.g., 39, 33, 38, 68, 31], the feasible set contains the rewards that make the expert's policy $\pi^E$ optimal, as defined below.

**Definition 3.1** (Feasible Set [31])**.** *Let $\mathcal{M}$ be an MDP without reward and let $\pi^E$ be the expert's policy. The* feasible set $\mathcal{R}_{p,\pi^E}$ *of rewards compatible with $\pi^E$ in $\mathcal{M}$ is defined as:*

$$\mathcal{R}_{p,\pi^E} := \{r \in \mathfrak{R} \mid J^{\pi^E}(r;p) = J^*(r;p)\}.$$

Without function approximation, the feasible set contains a variety of rewards for any deterministic policy. In Linear MDPs, due to the feature map, the feasible set might exhibit some degeneracy.[3] Definition 3.1 can be adapted to Linear MDPs with feature map $\phi$ as: $\mathcal{R}_{\phi,p,\pi^E} := \{r \in \mathfrak{R} \mid J^{\pi^E}(r;p) = J^*(r;p) \wedge \exists\theta : [\![H]\!] \to \mathbb{R}^d, \forall(s,a,h) \in \mathcal{S} \times \mathcal{A} \times [\![H]\!] : r_h(s,a) = \langle\phi(s,a),\theta_h\rangle\}$. We omit $\phi$ in $\mathcal{R}_{\phi,p,\pi^E}$ for notational simplicity.

**Proposition 3.1.** *Let $\mathcal{M}$ be a Linear MDP without reward with a finite state space, and let $\phi$ be a feature mapping. Let $\{\Phi_h^{\pi^E}\}_{h\in[\![H]\!]}$ and $\{\overline{\Phi}_h\}_{h\in[\![H]\!]}$ be the sets of expert's and non-expert's features, defined for every $h \in [\![H]\!]$ as:*

$$\Phi_h^{\pi^E} := \big\{\phi(s,a^E) \mid s \in \mathcal{S}_h^{p,\pi^E},\, a^E \in \mathcal{A}_h^E(s)\big\}, \qquad \overline{\Phi}_h := \big\{\phi(s,a) \mid s \in \mathcal{S}_h^{p,\pi^E},\, a \in \mathcal{A}\backslash\mathcal{A}_h^E(s)\big\},$$

*where $\mathcal{A}_h^E(s) := \{a \in \mathcal{A} \mid \pi_h^E(\cdot|s) > 0\}$ for every $s \in \mathcal{S}$. If for none of the $H$ pairs of sets $(\Phi_h^{\pi^E}, \overline{\Phi}_h)$ there exists a separating hyperplane, then $\mathcal{R}_{p,\pi^E} = \{\overline{r}\}$, with $\overline{r}_h(s,a) = 0 \,\forall(s,a,h) \in \mathcal{S} \times \mathcal{A} \times [\![H]\!]$ i.e., the feasible set with linear rewards in $\phi$ contains only the reward function that assigns zero reward everywhere.*

Intuitively, expert's actions must have the largest optimal $Q$-value among all actions, and linearity imposes the "separability" requirement. The result holds also for MDPs with linear rewards only. We exemplify Proposition 3.1 in Appendix B.1.

**Learning the Feasible Set.** In order to highlight the challenges of learning the feasible set with large-scale MDPs, based on [38, 31], we devise the following PAC requirement.

**Definition 3.2** (PAC Algorithm)**.** *Let $\epsilon,\delta \in (0,1)$, and let $\mathfrak{A}$ be an algorithm that collects $\tau^E$ samples about $\pi^E$ using a generative model, and $\tau$ episodes from a Linear MDP without reward $\mathcal{M} = (\mathcal{S}, \mathcal{A}, H, d_0, p)$ using a forward model. Let $\widehat{\mathcal{R}}$ be the estimate of the feasible set $\mathcal{R}_{p,\pi^E}$ outputted by $\mathfrak{A}$. Then, $\mathfrak{A}$ is $(\epsilon,\delta)$-PAC for IRL if $\mathbb{P}_{\mathcal{M},\mathfrak{A}}\big(\mathcal{H}_d(\mathcal{R}_{p,\pi^E}, \widehat{\mathcal{R}}) \leqslant \epsilon\big) \geqslant 1 - \delta$, where $\mathbb{P}_{\mathcal{M},\mathfrak{A}}$ is the probability measure induced by $\mathfrak{A}$ in $\mathcal{M}$, and $d(r,\widehat{r}) \propto \sup_{\pi\in\Pi} \sum_{h\in[\![H]\!]} \mathbb{E}_{(s,a)\sim d_h^{p,\pi}(\cdot,\cdot)} |r_h(s,a) - \widehat{r}_h(s,a)|$.[4] The* sample complexity *is the pair $(\tau^E, \tau)$.*

It is worth noting that in Definition 3.2, we are considering a generative model for collecting samples from the expert's policy, which represents the easiest learning scenario. The following result shows that, even in this convenient setting, estimating the feasible set is statistically inefficient.

**Theorem 3.2** (Statistical Inefficiency)**.** *Let $\mathcal{M} \cup \{\pi^E\}$ be a Linear IRL instance with finite state space $\mathcal{S}$ and deterministic expert's policy, and let $\epsilon,\delta \in (0,1)$. If an algorithm $\mathfrak{A}$ is $(\epsilon,\delta)$-PAC, then $\tau^E = \Omega(S)$, where $S := |\mathcal{S}|$ is the cardinality of the state space.*

In other words, even under the easiest learning conditions (i.e., generative model and deterministic expert), the sample complexity scales directly with the cardinality of the state space $S$, thus, it is infeasible when $S$ is large or even infinite. Observe that this result extends to any class of MDPs that contains Linear MDPs. In Appendix B.2, we analyze if additional assumptions can drop the $\Omega(S)$ dependence. Nevertheless, if $\pi^E$ is *known*, it is possible to construct sample efficient algorithms. Algorithm 1 (whose pseudocode is presented in Appendix B.3), under the assumption that $\pi^E$ is known, makes use of an inner RFE routine (Algorithm 1 of [62]) to recover the feasible set.

---

[3]We exemplify this proposition in Appendix B.1. In Appendix B.4 we generalize to infinite state spaces.

[4]For simplicity, we provide the full expression of distance $d$ in Appendix B.4, Equation (1).

**Theorem 3.3.** *Assume that $\pi^E$ (along with its support $\mathcal{S}^{p,\pi^E}$) is known. Then, for any $\epsilon, \delta \in (0,1)$, Algorithm 1 is $(\epsilon,\delta)$-PAC for IRL with a number of episodes $\tau$ upper bounded by:*

$$\tau \leqslant \widetilde{\mathcal{O}}\Big(\frac{H^5 d}{\epsilon^2}\Big(d + \log\frac{1}{\delta}\Big)\Big).$$

**Limitations of the Feasible Set.** We can now conclude that the feasible set suffers from two main limitations. $(i)$ *Sample Inefficiency*: If $\pi^E$ is unknown, it requires a number of samples that depends on the cardinality of the state space (Theorem 3.2). $(ii)$ *Lack of Practical Implementability*: It contains a continuum of rewards, thus, no practical algorithm can explicitly compute it. We will discuss in the next section how to overcome both these issues.

## 4 Rewards Compatibility

In this section, we present the main contribution of this work: *Rewards Compatibility*, a novel framework for IRL that allows us to conveniently rephrase the learning from demonstrations problem as a classification task. We anticipate that the presentation of the framework is completely general and independent of structural assumptions of the MDP (e.g., Linear MDP).

### 4.1 Compatible Rewards

In the following, for ease of presentation, we consider the exact setting, i.e., when $d_0$, $p$, and $\pi^E$ are known. In addition, we will drop the dependence on $p$ when clear from the context.

In IRL, an expert agent demonstrates policy $\pi^E$ assumed optimal under some (unknown) reward function $r^E$, i.e., $J^*(r^E) = J^{\pi^E}(r^E)$. The task is to recover a reward $r$ such that $J^*(r) = J^{\pi^E}(r)$. By definition, IRL tells us that $r^E$ makes the demonstrated policy $\pi^E$ optimal, but what about other policies? We *do not* and *cannot* know. Since there are (infinite) rewards making $\pi^E$ optimal (but they differ in the performance attributed to other policies) we realize that there are many rewards equally "compatible" with $\pi^E$.[5] Clearly, wih no additional information, we are unable to identify $r^E$.

The feasible set considers only these rewards, i.e., $r \in \mathfrak{R}$ for which $J^*(r) = J^{\pi^E}(r)$, and it refuses all the others. This can be interpreted as the feasible set carrying out a *classification* of rewards based on a "hard" notion of *compatibility* with demonstrations. In other words, rewards $r$ satisfying condition $J^*(r) = J^{\pi^E}(r)$ are compatible with $\pi^E$, and the others are not. Nevertheless, our insight is that some rewards are *"more" compatible* with $\pi^E$ than others.

**Example 4.1.** *Consider an MDP with one state and $H = 1$ in which the expert has three actions: Eating a muffin (M), a cake (C), or some (bad) vegetable soup (S). The true reward $r^E$ assigns $r^E(M) = +1, r^E(C) = +0.99$ and $r^E(S) = -1$, i.e., the expert has a (weak) preference for the muffin over the cake, while she hates the soup; thus, she will demonstrate $\pi^E = M$. Let $r_g, r_b$ be:*

$$r_g(M) = +0.99, r_g(C) = +1, r_g(S) = -1, \qquad r_b(M) = -1, r_b(C) = -1, r_b(S) = +1.$$

*Intuitively, $r_g$ is "more" compatible with $\pi^E$ than $r_b$, because it establishes that M and C are much better than S, while reward $r_b$ reverses the preferences. Clearly, we make a small error if we model the preferences of the expert with $r_g$ instead of the true reward $r^E$. However, the notion of feasible set is completely blind to the difference between $r_g$ and $r_b$ at modeling $r^E$, and it refuses both of them.*

We propose the following "soft" definition of (non)compatibility to capture this intuition.[6]

**Definition 4.1** (Rewards (non)Compatibility). *Let $\mathcal{M} \cup \{\pi^E\}$ be an IRL instance, and let $r \in \mathfrak{R}$ be any reward. We define the* (non)compatibility $\overline{\mathcal{C}}_{p,\pi^E} : \mathfrak{R} \to \mathbb{R}_{\geqslant 0}$ *of reward $r$ w.r.t. $\mathcal{M} \cup \{\pi^E\}$ as:*

$$\overline{\mathcal{C}}_{p,\pi^E}(r) := J^*(r;p) - J^{\pi^E}(r;p).$$

---

[5]See Appendix C.1 for a visual intuition.

[6]In Appendix C.2, a *multiplicative* alternative definition is presented.

In words, the (non)compatibility of reward $r$ w.r.t. policy $\pi^E$ in problem $\mathcal{M}$ quantifies the *sub-optimality* of $\pi^E$ in the MDP $\mathcal{M} \cup \{r\}$. By definition, rewards $r$ belonging to the feasible set (i.e., $r \in \mathcal{R}_{p,\pi^E}$) satisfy $\overline{\mathcal{C}}_{p,\pi^E}(r) = 0$, i.e., they have zero non-compatibility with $\pi^E$ in $\mathcal{M}$.[7]

**Example 4.1** (Continued). *(Non)compatibility discriminates between $r_g$ and $r_b$. Indeed, we have that $\overline{\mathcal{C}}_{p,\pi^E}(r^E) = 0$, $\overline{\mathcal{C}}_{p,\pi^E}(r_g) = 0.01$, and $\overline{\mathcal{C}}_{p,\pi^E}(r_b) = 2$. In words, reward $r_g$ suffers from very small (non)compatibility, while $r_b$ suffers from large (non)compatibility, thus we say that reward $r_g$ is more compatible with $\pi^E$ than $r_b$, as expected.*

By definition of IRL, the true reward $r^E$ makes the observed $\pi^E$ optimal, but reveals no information about the other policies. Thus, it is meaningful that $\overline{\mathcal{C}}_{p,\pi^E}$ considers the suboptimality of $\pi^E$ only, because demonstrations from $\pi^E$ do not provide information about other policies, as illustrated below.

**Example 4.2.** *Let $r_b'$ be such that $r_b'(M) = +0.99, r_b'(C) = -1, r_b'(S) = +1$. Clearly, $r_b'$ is much worse than $r_g$ at modeling $r^E$, because it does not capture the fact that the expert appreciates the cake but she hates the soup. However, demonstrations from $\pi^E$ alone do not provide information about C or S, but only about $\pi^E = M$ (i.e., the expert always eats the muffin). Thus, we have that $\overline{\mathcal{C}}_{p,\pi^E}(r_g) = \overline{\mathcal{C}}_{p,\pi^E}(r_b') = 0.01$, i.e., $r_g$ and $r_b'$ are equally compatible with the given demonstrations.*

For a discussion on comparing the (non)compatibility of different rewards, see Appendix C.4.

## 4.2 The IRL Classification Formulation

Our goal is to overcome the limitations of the feasible set highlighted in Section 3. Drawing inspiration from the notion of "membership checker" algorithm in [31], we propose a novel formulation of IRL.

**Definition 4.2** (IRL Classification Problem and IRL Algorithm). *An IRL Classification Problem instance is made of a tuple $(\mathcal{M}, \pi^E, \mathcal{R}, \Delta)$, where $\mathcal{M}$ is an MDP without reward, $\pi^E$ is the expert's policy, $\mathcal{R} \subseteq \mathfrak{R}$ is a set of rewards to classify, and $\Delta \in \mathbb{R}_{\geqslant 0}$ is some threshold. The goal is to classify all and only the rewards $r \in \mathcal{R}$ based on their (non)compatibility with $\pi^E$ in $\mathcal{M}$ w.r.t. $\Delta$. In symbols:*

$$\forall r \in \mathcal{R} : \quad \textbf{if} \quad \overline{\mathcal{C}}_{p,\pi^E}(r) \leqslant \Delta \quad \textbf{then} \quad \textbf{return} \text{ True,} \quad \textbf{else return} \text{ False.}$$

*An IRL algorithm takes in input a reward $r \in \mathcal{R}$ and outputs a boolean saying whether $\overline{\mathcal{C}}_{p,\pi^E}(r) \leqslant \Delta$.*

Given $r \in \mathcal{R}$, we output whether it makes the expert's policy $\pi^E$ at most $\Delta$-suboptimal or not. Intuitively, we classify rewards in $\mathcal{R}$ based on how good $\pi^E$ performs w.r.t. them. A $\Delta$-(non)compatible reward guarantees that, among its $\Delta$-optimal policies, there is $\pi^E$, but the optimal policy might be different from $\pi^E$ (see Appendix C.3 for how this relates to (forward) RL). Note that we allow for $\mathcal{R} \neq \mathfrak{R}$ to manage scenarios in which we have some prior knowledge on $r^E$, i.e., $r^E \in \mathcal{R} \subset \mathfrak{R}$.

**Remark 4.1.** *Permitting non-zero (non)compatibility is equivalent to enlarging the feasible set. Let $\mathcal{R} = \mathfrak{R}$, and define the set of rewards positively classified as $\mathcal{R}_\Delta$, i.e., $\mathcal{R}_\Delta := \{r \in \mathcal{R} \,|\, \overline{\mathcal{C}}_{p,\pi^E}(r) \leqslant \Delta\}$. For any $\Delta, \Delta'$ s.t. $0 \leqslant \Delta \leqslant \Delta' \leqslant 2H$, we have: $\mathcal{R}_{p,\pi^E} = \mathcal{R}_0 \subseteq \mathcal{R}_\Delta \subseteq \mathcal{R}_{\Delta'} \subseteq \mathcal{R}_{2H} = \mathfrak{R}$.*

**Discussion on Reward Compatibility.** It should be remarked that:

- *The limits of the rewards compatibility framework are the same as the limits of the feasible set*. We cannot identify $r^E$ from the feasible set or among the rewards with small (non)compatibility. As aforementioned, this is an inherent limit of IRL and cannot be overcome with a more refined objective formulation, unless further information on $r^E$ is available (e.g., preferences).
- *Rewards compatibility offers advantages over feasible set*. Differently from the feasible set, as we will see in Section 5, it is possible to *practically* implement algorithms that solve the IRL classification problem, with guarantees of sample efficiency even when the state space is large.

## 4.3 A Learning Framework for Online IRL Classification

In this section, we combine the online IRL setting presented in Section 2 with the IRL classification problem of Definition 4.2. Intuitively, the performance of an algorithm depends on its accuracy at estimating the (non)compatibility of the rewards, as formalized by the following PAC requirement.

---

[7]We use *(non)compatibility* since a reward $r \in \mathfrak{R}$ is maximally compatible when $\overline{\mathcal{C}}_{p,\pi^E}(r) = 0$. Thus, the larger $\overline{\mathcal{C}}_{p,\pi^E}(r)$, the more $r$ is non-compatible. In this sense, $\overline{\mathcal{C}}_{p,\pi^E}(r)$ quantifies the non-compatibility of $r$.

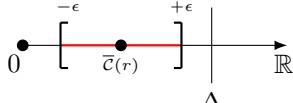 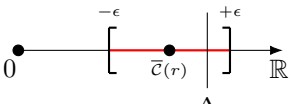 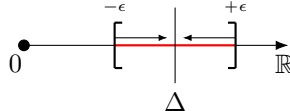

(a) Reward $r$ is classified correctly.

(b) Reward $r$ can be misclassified.

(c) Range of uncertain (non)compatibility values.

Figure 2: The axis represents (estimated) (non)compatibility values. (a) Rewards $r$ whose true (non)compatibility $\overline{C}(r) := \overline{C}_{p,\pi^E}(r)$ is far from threshold $\Delta$ by at least $\epsilon$, are correctly classified, while (b) in the opposite case, rewards can be mis-classified. (c) The red interval $[\Delta - \epsilon, \Delta + \epsilon]$ exemplifies the set of rewards $\{r \in \mathcal{R} \mid |\overline{C}(r) - \Delta| \leqslant \epsilon\}$ that are (potentially) mis-classified. The length of the interval reduces with $\epsilon$.

**Definition 4.3** (PAC Framework)**.** *Let $\epsilon, \delta \in (0, 1)$, and let $\mathcal{D}^E$ be a dataset of $\tau^E$ expert's trajectories. An algorithm $\mathfrak{A}$ exploring for $\tau$ episodes is $(\epsilon, \delta)$-PAC for the IRL classification problem if:*

$$\mathbb{P}_{\mathcal{M}, \pi^E, \mathfrak{A}} \left( \sup_{r \in \mathcal{R}} \left| \overline{C}_{p,\pi^E}(r) - \widehat{C}(r) \right| \leqslant \epsilon \right) \geqslant 1 - \delta,$$

*where $\mathbb{P}_{\mathcal{M}, \pi^E, \mathfrak{A}}$ is the joint probability measure induced by $\pi^E$ and $\mathfrak{A}$ in $\mathcal{M}$, and $\widehat{C}$ is the estimate of $\overline{C}_{p,\pi^E}$ computed by $\mathfrak{A}$. The* sample complexity *is defined by the pair $(\tau^E, \tau)$.*

Intuitively, our goal is to estimate the (non)compatibility of the rewards in $\mathcal{R}$ with sufficient accuracy, so that, given a threshold $\Delta \geqslant 0$, we are able to classify "most" of them correctly w.h.p. (with high probability). The concept is exemplified in Figure 2. Note that the estimation problem is independent of the threshold $\Delta$, which can be appropriately selected to cope with noise in the demonstrations, (unknown) expert suboptimality, or to manage the amount of "false negatives" and "false positives".

**Remark 4.2.** *For $\eta \geqslant 0$, let $\mathcal{R}_\eta := \{r \in \mathcal{R} \mid \overline{C}_{p,\pi^E}(r) \leqslant \eta\}$ and $\widehat{\mathcal{R}}_\eta := \{r \in \mathcal{R} \mid \widehat{C}(r) \leqslant \eta\}$ denote the sets of rewards positively classified using, respectively, the true (non)compatibility $\overline{C}_{p,\pi^E}$ and the estimate $\widehat{C}$ constructed by an $(\epsilon, \delta)$-PAC algorithm. Then, with probability $1 - \delta$, it holds that: $\widehat{\mathcal{R}}_{\Delta-\epsilon} \subseteq \mathcal{R}_\Delta \subseteq \widehat{\mathcal{R}}_{\Delta+\epsilon}$. Thus, we can trade-off the amount of "false negatives" (resp. "false positives") by, e.g., choosing the threshold $\Delta \leftarrow \Delta + \epsilon$ (resp. $\Delta \leftarrow \Delta - \epsilon$).*

# 5 CATY-IRL:
## A Provably Efficient Algorithm for IRL

In this section, we present CATY-IRL (CompATibilitY for IRL), a provably efficient algorithm for solving the *online* IRL *classification* problem. We consider three different kinds of structure for the MDPs: tabular MDPs, tabular MDPs with linear rewards, and Linear MDPs. Similarly to RFE, our online IRL classification setting is made of two phases: ($i$) an *exploration* phase, in which the algorithm explores the environment using the knowledge of $\mathcal{R}$ and of the expert's dataset $\mathcal{D}^E$ to collect samples about the dynamics of the MDP, and ($ii$) a *classification* phase, in which it performs the classification of a reward $r \in \mathcal{R}$ without interactions with the environment. A flow-chart is reported in Figure 1 (pseudocode in Appendix D).

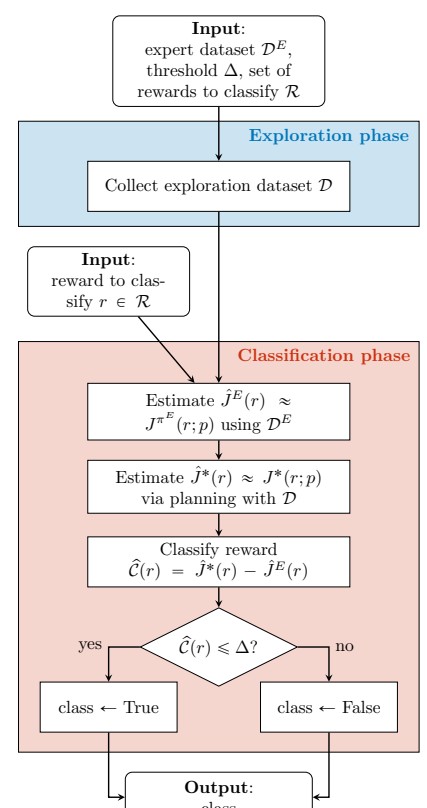

Figure 1: Flow-chart of CATY-IRL.

**Exploration phase.** The *exploration* phase collects a dataset $\mathcal{D}$ in a way that depends on the structure of the MDP and of the set of rewards $\mathcal{R}$ to be classified. Specifically, for Linear MDPs,

`CATY-IRL` executes RFLin [62]. Instead, for tabular MDPs (with or without linear reward), `CATY-IRL` instantiates either BPI-UCBVI [37] for each reward $r \in \mathcal{R}$ (when $|\mathcal{R}| = \Theta(1)$, i.e., a "small" constant w.r.t. to the size of the MDP, where "small" depends on the size of the state space, see Appendix D.2) or RF-Express [37]. Note that `CATY-IRL` in this phase does not use the expert's dataset $\mathcal{D}^E$.

**Classification phase.** The *classification* performs the estimation $\widehat{\mathcal{C}}(r)$ of the (non)compatibility term $\overline{\mathcal{C}}_{p,\pi^E}(r)$ for the single input reward $r \in \mathcal{R}$ by splitting it into two independent estimates: $\widehat{J}^E(r) \approx J^{\pi^E}(r;p)$, which is computed with $\mathcal{D}^E$ only, and $\widehat{J}^*(r) \approx J^*(r;p)$, which is computed with $\mathcal{D}$ only. Concerning $\widehat{J}^E(r)$, when the reward is linear $r_h(s,a) = \langle \phi(s,a), \theta_h \rangle$, `CATY-IRL` uses $\mathcal{D}^E$ to construct an empirical estimate $\widehat{\psi}^E \approx \psi^{p,\pi^E}$ of the expert's expected feature count [6]. Otherwise, it directly estimates $\widehat{d}^E \approx d^{p,\pi^E}$ the expert's occupancy measure. Such estimates can be used to derive $\widehat{J}^E(r)$ straightforwardly. Regarding $\widehat{J}^*(r)$, `CATY-IRL` exploits the *planning* phase of the corresponding RFE (or BPI) algorithm adopted at exploration phase.[8] Finally, `CATY-IRL` applies the (potentially negative) input threshold $\Delta$ to the difference $\widehat{J}^*(r) - \widehat{J}^E(r)$ to perform the classification. See Appendix D for the full pseudo-code. Clearly, `CATY-IRL` can be implemented in practice, since it considers a single reward at a time instead of computing the full feasible set, and it is computationally efficient in linear MDPs, since it uses a computationally efficient algorithm as subroutine (see [62]).

**Sample Efficiency.** The next result analyzes the sample complexity (Definition 4.3) of `CATY-IRL`.

**Theorem 5.1** (Sample Complexity of `CATY-IRL`). *Let $\epsilon, \delta \in (0,1)$. Then `CATY-IRL` is $(\epsilon, \delta)$-PAC for IRL with a sample complexity upper bounded by:*

*Tabular MDPs:*
$$\tau^E \leqslant \widetilde{\mathcal{O}}\Big(\frac{H^3 SA}{\epsilon^2} \log \frac{1}{\delta}\Big), \quad \tau \leqslant \widetilde{\mathcal{O}}\Big(\frac{H^3 SA}{\epsilon^2}\Big(N + \log \frac{1}{\delta}\Big)\Big),$$

*Tabular MDPs with linear rewards:*
$$\tau^E \leqslant \widetilde{\mathcal{O}}\Big(\frac{H^3 d}{\epsilon^2} \log \frac{1}{\delta}\Big), \quad \tau \leqslant \widetilde{\mathcal{O}}\Big(\frac{H^3 SA}{\epsilon^2}\Big(N + \log \frac{1}{\delta}\Big)\Big),$$

*Linear MDPs:*
$$\tau^E \leqslant \widetilde{\mathcal{O}}\Big(\frac{H^3 d}{\epsilon^2} \log \frac{1}{\delta}\Big), \quad \tau \leqslant \widetilde{\mathcal{O}}\Big(\frac{H^5 d}{\epsilon^2}\Big(d + \log \frac{1}{\delta}\Big)\Big),$$

*where $N = 0$ if $|\mathcal{R}| = \Theta(1)$, and $N = S$ otherwise.*

Some observations are in order. We conjecture that the $d^2$ dependence when $|\mathcal{R}| = \Theta(1)$ is unavoidable in Linear MDPs because of the lower bound for BPI in [62]. In tabular MDPs with deterministic expert, one might use the results in [66] to reduce the rate of $\tau^E$ from $\widetilde{\mathcal{O}}(SAH^3 \log(\delta^{-1})/\epsilon^2)$ to $\widetilde{\mathcal{O}}(SH^{3/2} \log(\delta^{-1})/\epsilon^2)$. Finally, note that the choice $\Delta = \epsilon$ allows us to positively classify all the rewards in the feasible set $\mathcal{R}_{p,\pi^E}$ w.h.p. and, in this case, other rewards positively classified have true (non)compatibility at most $2\epsilon$ w.h.p. In light of this result we conclude that *rewards compatibility* framework allows the *practical* development of *sample efficient* algorithms (e.g., `CATY-IRL`) in Linear MDPs with large/continuous state spaces.

## 6 Statistical Barriers and Objective-Free Exploration

In this section, we show that `CATY-IRL` is minimax optimal for the number of exploration episodes in tabular MDPs, and that RFE and IRL share the same theoretical sample complexity. This allows us to formulate *Objective-Free Exploration*, a unifying setting for exploration problems.

### 6.1 The Theoretical Limits of IRL (and RFE) in the Tabular Setting

In `CATY-IRL`, we use a minimax optimal RFE algorithm for exploration. However, this does not entail that `CATY-IRL` is minimax optimal for the IRL classification problem. There might exist another PAC algorithm with a sample complexity smaller than `CATY-IRL`. The following result states that, in the tabular setting, the bound in Theorem 5.1 is tight for the number of episodes $\tau$.

---

[8]RFE/BPI algorithms, at planning phase, return a policy, and not its estimated performance. Since BPI-UCBVI, RF-Express, and RFLin each compute an estimate of $J^*(r;p)$ as an intermediate step, with negligible abuse of notation, we assume that they output such estimate.

**Theorem 6.1** (IRL Classification - Lower Bound). *Let $\mathfrak{A}$ be an $(\epsilon, \delta)$-PAC algorithm for the IRL classification in tabular MDPs. Let $\tau$ be the number of exploration episodes. Then, there exists an IRL classification instance such that:*

$$\text{if } |\mathcal{R}| \geqslant 1: \ \tau \geqslant \Omega\left(\frac{H^3 S A}{\epsilon^2} \log \frac{1}{\delta}\right), \qquad \text{if } \mathcal{R} = \mathfrak{R}: \ \tau \geqslant \Omega\left(\frac{H^3 S A}{\epsilon^2}\left(S + \log \frac{1}{\delta}\right)\right).$$

In both cases, the lower bound is *matched* by `CATY-IRL`, up to logarithmic factors. Note that `CATY-IRL` explores without using $\mathcal{D}^E$, thus, minimax optimality for $\tau$ can be achieved without the knowledge of $\mathcal{D}^E$ at exploration phase. As a by-product, we observe that a similar lower bound construction can be made also for RFE, leading to the following result.

**Theorem 6.2** (RFE - Refined Lower Bound). *Let $\mathfrak{A}$ be an $(\epsilon, \delta)$-PAC algorithm for RFE in tabular MDPs. Let $\tau$ be the number of exploration episodes. Then, there exists an RFE instance such that:*

$$\tau \geqslant \Omega\left(\frac{H^3 S A}{\epsilon^2}\left(S + \log \frac{1}{\delta}\right)\right).$$

This bound improves the state-of-the-art RFE lower bound $\Omega\left(\frac{H^3 S A}{\epsilon^2}\left(\frac{S}{H} + \log \frac{1}{\delta}\right)\right)$ (obtained combining the bounds in [21] and [12]) by one $H$ factor, and it is matched by RF-Express [37].

## 6.2 Objective-Free Exploration (OFE)

What is the most efficient exploration strategy that can be performed in an unknown environment? It *depends* on the subsequent task that shall be solved. However, if the task is unknown at the exploration phase, we need a strategy that suffices for all the tasks that one might be interested in solving. Let us denote by $\mathscr{F}$ the set of RL and IRL classification tasks. Since `CATY-IRL` is a sample efficient algorithm for the IRL classification problem, and it uses RFE as a subroutine, we conclude that the RFE exploration strategy is sufficient (and also minimax optimal in tabular MDPs) to obtain guarantees for class $\mathscr{F}$. Are there other problems for which RFE exploration suffices when the specific problem instance is revealed *a posteriori* of the exploration phase? We believe so, and in Appendix E, we identify two additional problems, i.e., Matching Performance (MP) and Imitation Learning from Observations alone (ILfO) [34], that represent potential candidates to belong to $\mathscr{F}$.

More in general, we formulate the *Objective-Free Exploration (OFE)* problem as follows:

**Definition 6.1** (Objective-Free Exploration). *Given a tuple $(\mathcal{M}, \mathscr{F}, (\epsilon, \delta))$, where $\mathcal{M}$ is an unknown environment (e.g., MDP without reward), and $\mathscr{F}$ is a certain class of tasks (e.g., all RL and IRL problems), the* Objective-Free Exploration *(OFE) problem aims to find an exploration of the environment $\mathcal{M}$ (e.g., RFE exploration) that permits to solve* any task $f \in \mathscr{F}$ in an $(\epsilon, \delta)$-correct manner.

This problem is called "objective-free" because it does not require the knowledge of the specific "objective" $f \in \mathscr{F}$ to be solved. In Appendix F, we describe a use case for OFE. We believe this is an interesting problem to be studied in future.

## 7 Conclusions

In this paper, we have shown that the feasible set cannot be learned efficiently in problems with large/continuous state spaces even under the strong structure provided by Linear MDPs. For this reason, we have introduced the powerful framework of *compatible rewards*, which formalizes the intuitive notion of compatibility of a reward function with expert demonstrations, and it allows us to formulate the IRL problem as a *classification* task. In this context, we have devised `CATY-IRL`, a provably efficient IRL algorithm for Linear MDPs with large/continuous state spaces. Furthermore, in tabular MDPs, we have demonstrated the minimax optimality of `CATY-IRL` at exploration by presenting a novel lower bound to the IRL classification problem. As a by-product, our construction improves the current state-of-the-art lower bound for RFE. Finally, we have introduced OFE, a unifying problem setting for exploration problems, which generalizes both RFE and IRL.

**Limitations.** A limitation of our contributions concerns the adoption of the *Linear MDP* model, whose assumptions are overly strong to be consistently applied to real-world applications. Nevertheless, while the rewards compatibility framework is general and not tied to Linear MDPs, we believe that Linear MDPs represent an important initial step toward the development of provably efficient IRL

algorithms with more general function approximation structures. Although a lower bound for Linear MDPs is missing, we believe that it represents an interesting direction for future works. Finally, we note that the *empirical validation* of the proposed algorithm is out of the scope of this work.

**Future Directions.** Promising directions for future works concern the extension of the analysis of the *rewards compatibility* framework beyond Linear MDPs to general function approximation and to the offline setting. In addition, it might be fascinating to extend the notion of reward compatibility to other kinds of expert feedback (in the context of ReL), and to other IRL settings (e.g., suboptimal experts). Finally, we believe that OFE should be analysed in-depth given its practical importance.

## Acknowledgments and Disclosure of Funding

AI4REALNET has received funding from European Union's Horizon Europe Research and Innovation programme under the Grant Agreement No 101119527. Views and opinions expressed are however those of the author(s) only and do not necessarily reflect those of the European Union. Neither the European Union nor the granting authority can be held responsible for them.

Funded by the European Union - Next Generation EU within the project NRPP M4C2, Investment 1.,3 DD. 341 - 15 march 2022 - FAIR - Future Artificial Intelligence Research - Spoke 4 - PE00000013 - D53C22002380006.

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

# A Related Works

In this appendix, we report and describe the literature that most relates to this paper. Theoretical works concerning the online IRL problem can be grouped in works that concern the feasible set, and works that do not.

Let us begin with works related to the feasible set. While the notion of feasible set has been introduced implicitly in [42], the first paper that analyses the sample complexity of estimating the feasible set in online IRL is [39]. Authors in [39] adopt the simple generative model in tabular MDPs, and devise two sample efficient algorithms. [33] focuses on the same problem as [39], but adopts a forward model in tabular MDPs. By adopting RFE exploration algorithms, they devise sample efficient algorithms. However, as remarked in [68], paper [33] suffers from a limitation in the definition of the dissimilarity between feasible sets. [38] builds upon [39] to construct the first minimax lower bound for the problem of estimating the feasible set using a generative model. The lower bound is in the order of $\Omega\big(\frac{H^3 S A}{\epsilon^2}(S + \log\frac{1}{\delta})\big)$, where $S$ and $A$ are the cardinality of the state and action spaces, $H$ is the horizon, $\epsilon$ is the accuracy and $\delta$ the failure probability. In addition, [38] develops US-IRL, an efficient algorithm whose sample complexity matches the lower bound. [44] analyze a setting analogous to that of [38], in which there is availability of a single optimal expert and multiple suboptimal experts with known suboptimality. [31] analyse the problem of estimating the feasible set when no active exploration of the environment is allowed, but the learner is given a batch dataset collected by some behavior policy $\pi^b$. Interestingly, [31] focuses on two novel learning targets that are suited for the offline setting, i.e., a subset and a superset of the feasible set. Authors in [31] demonstrate that such sets are the tightest learnable subset and superset of the feasible set, and propose a pessimistic algoroithm, PIRLO, to estimate them. [68] analyses the same offline setting as [31], but instead of focusing on the notion of feasible set directly, it considers the notion of reward mapping, which considers reward functions as parametrized by their value and advantage functions, and whose image coincides with the feasible set.

With regards to online IRL works that do not consider the feasible set, we mention [36], which analyses an active learning framework for IRL. However, [36] assumes that the transition model is known, and its goal is to estimate the expert policy only. Works [28] and [29] provide, respectively, an upper bound and a lower bound to the sample complexity of IRL for $\beta$-strict separable problems in the tabular setting. However, both the setting considered and the bound obtained are fairly different from ours. Analogously, [11] provides a sample efficient IRL algorithm for $\beta$-strict separable problems with continuous state space. However, their setting is different from ours since they assume that the system can be modelled using a basis of orthonormal functions.

## A.1 Additional Related Works

In this section, we collect additional related works that deserve to be mentioned.

**Identifiability and Reward Learning.** As aforementioned, the IRL problem is ill-posed, thus, to retrieve a single reward, additional constraints shall be imposed. [5] analyses the setting in which demonstrations of an optimal policy for the same reward function are provided across environments with different transition models. In this way, authors can reduce the experimental unidentifiability, and recover the state-only reward function. [10] and [26] concern reward identifiability but in entropy-regularized MDPs [70, 15]. Such setting is in some sense easier than the common IRL setting, because entropy-regularization permits a unique optimal policy for any reward function. [10] uses expert demonstrations from multiple transition models and multiple discount factors to retrieve the reward function, while [26] analyses properties of the dynamics of the MDP to increase the constraints. With regards to the more general field of Reward Learning (ReL), we mention [19], which introduces a framework that formalizes the constraints imposed by various kinds of human feedback (like demonstrations or preferences [65]). Intuitively, multiple feedbacks about the same reward represent additional constraints beyond mere demonstrations. [56] characterizes the partial identifiability of the reward function based on various reward learning data sources.

**Linear MDPs and Extensions.** As explained for instance in [23], since lower bounds to the sample complexity of various RL tasks in tabular MDPs depend explicitly on the cardinality state space $S$, then we need to add structure to the problem if we want to develop efficient algorithms that scale to large state spaces. For this reason, the works [67, 23] analyze the Linear MDP model, which enforces

some linearity constraints to the common MDP model. In this way, authors are able to provide efficient algorithms for RL in problems with large/continuous state spaces. However, there are other settings beyond Linear MDPs that are analysed in the RL literature. [20] introduces the notion of Bellman rank as complexity measure, and provides a sample efficient algorithm for problems with small Bellman rank. [64] analyzes general value function approximation when the function class has a low eluder dimension. [22] generalizes both the eluder dimension and Bellman rank complexity measures by defining the Bellman eluder dimension and providing a provably efficient algorithm. [13] introduces bilinear classes, a structural framework that, among the others, generalizes Linear MDPs.

**Reward-Free Exploration (RFE) in Tabular and Linear MDPs.** The RFE problem was introduced in [21], where authors provided a sample efficient algorithm and a lower bound for tabular MDPs. Later on, the state-of-the-art sample-efficient algorithms for RFE in tabular MDPs have been developed in [24, 37, 32]. It should be remarked that RFE requires more samples than common RL in tabular MDPs. [63] proposes a sample efficient algorithm for RFE in linear MDPs. [62] improves the algorithm of [63] and, interestingly, demonstrates that RFE is no harder than RL in Linear MDPs.

**Online Apprenticeship Learning (AL).** The first works that provide a theoretical analysis of the AL setting when the transition model is unknown are [3, 60]. Recently, [53] formulates the online AL problem, which closely resembles the online IRL problem. The main difference is that in online AL the ultimate goal is to imitate the expert, while in IRL is to recover a reward function. [66] improves the results in [53] by combining an RFE algorithm with an efficient algorithm for the estimation of the visitation distribution of the deterministic expert's policy in tabular MDPs, presented in [47]. We mention also [46, 59] for the sample complexity of estimating the expert's policy in problems with linear function approximation. In the context of Imitation Learning from Observation alone (ILfO) [34], the work [57] proposes a probably efficient algorithm for large-scale MDPs with unknown transition model. [35] provides an efficient AL algorithm based on GAIL [18] in Linear Kernel Episodic MDPs [69] with unknown transition model.

**Others.** We mention work [27], which considers a classification approach for IRL. However, this is fairly different from our IRL problem formulation in Section 4.

# B  Additional Results and Proofs for Section 3

In this section, we provide additional results beyond those presented in Section 3, and then we report the missing proofs. Specifically, in Appendix B.1, we provide two numerical examples that explain Proposition 3.1, in Appendix B.2 we show that some additional regularity assumptions beyond the Linear MDP cannot remove the dependence on the cardinality of the state space in the sample complexity. In Appendix B.3, we report and describe the sample efficient algorithm mentioned in Section 3, while in Appendix B.4 we collect all the missing proofs of this section.

## B.1  Some Examples for Proposition 3.1

The following examples aim to explain Proposition 3.1 in a simple manner.

**Example B.1** (Non-degenerate feasible set). *Let $\mathcal{M} = (\mathcal{S}, \mathcal{A}, H, d_0) \cup \{\pi^E\}$ be an IRL instance such that $\mathcal{S} = \{s_1, s_2\}, \mathcal{A} = \{a_1, a_2\}, H = 1, d_0(s_1) = d_0(s_2) = 1/2, \pi^E(s_1) = \pi^E(s_2) = a_1$. Consider the feature mapping $\phi_1$ s.t. $\phi_1(s, a) = \mathbb{1}\{a = a_1\}$ for all $s \in \mathcal{S}$. Then, we have $\Phi^{\pi^E} = \{1\}$ and $\overline{\Phi} = \{0\}$. Clearly, these sets can be separated by any hyperplane $w \in \mathbb{R}_{>0}$, since $1 \cdot w > 0 \cdot w$, and so $\mathcal{R}_{p,\pi^E} \neq \{\overline{r}\}$, with $\overline{r}_h(s, a) = 0 \ \forall(s, a, h) \in \mathcal{S} \times \mathcal{A} \times [\![H]\!]$. Actually, $\mathcal{R}_{p,\pi^E} = \{r \in \mathfrak{R} \mid \exists \theta \in (0, 1] : r_1(s, a) = \langle \phi(s, a), \theta \rangle \ \forall(s, a) \in \mathcal{S} \times \mathcal{A}\}$.*

**Example B.2** (Degenerate feasible set). *Consider the same IRL instance as in the previous example, but this time consider the feature mapping $\phi_2$ s.t. $\phi_2(s_1, a) = \mathbb{1}\{a = a_1\}$, and $\phi_2(s_2, a) = \mathbb{1}\{a = a_2\}$. Then, we have $\Phi^{\pi^E} = \{0, 1\}$ and $\overline{\Phi} = \{0, 1\}$. Clearly, the two sets coincide, thus they cannot be separated, and $\mathcal{R}_{p,\pi^E} = \{\overline{r}\}$, with $\overline{r}_h(s, a) = 0 \ \forall(s, a, h) \in \mathcal{S} \times \mathcal{A} \times [\![H]\!]$.*

## B.2 Additional Regularity Assumptions of the State Space do not Make the Problem Learnable

In tabular MDPs with small state space $\mathcal{S}$, collecting samples from every state $s \in \mathcal{S}$ is feasible, and it is exactly what previous works do:

- Under the assumption that $\pi^E$ is deterministic, [39, 38] collect one sample from every $(s, h) \in \mathcal{S} \times [\![H]\!]$ using a generative model, obtaining $\pi^E$ exactly.

- If $\pi^E$ is stochastic, under the assumption that all actions in the support of the expert's policy are played with probability at least $\pi_{\min}$ (see Assumption D.1 of [38]), both [38, 68] are able to learn the support of $\pi^E$ exactly w.h.p. using $\propto 1/\pi_{\min}$ samples in the online setting.[9]

- In the offline setting, assuming that the occupancy measure of the expert's policy is at least $d_{\min}$ in all reachable $(s, a) \in \mathcal{S} \times \mathcal{A}$, then [31] learns the support of $\pi^E$ exactly w.h.p. using $\propto 1/d_{\min}$ episodes.

However, when $\mathcal{S}$ is large, even under the Linear MDP assumption, this is not possible. In Section 3, we have formalized this fact with the following proposition:

**Theorem 3.2** (Statistical Inefficiency). *Let $\mathcal{M} \cup \{\pi^E\}$ be a Linear IRL instance with finite state space $\mathcal{S}$ and deterministic expert's policy, and let $\epsilon, \delta \in (0, 1)$. If an algorithm $\mathfrak{A}$ is $(\epsilon, \delta)$-PAC, then $\tau^E = \Omega(S)$, where $S := |\mathcal{S}|$ is the cardinality of the state space.*

Theorem 3.2 tells us that the Linear MDP assumption is too weak for the feasible set to be learnable using the PAC framework of Definition 3.2 with a number of samples independent of the cardinality of the state space. Therefore, we can try to introduce an additional assumption on the structure of the IRL problem $\mathcal{M} \cup \{\pi^E\}$ and see whether it helps in alleviating the issue. Let us consider the following first assumption.

**Assumption B.1.** *We assume a Lipschitz continuity property between features and states:*

$$\forall (s, a, s') \in \mathcal{S} \times \mathcal{A} \times \mathcal{S}: \quad \|\phi(s, a) - \phi(s', a)\|_2 \leqslant L\|s - s'\|,$$

*for some $L > 0$ and some distance $\|\cdot - \cdot\|$ in $\mathcal{S}$.*

The intuition is that, based on the fact that in Linear MDPs the $Q$-function of any policy $\pi$ is linear in the feature mapping $Q_h^\pi(\cdot, \cdot) = (\phi(\cdot, \cdot), w_h^\pi)$ for some parameter vector $w_h^\pi \in \mathbb{R}^d$ (see [23]), then if we are able to $\epsilon$-cover the state space $\mathcal{S}$, we can approximate the $Q$-function $Q_h^\pi(s, \cdot)$ in any $s \in \mathcal{S}$ with the $Q$-function $Q_h^\pi(s', \cdot)$ of the closest point $s'$ int the covering, so that $|Q_h^\pi(s, a) - Q_h^\pi(s', a)| = |(\phi(s, a) - \phi(s', a))^\mathsf{T} w_h^\pi| \leqslant \|\phi(s, a) - \phi(s', a)\|_2 \|w_h^\pi\|_2 \leqslant L\epsilon\|w_h^\pi\|_2$. However, this assumption is not sufficient.

**Proposition B.1.** *Under the setting of Proposition 3.2, even under Assumption B.1, then an algorithm is $(\epsilon, \delta)$-PAC only if $\tau^E = \Omega(S)$.*

Assumption B.1 fails because it does not provide any information about how the knowledge of the expert's policy at a state can be "transferred" to other states, and thus we still need to sample almost all the states of $\mathcal{S}^{p, \pi^E}$ to get an acceptable feasible set.

We devise another assumption to attempt to fix this issue.

**Assumption B.2.** *We assume the following Lipschitz continuity property:*

$$\forall (s, s') \in \mathcal{S} \times \mathcal{S}: \quad \|\phi(s, \pi_h^E(s)) - \phi(s', \pi_h^E(s'))\|_2 \leqslant L\|s - s'\|,$$

*for some $L > 0$ and some distance $\|\cdot - \cdot\|$ in $\mathcal{S}$.*

This assumption says that states that are close to each other cannot have the features corresponding to the expert's action too far away from each other. From a high-level point of view, it says that the features are "somehow" regular with $\pi^E$, so that when the expert lies in $s'$ which is really close to $s$, then she plays an action which has the same "effect" (i.e., same transition model and same reward, due to the Linear MDP assumption) as the expert's action in $s$.

Assumption B.2 is not comparable with Assumption B.1 since, on the one hand, it does not hold for all actions in $\mathcal{A}$, but only for those corresponding to $\pi^E$, but, on the other hand, provides information on how to transfer knowledge about $\pi^E$ to neighbor states.

---

[9]Actually, [68] makes use of a concentrability assumption too.

Let $\Delta' := \min_{s \in \mathcal{S}, a, a' \in \mathcal{A}: \phi(s,a) \neq \phi(s,a')} \|\phi(s,a) - \phi(s,a')\|_2$, i.e., the smallest non-zero distance between the features of different actions. Clearly, when $\mathcal{S}$ is finite, since in Linear MDPs also $A := |\mathcal{A}|$ is finite, then $\Delta'$ is finite too. So we can define a new quantity $\Delta$ to be any number $0 < \Delta < \Delta'$.

**Proposition B.2.** *Under the setting of Proposition 3.2, under Assumption B.2, then a number of samples $\tau^E = |\mathcal{N}(\frac{\Delta}{2L}; \mathcal{S}, \|\cdot\|)|$ is sufficient to recover $\pi^E$ exactly in any $(s,h) \in \mathcal{S}$, where $|\mathcal{N}(\frac{\Delta}{2L}; \mathcal{S}, \|\cdot\|)|$ is the $\Delta/(2L)$-covering number of space $\mathcal{S}$ w.r.t. distance $\|\cdot\|$.*

Intuitively, by constructing a covering with a sufficiently small radius in the state space $\mathcal{S}$, then we are able to retrieve the exact expert's action in the neighborood of each state of the covering. Doing so, we are able to construct $\epsilon$-correct estimates of the feasible set. Of course, this is possible as long as $\Delta'$ is not too small, and $L$ is not too large. When $\mathcal{S}$ is infinitely large or continuous, it might be possible to construct feature mappings in which $\Delta' \to 0$, and so the approach would still require too many samples.

However, even for cases with finite and not too small $\Delta'$, the result in Proposition B.2 is not satisfactory, because it just allows to retrieve $\pi^E$ under a stronger assumption than Linear MDPs, but not to perform an interesting learning process. We observe that the feasible set is an "unstable" concept, in the sense that, based on Proposition 3.1, changing the expert action in a single state might reduce the feasible set from a continuum of rewards to a singleton, or vice versa.

**Remark B.1.** *If we want to be able to recover the exact feasible set efficiently, we need to recover the exact expert's policy almost everywhere.*

### B.3  Algorithm

By exploiting an RFE algorithm as sub-routine like that of Algorithm 1 in [63] or Algorithm 1 in [62], we are able to construct estimates of the transition model $\widehat{p}$, that can be used to compute an "empirical" estimate of the feasible set $\widehat{\mathcal{R}} \approx \mathcal{R}_{\widehat{p}, \pi^E}$ (since $\phi$ and $\pi^E$ are known). The algorithm is presented in Algorithm 1.

---

**Algorithm 1:** IRL for Linear MDPs (known expert's policy)

**Data:** failure probability $\delta > 0$, error tolerance $\epsilon > 0$, expert policy $\pi^E$, all sets $\mathcal{Z} \subseteq \mathcal{S} \times [\![H]\!]$
that coincide with $\mathcal{S}^{p,\pi^E}$ almost everywhere based on measure $d^{p,\pi^E}$

1  $\mathcal{D} \leftarrow \text{RFE\_Exploration}(\delta, \epsilon)$                    /* Various choices */
2  **for** $h$ *in* $\{H, H-1, \ldots, 2, 1\}$ **do**
3  $\quad \Lambda_h \leftarrow I + \sum\limits_{k=1}^{\tau} \phi(s_h^k, a_h^k)\phi(s_h^k, a_h^k)^\intercal$
4  $\quad \widehat{\mu}_h(\cdot) \leftarrow \Lambda_h^{-1} \sum\limits_{k=1}^{\tau} \phi(s_h^k, a_h^k)\delta(\cdot, s_{h+1}^k)$
5  **end**
6  $\widehat{p}_h(\cdot|s,a) \leftarrow \langle \phi(s,a), \widehat{\mu}_h(\cdot) \rangle$ for all $(s,a,h) \in \mathcal{S} \times \mathcal{A} \times [\![H]\!]$
7  $\widehat{\mathcal{R}} \leftarrow \left\{ \widehat{r} \in \mathfrak{R} \,\middle|\, \exists \mathcal{Z}, \forall(s,h) \in \mathcal{Z}, \forall a \in \mathcal{A} : \underset{a' \sim \pi_h^E(\cdot|s)}{\mathbb{E}} Q_h^*(s, a'; \widehat{p}, \widehat{r}) \geqslant Q_h^*(s, a; \widehat{p}, \widehat{r}) \right\}$
8  Return $\widehat{\mathcal{R}}$

---

Simply put, Algorithm 1 uses the dataset collected by an RFE algorithm to compute a least-squares estimate of the transition model $\widehat{p}$, and then it returns the feasible set defined according to it (recall that $\phi$ and $\pi^E$ are known). Notice that this algorithm cannot be implemented in practice due to various reasons, like the presence of the Dirac delta $\delta$ measure in the definition of some quantities (see Appendix B.4.3), and the fact that the feasible set is, potentially, a set containing infinite rewards. Nevertheless, Theorem 3.3 states that this algorithm is sample efficient. The proof of the theorem is provided in Appendix B.4.3.

It should be remarked that Algorithm 1 takes in input also the true support of the visit distribution of the expert policy $\mathcal{S}^{p,\pi^E}$ in case $\mathcal{S}$ is finite, and all the possible sets $\mathcal{Z}$ that agree with $\mathcal{S}^{p,\pi^E}$ a.e. based on the measure $d^{p,\pi^E}$ in case $\mathcal{S}$ is infinite. Intuitively, this set ($\mathcal{S}^{p,\pi^E}$) of $(s,h)$ pairs represents

the domain in which $\pi^E$ is defined. Indeed, since the expert in the true problem $p$ never visits pairs $(s', h') \notin \mathcal{S}^{p,\pi^E}$, its expert policy might reasonably be non well-defined there. When $\mathcal{S}$ is infinite, we require all sets $\mathcal{Z}$ because otherwise we cannot know which are the sets $\mathcal{S}^{p,\pi^E} \setminus \mathcal{Z}$ with zero measure, i.e., in which the reward can induce an optimal action different from the expert's one, since the overall contribution to the expected return is zero.

The proof of Theorem 3.3 is obtained by using Algorithm 1 of [62] at Line 1 of Algorithm 1. In Appendix B.4.3, we demonstrate an upper bound also if we use Algorithm 1 in [63].

## B.4 Missing Proofs

Before diving into the proofs, we recall some important properties of the feasible set and of the Linear MDPs that will be useful in the proofs. First, we provide an explicit form for the feasible set presented at Definition 3.1.

**Lemma B.3** (Lemma E.1 in [31]). *In the setting of Definition 3.1, if $\mathcal{S}$ is finite, then the feasible set $\mathcal{R}_{p,\pi^E}$ satisfies:*

$$\mathcal{R}_{p,\pi^E} = \left\{ r \in \mathfrak{R} \,\middle|\, \forall (s,h) \in \mathcal{S}^{p,\pi^E}, \forall a \in \mathcal{A} : \underset{a' \sim \pi_h^E(\cdot|s)}{\mathbb{E}} Q_h^*(s,a';p,r) \geqslant Q_h^*(s,a;p,r) \right\}.$$

Notice that we have extended Lemma E.1 in [31] to consider stochastic expert policies (the extension is trivial). We can easily extend it to problems with large/continuous $\mathcal{S}$.

**Lemma B.4** (Feasible Set Explicit). *In the setting of Definition 3.1, then the feasible set $\mathcal{R}_{p,\pi^E}$ satisfies:*

$$\mathcal{R}_{p,\pi^E} = \left\{ r \in \mathfrak{R} \,\middle|\, \forall h \in [\![H]\!], \exists \overline{\mathcal{S}} \subseteq \mathcal{S}_h^{p,\pi^E} : d_h^{p,\pi^E}(\overline{\mathcal{S}}) = 0 \wedge \forall s \notin \overline{\mathcal{S}}, \forall a \in \mathcal{A} : \right.$$
$$\left. \underset{a' \sim \pi_h^E(\cdot|s)}{\mathbb{E}} Q_h^*(s,a';p,r) \geqslant Q_h^*(s,a;p,r) \right\}.$$

Simply, Lemma B.4 improves on Lemma B.3 by allowing the reward to enforce the "wrong" action (i.e., different from the expert's action) in a subset with zero measure based on the visitation distribution.

*Proof.* The proof is completely analogous to that of Lemma E.1 in [31]. We just need to observe that if set $\overline{\mathcal{S}}$ has zero measure (and the set of rewards $\mathfrak{R}$ contains bounded rewards), then it does not affect the expected return. $\square$

Another useful property that we need is that the $Q$-function is always linear in the feature map for any policy in Linear MDPs.

**Proposition B.5** (Proposition 2.3 in [23]). *For a Linear MDP, for any policy $\pi$, there exist weights $\{w_h^\pi\}_{h \in [\![H]\!]}$ such that, for any $(s,a,h) \in \mathcal{S} \times \mathcal{A} \times [\![H]\!]$, we have $Q_h^\pi(s,a) = \langle \phi(s,a), w_h^\pi \rangle$.*

We can combine the results of Lemma B.4 and Proposition B.5 to obtain the following characterization of the feasible set in Linear MDPs.

**Lemma B.6.** *In the setting of Definition 3.1, the feasible set $\mathcal{R}_{p,\pi^E}$ satisfies:*

$$\mathcal{R}_{p,\pi^E} = \left\{ r \in \mathfrak{R} \,\middle|\, \exists \{w_h\}_{h \in [\![H]\!]}, \forall (s,a,h) \in \mathcal{S} \times \mathcal{A} \times [\![H]\!] : r_h(s,a) = \langle \phi(s,a), \theta_h \rangle \right.$$
$$\wedge \forall h \in [\![H]\!], \exists \overline{\mathcal{S}} \subseteq \mathcal{S}_h^{p,\pi^E} : d_h^{p,\pi^E}(\overline{\mathcal{S}}) = 0 \wedge \forall s \notin \overline{\mathcal{S}}, \forall a^E \in \mathcal{A}_h^E(s) :$$
$$\left. \langle \phi(s,a^E), w_h \rangle = \max_{a \in \mathcal{A}} \langle \phi(s,a), w_h \rangle \right\},$$

*where $\theta_h := w_h - \int_{\mathcal{S}} \max_{a' \in \mathcal{A}} \langle \phi(s',a'), w_{h+1} \rangle d\mu_h(s')$ for all $h \in [\![H]\!]$, and $\mathcal{A}_h^E(s) := \{a \in \mathcal{A} | \pi_h^E(a|s) > 0\}$.*

*Proof.* From [45], we know that in any MDP there exists an optimal policy. Therefore, thanks to Proposition B.5, we know that the optimal $Q$-function $Q^*$ is linear in the feature map too. So, there

exist parameters $\{w_h\}_h$ such that, for any $(s, a, h) \in \mathcal{S} \times \mathcal{A} \times [\![H]\!]$, the optimal $Q$-function can be rewritten as $Q_h^*(s, a) = \langle \phi(s, a), w_h \rangle$. From the Bellman equation, we know that:

$$Q_h^*(s, a; p, r) = r_h(s, a) + \int_{\mathcal{S}} V_{h+1}^*(s'; p, r) dp_h(s'|s, a)$$

$$= \langle \phi(s, a), \theta_h \rangle + \langle \phi(s, a), \int_{\mathcal{S}} \max_{a' \in \mathcal{A}} \langle \phi(s', a'), w_{h+1} \rangle d\mu_h(s') \rangle.$$

By rearranging this equation, and removing the dot product with $\phi(s, a)$, we obtain that:

$$\theta_h = w_h - \int_{\mathcal{S}} \max_{a' \in \mathcal{A}} \langle \phi(s', a'), w_{h+1} \rangle d\mu_h(s').$$

Now, this holds in any Linear MDP. If we desire to enforce the constraints in Lemma B.4, we simply have to impose the constraint on the optimal $Q$-function using parameters $\{w_h\}_h$ outside some $\overline{\mathcal{S}}$. This concludes the proof. $\qquad \square$

It is useful to introduce the following definitions. First we define the set of parameters that induce a $Q$-function compatible with $\pi^E$:

$$\mathcal{W}_{p, \pi^E} := \left\{ w : [\![H]\!] \to \mathbb{R}^d \,\middle|\, \forall h \in [\![H]\!], \exists \overline{\mathcal{S}} \subseteq \mathcal{S}_h^{p, \pi^E} : d_h^{p, \pi^E}(\overline{\mathcal{S}}) = 0 \wedge \forall s \notin \overline{\mathcal{S}}, \forall a^E \in \mathcal{A}_h^E(s) : \right.$$

$$\left. \langle \phi(s, a^E), w_h \rangle = \max_{a \in \mathcal{A}} \langle \phi(s, a), w_h \rangle \right\}.$$

Next, we define the set of parameters of the reward function obtained by using $Q$-functions parametrized by $w \in \mathcal{W}_{p, \pi^E}$:

$$\Theta_{p, \pi^E} := \left\{ \theta : [\![H]\!] \to \mathbb{R}^d \,\middle|\, \exists \{w_h\}_h \in \mathcal{W}_{p, \pi^E} : \theta_h = w_h - \int_{\mathcal{S}} \max_{a' \in \mathcal{A}} \langle \phi(s', a'), w_{h+1} \rangle d\mu_h(s') \right\}.$$

Irrespective of the transition model $\{\mu_h\}_h$ and the feature map $\phi$, we see that it is always possible to construct a surjective map from $\Theta_{p, \pi^E}$ to $\mathcal{W}_{p, \pi^E}$ (the map in the definition of $\Theta_{p, \pi^E}$). Thanks to these definitions, the feasible set can be rewritten as:

$$\mathcal{R}_{p, \pi^E} = \{ r \in \mathfrak{R} \,|\, \exists \{\theta_h\}_h \in \Theta_{p, \pi^E}, \forall (s, a, h) \in \mathcal{S} \times \mathcal{A} \times [\![H]\!] : r_h(s, a) = \langle \phi(s, a), \theta_h \rangle \}.$$

We are now ready to provide the proofs of the various results of this section.

### B.4.1 Proof of Proposition 3.1

**Proposition 3.1.** *Let $\mathcal{M}$ be a Linear MDP without reward with a finite state space, and let $\phi$ be a feature mapping. Let $\{\Phi_h^{\pi^E}\}_{h \in [\![H]\!]}$ and $\{\overline{\Phi}_h\}_{h \in [\![H]\!]}$ be the sets of expert's and non-expert's features, defined for every $h \in [\![H]\!]$ as:*

$$\Phi_h^{\pi^E} := \{ \phi(s, a^E) \,|\, s \in \mathcal{S}_h^{p, \pi^E}, \, a^E \in \mathcal{A}_h^E(s) \}, \qquad \overline{\Phi}_h := \{ \phi(s, a) \,|\, s \in \mathcal{S}_h^{p, \pi^E}, \, a \in \mathcal{A} \backslash \mathcal{A}_h^E(s) \},$$

*where $\mathcal{A}_h^E(s) := \{ a \in \mathcal{A} | \pi_h^E(\cdot|s) > 0 \}$ for every $s \in \mathcal{S}$. If for none of the $H$ pairs of sets $(\Phi_h^{\pi^E}, \overline{\Phi}_h)$ there exists a separating hyperplane, then $\mathcal{R}_{p, \pi^E} = \{\overline{r}\}$, with $\overline{r}_h(s, a) = 0 \,\forall (s, a, h) \in \mathcal{S} \times \mathcal{A} \times [\![H]\!]$ i.e., the feasible set with linear rewards in $\phi$ contains only the reward function that assigns zero reward everywhere.*

*Proof.* From [8], we recall that two sets $\mathcal{Y}_1, \mathcal{Y}_2$ are separated by a hyperplane $H = \{x | a^\mathsf{T} x = b\}$ if each lies in a different closed halfspace associated with $H$, i.e., if either:

$$a^\mathsf{T} y_1 \leqslant b \leqslant a^\mathsf{T} y_2, \quad \forall y_1 \in \mathcal{Y}_1, \forall y_2 \in \mathcal{Y}_2,$$

or:

$$a^\mathsf{T} y_2 \leqslant b \leqslant a^\mathsf{T} y_1, \quad \forall y_1 \in \mathcal{Y}_1, \forall y_2 \in \mathcal{Y}_2.$$

By definition of $\mathcal{W}_{p, \pi^E}$, for each stage $h \in [\![H]\!]$, we are looking for vectors $w_h \in \mathbb{R}^d$ such that $\forall (s, h) \in \mathcal{S}^{p, \pi^E}$, it holds that:

$$w_h^\mathsf{T} \phi(s, a) \leqslant w_h^\mathsf{T} \phi(s, a^E) \quad \forall a^E \in \mathcal{A}_h^E(s), \forall a \in \mathcal{A} \backslash \mathcal{A}_h^E(s).$$

In words, for each $(s,h) \in \mathcal{S}^{p,\pi^E}$, we are looking for non-affine separating hyperplanes between features of expert and non-expert actions. However, since the hyperplane parameter $w_h$ is common to all states $s \in \mathcal{S}_h^{p,\pi^E}$, then it must separate expert from non-expert actions at all states. This is equivalent to finding the separating hyperplanes to the sets $\Phi_h^{\pi^E}$ and $\overline{\Phi}_h$ which contain all the points. Clearly, when the separating hyperplanes do not exist at all $h \in [\![H]\!]$, then the condition in $\mathcal{W}_{p,\pi^E}$ is satisfied by the zero vector alone. As a consequence, set $\Theta_{p,\pi^E}$ contains only the zero vector, and so does $\mathcal{R}_{p,\pi^E}$. $\qquad\square$

**Remark B.2.** *By using the result of Lemma B.4, we can easily convert Proposition 3.1 into a more general result by considering the impossibility of separating any pair of sets constructed by varying at will some subsets with zero measure. We will not provide such result explicitly.*

### B.4.2 Proofs of Proposition 3.2 and Appendix B.2

In the PAC framework of Definition 3.2, we have not specified formally the inner distance $d$:

$$d(r,\widehat{r}) := \frac{1}{M_{r,\widehat{r}}} \sup_{\pi \in \Pi} \sum_{h \in [\![H]\!]} \mathbb{E}_{(s,a) \sim d_h^{p,\pi}(\cdot,\cdot)} |r_h(s,a) - \widehat{r}_h(s,a)|, \tag{1}$$

where:

$$M_{r,\widehat{r}} := \max\{\sqrt{d}, \max_{h \in [\![H]\!]} \|\theta_h\|_2, \max_{h \in [\![H]\!]} \|\widehat{\theta}_h\|_2\}/\sqrt{d},$$

where $\{\theta_h\}_h$ and $\{\widehat{\theta}_h\}_h$ are the (unbounded) parameters of rewards $r$ and $\widehat{r}$. As explained in [31], such normalization term allows us to work with unbounded reward functions. In practice, we are relaxing the Linear MDP assumption presented in Section 2 about the boundedness of the parameters $\theta$ of the rewards to avoid the issue described in [38] and [31]. We still assume that the feature mapping is bounded. Observe that this relaxation does *not* affect the results we present, which would hold even if we considered bounded parameters $\theta$. Indeed, as visible in the proofs, the instances do not need to be constructed with unbounded $\theta$.

**Theorem 3.2** (Statistical Inefficiency). *Let $\mathcal{M} \cup \{\pi^E\}$ be a Linear IRL instance with finite state space $\mathcal{S}$ and deterministic expert's policy, and let $\epsilon, \delta \in (0,1)$. If an algorithm $\mathfrak{A}$ is $(\epsilon,\delta)$-PAC, then $\tau^E = \Omega(S)$, where $S := |\mathcal{S}|$ is the cardinality of the state space.*

*Proof.* We construct two problem instances that lie at a finite Hausdorff distance, and show that, with less than $S$ calls to the sampling oracle, we are not able to discriminate between the two instances.

Let $\mathcal{S}$ be the finite state space with cardinality $S$, $\mathcal{A} = \{a_1, a_2\}, H = 1, d_0(s) = 1/S \; \forall s \in \mathcal{S}$, $\phi(s,a) = \mathbb{1}\{a = a_1\}$, and consider two deterministic expert's policies $\pi_1^E(s) = a_1 \; \forall s \in \mathcal{S}$, and $\pi_2^E(s) = a_1 \; \forall s \in \mathcal{S} \backslash \{\overline{s}\}$, and $\pi_2^E(\overline{s}) = a_2$, for a certain $\overline{s} \in \mathcal{S}$. The set of parameters compatible with $\pi_1^E$ is:

$$\Theta_{p,\pi_1^E} = \{\theta \in \mathbb{R} \,|\, \theta \geqslant 0\},$$

since $Q^{\pi_1^E}(s,a_1) \geqslant Q^{\pi_1^E}(s,a_2) \iff r(s,a_1) \geqslant r(s,a_2) \iff \phi(s,a_1)\theta \geqslant \phi(s,a_2)\theta \iff 1 \cdot \theta \geqslant 0 \cdot \theta$. Observe that, for $\pi_2^E$, due to the presence of $\overline{s}$, we have:

$$\Theta_{p,\pi_2^E} = \{\theta \in \mathbb{R} \,|\, \theta = 0\},$$

since $\overline{s}$ imposes $\theta \leqslant 0$, and the other states impose $\theta \geqslant 0$.

Therefore, the Hausdorff distance between the two problems is:

$$\mathcal{H}(\mathcal{R}_{\pi_1^E}, \mathcal{R}_{\pi_2^E}) = \sup_{\theta \geqslant 0} \frac{1}{\max\{1,\theta,0\}}\theta = \sup_{\theta \geqslant 0} \frac{1}{\max\{1,\theta\}}\theta = 1.$$

Obviously, we need a $\Omega(S)$ samples to spot, if it exists, state $\overline{s}$, and thus distinguish between $\mathcal{R}_{\pi_1^E}$ and $\mathcal{R}_{\pi_2^E}$. $\qquad\square$

**Proposition B.1.** *Under the setting of Proposition 3.2, even under Assumption B.1, then an algorithm is $(\epsilon,\delta)$-PAC only if $\tau^E = \Omega(S)$.*

*Proof.* The same proof of Proposition 3.2 works here.

In particular, we now show that Assumption B.1 does not help. The Hausdorff distance between the instances in the proof of Proposition 3.2 can be written as:

$$\mathcal{H}(\mathcal{R}_{\pi_1^E}, \mathcal{R}_{\pi_2^E}) = \sup_{\theta_1 \geqslant 0} \inf_{\theta_2 = 0} \frac{1}{\max\{1, \theta_1, \theta_2\}} \sup_{\pi \in \Pi} \mathbb{E}_{s \sim d_0(\cdot), a \sim \pi(\cdot|s)} |r_1(s,a) - r_2(s,a)|$$

$$= \sup_{\theta_1 \geqslant 0} \inf_{\theta_2 = 0} \frac{1}{\max\{1, \theta_1\}} \sup_{\pi \in \Pi} \mathbb{E}_{s \sim d_0(\cdot), a \sim \pi(\cdot|s)} |\phi(s,a)\theta_1 - \phi(s,a)\theta_2$$
$$\pm \phi(s',a)\theta_1 \pm \phi(s',a)\theta_2|$$

$$\leqslant \sup_{\pi \in \Pi} \mathbb{E}_{s \sim d_0(\cdot), a \sim \pi(\cdot|s)} |\phi(s,a) - \phi(s',a)| + 0$$

$$+ \sup_{\pi \in \Pi} \sup_{\theta_1 \geqslant 0} \frac{1}{\max\{1, \theta_1\}} \inf_{\theta_2 = 0} \mathbb{E}_{s \sim d_0(\cdot), a \sim \pi(\cdot|s)} |\phi(s',a)\theta_1 - \phi(s',a)\theta_2|$$

$$= \sup_{\pi \in \Pi} \mathbb{E}_{s \sim d_0(\cdot), a \sim \pi(\cdot|s)} |\phi(s,a) - \phi(s',a)| + \sup_{\pi \in \Pi} \mathbb{E}_{s \sim d_0(\cdot), a \sim \pi(\cdot|s)} \phi(s',a),$$

where $s'$ is the state in the covering closest to state $s$; while the first term can be bounded, the assumption does not help us with the second term. $\square$

**Proposition B.2.** *Under the setting of Proposition 3.2, under Assumption B.2, then a number of samples $\tau^E = |\mathcal{N}(\frac{\Delta}{2L}; \mathcal{S}, \|\cdot\|)|$ is sufficient to recover $\pi^E$ exactly in any $(s,h) \in \mathcal{S}$, where $|\mathcal{N}(\frac{\Delta}{2L}; \mathcal{S}, \|\cdot\|)|$ is the $\Delta/(2L)$-covering number of space $\mathcal{S}$ w.r.t. distance $\|\cdot\|$.*

*Proof.* For any state $s \in \mathcal{S}^{p, \pi^E}$, by definition of covering $\mathcal{N}(\frac{\Delta}{2L}; \mathcal{S}, \|\cdot\|)$, there always exist another state $s' \in \mathcal{N}(\frac{\Delta}{2L}; \mathcal{S}, \|\cdot\|)$ such that $\|s' - s\| \leqslant \frac{\Delta}{2L}$. By Assumption B.2 we know that:

$$\|\phi(s, \pi_h^E(s)) - \phi(s', \pi_h^E(s'))\|_2 \leqslant L\|s' - s\| \leqslant \frac{\Delta}{2},$$

and since $\pi_h^E(s')$ and thus $\phi(s', \pi_h^E(s'))$ is known, then the fact that $\Delta$ is finite guarantees us that $\pi_h^E(s)$ is equal to the action $a$ that minimizes the distance to $\phi(s', \pi_h^E(s'))$. Notice that if, by contradiction, there were two actions $a_1, a_2$ with $\|\phi(s, a_1) - \phi(s', \pi_h^E(s'))\|_2 \leqslant \frac{\Delta}{2}$ and $\|\phi(s, a_2) - \phi(s', \pi_h^E(s'))\|_2 \leqslant \frac{\Delta}{2}$, then by triangle inequality and finiteness of $\Delta$, we would have:

$$\Delta < \|\phi(s, a_1) - \phi(s, a_2)\|_2$$
$$\leqslant \|\phi(s, a_1) - \phi(s', \pi_h^E(s'))\|_2 + \|\phi(s, a_2) - \phi(s', \pi_h^E(s'))\|_2$$
$$\leqslant \frac{\Delta}{2} + \frac{\Delta}{2} = \Delta,$$

which is clearly a contradiction. $\square$

### B.4.3 Proof of Theorem 3.3

The proof is based on deriving an upper bound to the Hausdorff distance between the true feasible set and its estimate. To do so, first, using the notation of [23], let us define the following quantities:

$$\mathbb{P}_h(\cdot|s,a) := \langle \phi(s,a), \mu_h(\cdot) \rangle,$$

$$\widehat{\mathbb{P}}_h(\cdot|s,a) := \phi(s,a)^\intercal \Lambda_h^{-1} \sum_{k=1}^{\tau} \phi(s_h^k, a_h^k) \delta(\cdot, s_{h+1}^k),$$

$$\overline{\mathbb{P}}_h(\cdot|s,a) := \phi(s,a)^\intercal \Lambda_h^{-1} \sum_{k=1}^{\tau} \phi(s_h^k, a_h^k) \mathbb{P}_h(\cdot|s_h^k, a_h^k),$$

where $\delta(\cdot, x)$ is the Dirac measure, and $(s_h^k, a_h^k)$ represents the state-action pair visited at stage $h$ of exploration episode $k \in [\![\tau]\!]$. In words, $\mathbb{P}$ denotes the true transition model, $\widehat{\mathbb{P}}$ denotes the least squares estimate computed by Algorithm 1, and $\overline{\mathbb{P}}$ represents a bridge between the two. As we will see, the core of the proof consists in upper bounding the term $|(\mathbb{P}_h - \widehat{\mathbb{P}})V_{h+1}(s,a)|$ at all $h \in [\![H]\!]$

and reachable $(s, a) \in \mathcal{S} \times \mathcal{A}$, for all the bounded linear functions $V$ in class $\mathcal{V}$, defined as:

$$\mathcal{V} := \left\{ V : \mathcal{S} \times \llbracket H \rrbracket \to [-H, +H] \,\middle|\, V(\cdot) = \max_{a \in \mathcal{A}} \phi(\cdot, a)^\mathsf{T} w, \, \|w\|_2 \leqslant 2H\sqrt{d} \right\}. \tag{2}$$

To achieve this goal, it will be useful to apply triangle inequality and to bound the following two terms separately:

$$\left| \left( \mathbb{P}_h - \widehat{\mathbb{P}} \right) V_{h+1}(s, a) \right| \leqslant \left| \left( \mathbb{P}_h - \overline{\mathbb{P}} \right) V_{h+1}(s, a) \right| + \left| \left( \overline{\mathbb{P}}_h - \widehat{\mathbb{P}} \right) V_{h+1}(s, a) \right|.$$

Lemma B.7 and Lemma B.8, which we now present, serve exactly this purpose.

**Lemma B.7.** *For any value function $V$ in the class $\mathcal{V}$, for any $(s, a, h) \in \mathcal{S} \times \mathcal{A} \times \llbracket H \rrbracket$, it holds that:*

$$\left| \left( \overline{\mathbb{P}}_h - \mathbb{P}_h \right) V_{h+1}(s, a) \right| \leqslant \min \left\{ H\sqrt{d} \|\phi(s, a)\|_{\Lambda_h^{-1}}, 2H \right\}.$$

*Proof.* We have:

$$\left( \overline{\mathbb{P}}_h - \mathbb{P}_h \right) V_{h+1}(s, a) = \phi(s, a)^\mathsf{T} \Lambda_h^{-1} \sum_{k=1}^{\tau} \phi(s_h^k, a_h^k) \mathbb{P}_h V_{h+1}(s_h^k, a_h^k) - \mathbb{P}_h V_{h+1}(s, a)$$

$$\overset{(1)}{=} \phi(s, a)^\mathsf{T} \Lambda_h^{-1} \sum_{k=1}^{\tau} \phi(s_h^k, a_h^k) \mathbb{P}_h V_{h+1}(s_h^k, a_h^k) - \phi(s, a)^\mathsf{T} \widetilde{w}_h$$

$$= \phi(s, a)^\mathsf{T} \Lambda_h^{-1} \sum_{k=1}^{\tau} \phi(s_h^k, a_h^k) \mathbb{P}_h V_{h+1}(s_h^k, a_h^k) - \phi(s, a)^\mathsf{T} \Lambda_h^{-1} \Lambda_h \widetilde{w}_h$$

$$= \phi(s, a)^\mathsf{T} \Lambda_h^{-1} \left[ \sum_{k=1}^{\tau} \phi(s_h^k, a_h^k) \mathbb{P}_h V_{h+1}(s_h^k, a_h^k) - \Lambda_h \widetilde{w}_h \right]$$

$$\overset{(2)}{=} \phi(s, a)^\mathsf{T} \Lambda_h^{-1} \left[ \sum_{k=1}^{\tau} \phi(s_h^k, a_h^k) \mathbb{P}_h V_{h+1}(s_h^k, a_h^k) \right.$$

$$\left. - I\widetilde{w}_h - \sum_{k=1}^{\tau} \phi(s_h^k, a_h^k) \phi(s_h^k, a_h^k)^\mathsf{T} \widetilde{w}_h \right]$$

$$\overset{(3)}{=} \phi(s, a)^\mathsf{T} \Lambda_h^{-1} \left[ \sum_{k=1}^{\tau} \phi(s_h^k, a_h^k) \mathbb{P}_h V_{h+1}(s_h^k, a_h^k) \right.$$

$$\left. - \sum_{k=1}^{\tau} \phi(s_h^k, a_h^k) \mathbb{P}_h V_{h+1}(s_h^k, a_h^k) - \widetilde{w}_h \right]$$

$$= -\phi(s, a)^\mathsf{T} \Lambda_h^{-1} \widetilde{w}_h,$$

where at (1) we have defined vector $\widetilde{w}_h := \int_{\mathcal{S}} V_{h+1}(s') d\mu_h(s')$, at (2) we have used the definition of $\Lambda_h$, and at (3) we have recognized that $\phi(s_h^k, a_h^k)^\mathsf{T} \widetilde{w}_h = \mathbb{P}_h V_{h+1}(s_h^k, a_h^k)$.

By taking the absolute value, we can write:

$$\left| \left( \overline{\mathbb{P}}_h - \mathbb{P}_h \right) V_{h+1}(s, a) \right| = \left| \phi(s, a)^\mathsf{T} \Lambda_h^{-1} \widetilde{w}_h \right|$$

$$\overset{(4)}{\leqslant} \|\widetilde{w}_h\|_{\Lambda_h^{-1}} \|\phi(s, a)\|_{\Lambda_h^{-1}}$$

$$\overset{(5)}{\leqslant} \|\widetilde{w}_h\|_2 \|\phi(s, a)\|_{\Lambda_h^{-1}}$$

$$\overset{(6)}{\leqslant} H\sqrt{d} \|\phi(s, a)\|_{\Lambda_h^{-1}},$$

where at (4) we have applied Cauchy-Schwarz's inequality, at (5) we have bounded the quadratic form with the 2-norm and the largest eigenvector of the matrix, i.e., $\|\widetilde{w}_h\|_{\Lambda_h^{-1}} = \sqrt{\widetilde{w}_h^\mathsf{T} \Lambda_h^{-1} \widetilde{w}_h} \leqslant \sqrt{\sigma} \|\widetilde{w}_h\|_2$, where $\sigma$ is the largest eigenvalue of matrix $\Lambda_h^{-1}$, and then we have upper bounded $\sigma \leqslant 1$, since 1 is the smallest eigenvalue of invertible matrix $\Lambda_h$ (see [23]); finally, at (6) we have used the fact that $|V_{h+1}(\cdot)| \leqslant H$, and so that $\|\widetilde{w}_h\|_2 = \|\int_{\mathcal{S}} V_{h+1}(s') d\mu_h(s')\|_2 \leqslant H\|\mu_h(\mathcal{S})\|_2 \leqslant H\sqrt{d}$.

The result follows by noticing that the quantity to bound cannot be larger than $2H$. $\qquad\square$

**Lemma B.8.** *Let $\delta \in (0,1)$. For any value function $V$ in the class $\mathcal{V}$, for any $(s,a,h) \in \mathcal{S} \times \mathcal{A} \times \llbracket H \rrbracket$, with probability at least $1 - \delta/2$, it holds that:*

$$\left| \left( \widehat{\mathbb{P}}_h - \overline{\mathbb{P}}_h \right) V_{h+1}(s,a) \right| \leqslant \min \left\{ cH \sqrt{d \log \left( 1 + \tau \right) + \log \frac{H}{\delta}} \, \|\phi(s,a)\|_{\Lambda_h^{-1}}, 2H \right\},$$

*for some constant $c$.*

*Proof.* We can write:

$$\left| \left( \widehat{\mathbb{P}}_h - \overline{\mathbb{P}}_h \right) V_{h+1}(s,a) \right| = \left| \phi(s,a)^{\mathsf{T}} \Lambda_h^{-1} \sum_{k=1}^{\tau} \phi(s_h^k, a_h^k) \left[ V_{h+1}(s_{h+1}^k) - \mathbb{P}_h V_{h+1}(s_h^k, a_h^k) \right] \right|$$

$$\overset{(1)}{\leqslant} \left\| \sum_{k=1}^{\tau} \phi(s_h^k, a_h^k) \left[ V_{h+1}(s_{h+1}^k) - \mathbb{P}_h V_{h+1}(s_h^k, a_h^k) \right] \right\|_{\Lambda_h^{-1}} \left\| \phi(s,a) \right\|_{\Lambda_h^{-1}}$$

$$\overset{(2)}{\leqslant} \sqrt{4H^2 \left( \frac{d}{2} \log(1+\tau) + \log \frac{2\mathcal{N}_\epsilon}{\delta} \right) + 8\tau^2 \epsilon^2} \left\| \phi(s,a) \right\|_{\Lambda_h^{-1}}$$

$$\overset{(3)}{\leqslant} \sqrt{4H^2 \left( \frac{d}{2} \log(1+\tau) + 2d \log \left( 1 + \frac{H\sqrt{d}}{\epsilon} \right) + \log \frac{1}{\delta} \right) + 8\tau^2 \epsilon^2} \left\| \phi(s,a) \right\|_{\Lambda_h^{-1}}$$

$$\overset{(4)}{=} \sqrt{4H^2 \left( \frac{d}{2} \log(1+\tau) + 2d \log \left( 1 + 4\tau \right) + \log \frac{1}{\delta} \right) + 8H^2 d} \left\| \phi(s,a) \right\|_{\Lambda_h^{-1}}$$

$$\overset{(5)}{\leqslant} cH \sqrt{d \log(1+\tau) + \log \frac{1}{\delta}} \left\| \phi(s,a) \right\|_{\Lambda_h^{-1}},$$

where at (1) we have applied Cauchy-Schwarz's inequality, at (2) we have applied Lemma B.13, at (3) we have upper bounded $\mathcal{N}_\epsilon$ using Lemma B.12, at (4), similarly to [62], unlike [23], we see that no union bound is needed (because there is no dependence on $\Lambda$), thus by choosing $\epsilon = H\sqrt{d}/\tau$, we get the passage. Passage (5) follows for some constant $c$.

The result follows by a union bound over $h \in \llbracket H \rrbracket$, and by noticing that the quantity to bound cannot be larger than $2H$. $\qquad\square$

We are now ready to upper bound the Hausdorff distance using the two lemmas just presented. Recall that we work with unbounded rewards (parameters $\theta$), and that the definition of inner distance $d$ is provided in Equation (1).

**Lemma B.9.** *With probability at least $1 - \delta/2$, the Hausdorff distance between the true feasible set $\mathcal{R}_{p,\pi^E}$ and its estimate $\widehat{\mathcal{R}}$ returned by Algorithm 1 can be upper bounded by:*

$$\mathcal{H}(\mathcal{R}_{p,\pi^E}, \widehat{\mathcal{R}}) \leqslant 4J^*(u;p),$$

*where $u_h(s,a) := \min\{\beta \|\phi(s,a)\|_{\Lambda_h^{-1}}, H\}$ for all $(s,a,h) \in \mathcal{S} \times \mathcal{A} \times \llbracket H \rrbracket$, and $\beta := cH\sqrt{d\log(1+\tau) + \log(H/\delta)}$ for some absolute constant $c > 0$.*

*Proof.* Let us begin to bound the first branch of the Hausdorff distance.

$$\sup_{r \in \mathcal{R}_{p,\pi^E}} \inf_{\widehat{r} \in \widehat{\mathcal{R}}} d(r, \widehat{r}) = \sup_{r \in \mathcal{R}_{p,\pi^E}} \inf_{\widehat{r} \in \widehat{\mathcal{R}}} \frac{1}{M_{r,\widehat{r}}} \sup_{\pi \in \Pi} \sum_{h \in \llbracket H \rrbracket} \mathbb{E}_{(s,a) \sim d_h^{p,\pi}(\cdot,\cdot)} |r_h(s,a) - \widehat{r}_h(s,a)|$$

$$\overset{(1)}{=} \sup_{r \in \mathcal{R}_{p,\pi^E}} \inf_{\widehat{r} \in \widehat{\mathcal{R}}} \frac{1}{M_{r,\widehat{r}}} \sup_{\pi \in \Pi} \sum_{h \in \llbracket H \rrbracket} \mathbb{E}_{(s,a) \sim d_h^{p,\pi}(\cdot,\cdot)} \left| Q_h^*(s,a;p,r) - \mathbb{P}_h V_{h+1}^*(s,a;p,r) \right.$$

$$\left. -Q_h^*(s,a;\widehat{p},\widehat{r}) + \widehat{\mathbb{P}}_h V_{h+1}^*(s,a;\widehat{p},\widehat{r}) \right|$$

$$\overset{(2)}{\leqslant} \sup_{r \in \mathcal{R}_{p,\pi^E}} \frac{1}{M_{r,\widetilde{r}}} \sup_{\pi \in \Pi} \sum_{h \in \llbracket H \rrbracket} \mathbb{E}_{(s,a) \sim d_h^{p,\pi}(\cdot,\cdot)} \left| Q_h^*(s,a;p,r) - \mathbb{P}_h V_{h+1}^*(s,a;p,r) \right.$$

$$\left. - Q_h^*(s,a;\widehat{p},\widetilde{r}) + \widehat{\mathbb{P}}_h V_{h+1}^*(s,a;\widehat{p},\widetilde{r}) \right|,$$

$$\overset{(3)}{=} \sup_{r\in\mathcal{R}_{p,\pi^E}} \frac{1}{M_{r,\tilde{r}}} \sup_{\pi\in\Pi} \sum_{h\in[\![H]\!]} \mathbb{E}_{(s,a)\sim d_h^{p,\pi}(\cdot,\cdot)} \Big| Q_h^*(s,a;p,r) - \mathbb{P}_h V_{h+1}^*(s,a;p,r)$$

$$- Q_h^*(s,a;p,r) + \widehat{\mathbb{P}}_h V_{h+1}^*(s,a;p,r) \Big|$$

$$= \sup_{r\in\mathcal{R}_{p,\pi^E}} \frac{1}{M_{r,\tilde{r}}} \sup_{\pi\in\Pi} \sum_{h\in[\![H]\!]} \mathbb{E}_{(s,a)\sim d_h^{p,\pi}(\cdot,\cdot)} \Big| \big(\widehat{\mathbb{P}}_h - \mathbb{P}_h\big) V_{h+1}^*(s,a;p,r) \Big|$$

$$\overset{(4)}{\leqslant} \sup_{r\in\mathcal{R}_{p,\pi^E}} \sum_{h\in[\![H]\!]} \mathbb{E}_{(s,a)\sim d_h^{p,\pi}(\cdot,\cdot)} \Big| \big(\widehat{\mathbb{P}}_h - \mathbb{P}_h\big) \frac{V_{h+1}^*(s,a;p,r)}{\max\{1,\max_h \|\theta_h\|_2/\sqrt{d}\}} \Big|$$

$$\overset{(5)}{=} \sup_{r\in\mathcal{R}_{p,\pi^E}} \sum_{h\in[\![H]\!]} \mathbb{E}_{(s,a)\sim d_h^{p,\pi}(\cdot,\cdot)} \Big| \big(\widehat{\mathbb{P}}_h - \mathbb{P}_h\big) V_{h+1}^*(s,a;p,\tfrac{r}{K}) \Big|$$

$$\overset{(6)}{\leqslant} \sup_{V\in\mathcal{V}} \sup_{\pi\in\Pi} \sum_{h\in[\![H]\!]} \mathbb{E}_{(s,a)\sim d_h^{p,\pi}(\cdot,\cdot)} \Big| \big(\widehat{\mathbb{P}}_h - \mathbb{P}_h\big) V_{h+1}(s,a) \Big|,$$

where at (1) we have simply applied the Bellman optimality equation twice w.r.t. the reward function, at (2) we have upper bounded the infimum over the second set of rewards $\widehat{\mathcal{R}}$ with the specific choice of reward $\tilde{r}\in\widehat{\mathcal{R}}$ provided by Lemma B.11, at (3) we use the property of $\tilde{r}$ described in Lemma B.11, at (4) we bring term $1/M_{r,\tilde{r}}$ inside, and then we upper bound it by: $1/M_{r,\tilde{r}} := 1/\max\{\sqrt{d}, \max_h \|\theta_h\|_2, \max_h \|\tilde{\theta}_h\|_2\}/\sqrt{d} \leqslant 1/\max\{1, \max_h \|\theta_h\|_2/\sqrt{d}\}$, i.e., by simply removing one of the terms inside the maximum operator at denominator. At (5) we define $K := \max\{1, \max_h \|\theta_h\|_2/\sqrt{d}\}$, and, since the value function is linear in the reward, we apply $K$ directly to the reward. At (6) we realize that the possible optimal value functions that can be constructed in $p$ using rewards in $\mathcal{R}_{p,\pi^E}$ normalized by $K$ are a subset of the value functions in class $\mathcal{V}$, i.e., of all the possible optimal value functions with parameters $\|w_h\|_2 \leqslant 2H\sqrt{d}$. This is not trivial since we are working with *unbounded* rewards $r$, and thus their parameters $\{\theta_h\}_h$ can be any. The normalization by $K$ permits this in the following manner. For any $h\in[\![H]\!]$, we have $r_h(\cdot,\cdot)/K = \langle\phi(\cdot,\cdot), \theta_h/K\rangle = \langle\phi(\cdot,\cdot), \theta_h/\max\{1, \max_{h'} \|\theta_{h'}\|_2/\sqrt{d}\}\rangle$. Therefore, if $\max_{h'} \|\theta_{h'}\|_2 > \sqrt{d}$, then the normalization makes sure that $\max_{h'} \|\theta_{h'}\|_2 = \sqrt{d}$, while if $\max_{h'} \|\theta_{h'}\|_2 \leqslant \sqrt{d}$, then the normalization is by 1 and it has no effect. In this way, we see that value functions $V_{h+1}^*(s,a;p,\tfrac{r}{K})$ can be created by a simple $r'$ with parameters $\{\theta_h'\}_h$ with 2-norms bounded by $\sqrt{d}$. This guarantees that, since by hypothesis of Linear MDPs $\|\phi(\cdot,\cdot)\|_2 \leqslant 1$, the value function never exceeds $H$, and that the norm of the $Q$-function parameters $\{w_h^\pi\}_h$ for any policy $\pi$ can be bounded as: $\|w_h^\pi\|_2 \leqslant \|\theta_h/K\|_2 + \|\int_{\mathcal{S}} V_{h+1}^\pi(s')d\mu_h(s')\|_2 \leqslant \sqrt{d} + H\|\mu_h(\mathcal{S})\|_2 \leqslant \sqrt{d} + H\sqrt{d} \leqslant 2H\sqrt{d}$ (similarly to Lemma B.1 of [23]). It should be remarked that class $\mathcal{V}$ is more general than the actual set of optimal value functions that can be obtained using $r\in\mathcal{R}_{p,\pi^E}$ in $p$, since such rewards induce optimal value functions for which the optimal action in $\mathcal{S}^{p,\pi^E}$ is always the expert's action/s $\pi^E(s)$.

Notice that the same derivation can be carried out also for the other branch of the Hausdorff distance, ending up with the same expression. Therefore, the last line is an upper bound to the Hausdorff distance:

$$\mathcal{H}_d(\mathcal{R}_{p,\pi^E}, \widehat{\mathcal{R}}) \leqslant \sup_{V\in\mathcal{V}} \sup_{\pi\in\Pi} \sum_{h\in[\![H]\!]} \mathbb{E}_{(s,a)\sim d_h^{p,\pi}(\cdot,\cdot)} \Big| \big(\widehat{\mathbb{P}}_h - \mathbb{P}_h\big) V_{h+1}(s,a) \Big|$$

$$\overset{(7)}{=} \sup_{\pi\in\Pi} \sum_{h\in[\![H]\!]} \mathbb{E}_{(s,a)\sim d_h^{p,\pi}(\cdot,\cdot)} \sup_{V\in\mathcal{V}} \Big| \big(\widehat{\mathbb{P}}_h - \mathbb{P}_h\big) V_{h+1}(s,a) \Big|$$

$$= \sup_{\pi\in\Pi} \sum_{h\in[\![H]\!]} \mathbb{E}_{(s,a)\sim d_h^{p,\pi}(\cdot,\cdot)} \sup_{V\in\mathcal{V}} \Big| \big(\widehat{\mathbb{P}}_h - \mathbb{P}_h\big) V_{h+1}(s,a) \pm \overline{\mathbb{P}}_h V_{h+1}(s,a) \Big|$$

$$\overset{(8)}{\leqslant} \sup_{\pi\in\Pi} \sum_{h\in[\![H]\!]} \mathbb{E}_{(s,a)\sim d_h^{p,\pi}(\cdot,\cdot)} \sup_{V\in\mathcal{V}} \Big| \big(\overline{\mathbb{P}}_h - \mathbb{P}_h\big) V_{h+1}(s,a) \Big| + \Big| \big(\widehat{\mathbb{P}}_h - \overline{\mathbb{P}}_h\big) V_{h+1}(s,a) \Big|$$

$$\overset{(9)}{\leqslant} \sup_{\pi\in\Pi} \sum_{h\in[\![H]\!]} \mathbb{E}_{(s,a)\sim d_h^{p,\pi}(\cdot,\cdot)} \min\left\{ c_1 H \sqrt{d\log(1+\tau) + \log\frac{H}{\delta}} \|\phi(s,a)\|_{\Lambda_h^{-1}}, 4H \right\}$$

$$\leqslant 4 \sup_{\pi \in \Pi} \sum_{h \in \llbracket H \rrbracket} \mathop{\mathbb{E}}_{(s,a) \sim d_h^{p,\pi}(\cdot,\cdot)} \min \Big\{ \underbrace{c_2 H \sqrt{d \log(1+\tau) + \log \frac{H}{\delta}}}_{=:\beta} \|\phi(s,a)\|_{\Lambda_h^{-1}}, H \Big\}$$

$$= 4 \sup_{\pi \in \Pi} \sum_{h \in \llbracket H \rrbracket} \mathop{\mathbb{E}}_{(s,a) \sim d_h^{p,\pi}(\cdot,\cdot)} \underbrace{\min \big\{ \beta \|\phi(s,a)\|_{\Lambda_h^{-1}}, H \big\}}_{=:u_h(s,a)}$$

$$= 4 \sup_{\pi \in \Pi} \sum_{h \in \llbracket H \rrbracket} \mathop{\mathbb{E}}_{(s,a) \sim d_h^{p,\pi}(\cdot,\cdot)} u_h(s,a)$$

$$= 4 J^*(u; p),$$

where at (7) we have noticed that class $\mathcal{V}$ contains the cartesian product of $H$ sets, one for each stage, and therefore the supremum can be brought inside the summation, at (8) we have applied triangle inequality, at (9) we have applied Lemma B.7 and Lemma B.8 and used some absolute constants $c_1, c_2 > 0$, and also the fact that for any numbers $x, y, w, z$, we have $\min\{x,y\} + \min\{w,z\} \leqslant \min\{x+w, y+z\}$.

$\square$

To conclude the proof of the main theorem, we simply have to observe that any RFE algorithm provides a bound to $J^*(u'; p)$ for some $u'$ similar to $u$. Depending on the RFE algorithm instantiated as sub-routine, the sample complexity of Algorithm 1 varies.

**Theorem 3.3.** *Assume that $\pi^E$ (along with its support $\mathcal{S}^{p,\pi^E}$) is known. Then, for any $\epsilon, \delta \in (0,1)$, Algorithm 1 is $(\epsilon, \delta)$-PAC for IRL with a number of episodes $\tau$ upper bounded by:*

$$\tau \leqslant \widetilde{\mathcal{O}}\Big( \frac{H^5 d}{\epsilon^2} \Big( d + \log \frac{1}{\delta} \Big) \Big).$$

*Proof.* To get the result, we instantiate Algorithm 1 of [62] as RFE sub-routine. Simply, observe that [62] sets $\beta'$ so that $\beta' \geqslant \widetilde{\beta} := c' H \sqrt{d \log(1 + dH\tau) + \log(H/\delta)} \geqslant \beta$. By Lemma B.9, we know that:

$$\mathcal{H}(\mathcal{R}_{p,\pi^E}, \widehat{\mathcal{R}}) \leqslant 4 \sup_{\pi \in \Pi} \sum_{h \in \llbracket H \rrbracket} \mathop{\mathbb{E}}_{(s,a) \sim d_h^{p,\pi}(\cdot,\cdot)} \min \big\{ \beta \|\phi(s,a)\|_{\Lambda_h^{-1}}, H \big\}$$

$$\leqslant 2 c_1 \beta' \sum_{h \in \llbracket H \rrbracket} \sup_{\pi \in \Pi} \mathop{\mathbb{E}}_{(s,a) \sim d_h^{p,\pi}(\cdot,\cdot)} \|\phi(s,a)\|_{\Lambda_h^{-1}},$$

for some absolute constant $c_1 > 0$. It should be remarked that the quantity in the last line is, modulo $c_1$, the quantity that [62] bound in the proof of their Theorem 1 using their algorithm. Specifically, by taking:

$$\tau \leqslant \widetilde{\mathcal{O}}\Big( \frac{H^5 d}{\epsilon^2} \Big( d + \log \frac{1}{\delta} \Big) + \frac{H^6 d^{9/2}}{\epsilon} \log^4 \frac{1}{\delta} \Big),$$

and a union bound over the two events that hold w.p. $1 - \delta/2$, and re-setting $\epsilon \leftarrow c_1 \epsilon$, we get the result. $\square$

Notice that if we run Algorithm 1 of [63] for exploration instead of Algorithm 1 of [62], we obtain:

**Theorem B.10.** *If we use Algorithm 1 of [63] at Line 1 of Algorithm 1, then for any $\epsilon, \delta \in (0,1)$, such algorithm is $(\epsilon, \delta)$-PAC for IRL with a number of episodes $\tau$ upper bounded by:*

$$\tau \leqslant \widetilde{\mathcal{O}}\Big( \frac{H^6 d^3}{\epsilon^2} \log \frac{1}{\delta} \Big).$$

*Proof.* By Lemma B.9, we know that:

$$\mathcal{H}(\mathcal{R}_{p,\pi^E}, \widehat{\mathcal{R}}) \leqslant 4 J^*(u; p)$$

$$= 4 \sup_{\pi \in \Pi} \sum_{h \in \llbracket H \rrbracket} \mathop{\mathbb{E}}_{(s,a) \sim d_h^{p,\pi}(\cdot,\cdot)} \min \big\{ \beta \|\phi(s,a)\|_{\Lambda_h^{-1}}, H \big\},$$

for $\beta := cH\sqrt{d\log(1+\tau) + \log(H/\delta)}$. Now, let us define, similarly to Appendix A of [63], the quantities $u'_h(s,a) := \min\{\beta'\|\phi(s,a)\|_{\Lambda_h^{-1}}, H\}$ for all $(s,a,h) \in \mathcal{S} \times \mathcal{A} \times [\![H]\!]$, and $\beta' := c'dH\sqrt{\log(dH/\delta/\epsilon)}$ for some absolute constant $c' > 0$. In addition, set the number of exploration episodes $\tau$ to $\tau = c''d^3H^6\log(dH\delta^{-1}\epsilon^{-1})/\epsilon^2$, and notice that, for appropriate choices of $c', c''$, it holds that: $\beta' \geqslant c'dH\sqrt{\log(dH\tau/\delta)} \geqslant \beta := cH\sqrt{d\log(1+\tau) + \log(1/\delta)}$. This entails that $u'_h(s,a) \geqslant u_h(s,a)$ at all $s,a,h$, and so:

$$\mathcal{H}(\mathcal{R}_{p,\pi^E}, \widehat{\mathcal{R}}) \leqslant cJ^*(u';p)$$
$$= cHJ^*(u'/H;p)$$
$$\overset{(1)}{\leqslant} c_1 H\sqrt{\frac{d^3H^4\log\frac{d\tau H}{\delta}}{\tau}}$$
$$\overset{(2)}{\leqslant} c_2\epsilon,$$

where at (1) we have applied Lemma 3.2 of [63] (reported in Lemma B.14 for simplicity) with some new constant $c_1 > 0$, and at (2) we have simply replaced $\tau$ with its value defined in Algorithm 1 of [63].

The result follows by union bound between the two events that hold w.p. $1 - \delta/2$ to get $1 - \delta$, and by noticing that $c_2$ is a constant, thus setting $\epsilon \leftarrow c_2\epsilon$ provides the result. $\qquad\square$

**Lemma B.11.** *Let $\mathcal{R}_{p,\pi^E}$ be the feasible set of policy $\pi^E$ w.r.t. transition models $p$, and let $\widehat{\mathcal{R}}$ be its estimate constructed as in Algorithm 1 using the true $\pi^E, \mathcal{S}^{p,\pi^E}$ (or sets $\mathcal{Z}$) and some $\widehat{p}$. For any reward $r \in \mathcal{R}_{p,\pi^E}$, the reward $\widehat{r}$ such that, for all $(s,a,h) \in \mathcal{S} \times \mathcal{A} \times [\![H]\!]$:*

$$\widehat{r}_h(s,a) = r_h(s,a) + \int_{s'\in\mathcal{S}} p_h(s'|s,a)V^*_{h+1}(s';p,r) - \int_{s'\in\mathcal{S}} \widehat{p}_h(s'|s,a)V^*_{h+1}(s';\widehat{p},\widehat{r}),$$

*belongs to $\widehat{\mathcal{R}}$. Moreover, observe that: $Q^*_h(s,a;p,r) = Q^*_h(s,a;\widehat{p},\widehat{r})$ at all $(s,a,h) \in \mathcal{S} \times \mathcal{A} \times [\![H]\!]$. In addition, for any reward $\widehat{r} \in \widehat{\mathcal{R}}$, it is possible to construct a reward $r$ in analogous manner so that $r \in \mathcal{R}_{p,\pi^E}$, and such that $Q^*_h(s,a;p,r) = Q^*_h(s,a;\widehat{p},\widehat{r})$ at all $(s,a,h) \in \mathcal{S} \times \mathcal{A} \times [\![H]\!]$.*

*Proof.* First, we consider the case when $\mathcal{S}$ is finite. By rearranging the terms in the definition of $\widehat{r}$, we see that, for all $(s,a,h) \in \mathcal{S} \times \mathcal{A} \times [\![H]\!]$:

$$\widehat{r}_h(s,a) + \sum_{s'\in\mathcal{S}} \widehat{p}_h(s'|s,a)V^*_{h+1}(s';\widehat{p},\widehat{r}) = r_h(s,a) + \sum_{s'\in\mathcal{S}} p_h(s'|s,a)V^*_{h+1}(s';p,r),$$

which, by the Bellman optimality equation, entails that $Q^*_h(s,a;p,r) = Q^*_h(s,a;\widehat{p},\widehat{r})$.

We recall that $\widehat{\mathcal{R}}$ is defined as:

$$\widehat{\mathcal{R}} = \Big\{\widehat{r} \in \mathfrak{R} \,\Big|\, \forall(s,h) \in \mathcal{S}^{p,\pi^E}, \forall a \in \mathcal{A} : \underset{a'\sim\pi^E_h(\cdot|s)}{\mathbb{E}} Q^*_h(s,a';\widehat{p},\widehat{r}) \geqslant Q^*_h(s,a;\widehat{p},\widehat{r})\Big\},$$

while thanks to Lemma B.4, the feasible set $\mathcal{R}_{p,\pi^E}$ can be written as:

$$\mathcal{R}_{p,\pi^E} = \Big\{r \in \mathfrak{R} \,\Big|\, \forall(s,h) \in \mathcal{S}^{p,\pi^E}, \forall a \in \mathcal{A} : \underset{a'\sim\pi^E_h(\cdot|s)}{\mathbb{E}} Q^*_h(s,a';p,r) \geqslant Q^*_h(s,a;p,r)\Big\}.$$

It is clear that, if $Q^*_h(s,a;p,r) = Q^*_h(s,a;\widehat{p},\widehat{r})$ for all $(s,a,h) \in \mathcal{S} \times \mathcal{A} \times [\![H]\!]$, then since $r \in \mathcal{R}_{p,\pi^E}$ we necessarily have $\widehat{r} \in \widehat{\mathcal{R}}$.

The proof of the opposite case is completely analogous.

In the case with infinite $\mathcal{S}$, notice that both the feasible set $\mathcal{R}_{p,\pi^E}$ in Lemma B.4 and the definition of $\widehat{\mathcal{R}}$ in Algorithm 1 make use of the same sets $\mathcal{Z}$. Thus, we simply make the choice of reward with same $\mathcal{Z}$ and proceed like in the finite case. $\qquad\square$

**Lemma B.12** (Covering Number of Class $\mathcal{V}$)**.** *Let $\mathcal{V}$ be defined as in Equation (2), and define distance dist in $\mathcal{V}$ as $dist(V,V') := \sup_{s\in\mathcal{S}}|V(s) - V'(s)|$. Then, the $\epsilon$-covering number $|\mathcal{N}(\epsilon;\mathcal{V},dist)|$ of*

*set $\mathcal{V}$ with distance dist can be bounded as:*

$$\log |\mathcal{N}(\epsilon; \mathcal{V}, dist)| \leqslant d \log \left( 1 + \frac{4H\sqrt{d}}{\epsilon} \right).$$

*Proof.* The proof follows that of Lemma D.6 of [23], but is simpler because of the different form of $\mathcal{V}$.

For any $V_1, V_2 \in \mathcal{V}$ parametrized by $w_1, w_2$, we write:

$$
\begin{aligned}
\text{dist}(V_1, V_2) &= \sup_{s \in \mathcal{S}} \left| \max_{a \in \mathcal{A}} \langle \phi(s,a), w_1 \rangle - \max_{a \in \mathcal{A}} \langle \phi(s,a), w_2 \rangle \right| \\
&\overset{(1)}{\leqslant} \max_{(s,a) \in \mathcal{S} \times \mathcal{A}} \left| \phi(s,a)^{\mathsf{T}}(w_1 - w_2) \right| \\
&\overset{(2)}{\leqslant} \sup_{\phi : \|\phi\|_2 \leqslant 1} \left| \phi^{\mathsf{T}}(w_1 - w_2) \right| \\
&\overset{(3)}{=} \|w_1 - w_2\|_2,
\end{aligned}
$$

where at (1) we have used the common bound that the absolute difference of maxima is upper bounded by the maximum of the absolute difference of the two functions, at (2) we have used the fact that the feature map is always bounded by 1 in 2-norm, and at (3) we have recognized the dual norm of the 2-norm, i.e., itself.

If we construct an $\epsilon$-cover of $\mathcal{W} := \{ w \in \mathbb{R}^d \mid \|w\|_2 \leqslant 2H\sqrt{d} \}$ w.r.t. the 2-norm, we get a covering number bounded by $|\mathcal{N}(\epsilon; \mathcal{W}, \| \cdot \|_2)| \leqslant (1 + 4H\sqrt{d}/\epsilon)^d$. Clearly, this value upper bounds the covering number of class $\mathcal{V}$ and the result follows. $\qquad \square$

**Lemma B.13** (Lemma D.4 of [23])**.** *Let $\{s_k\}_{k=1}^{\infty}$ be a stochastic process on state space $\mathcal{S}$ with corresponding filtration $\{\mathcal{F}_k\}_{k=0}^{\infty}$. Let $\{\phi_k\}_{k=0}^{\infty}$ be an $\mathbb{R}^d$-valued stochastic process where $\phi_k \in \mathcal{F}_{k-1}$, and $\|\phi_k\|_2 \leqslant 1$. Let $\Lambda_\tau = I + \sum_{k=1}^{\tau} \phi_k \phi_k^{\mathsf{T}}$. Then, for any $\delta > 0$, with probability at least $1 - \delta$, for all $\tau \geqslant 0$, and any $V \in \mathcal{V}$ so that $\sup_{s \in \mathcal{S}} |V(s)| \leqslant H$, we have:*

$$\left\| \sum_{k=1}^{\tau} \phi_k \Big( V(s_k) - \mathbb{E}\left[ V(s_k) | \mathcal{F}_{k-1} \right] \Big) \right\|_{\Lambda_\tau^{-1}} \leqslant 4H^2 \left[ \frac{d}{2} \log(1 + \tau) + \log \frac{\mathcal{N}_\epsilon}{\delta} \right] + 8\tau^2 \epsilon^2,$$

*where $\mathcal{N}_\epsilon$ is the $\epsilon$-covering number of $\mathcal{V}$ with respect to the distance $dist(V, V') := \sup_{s \in \mathcal{S}} |V(s) - V'(s)|$.*

**Lemma B.14** (Lemma 3.2 of [63])**.** *With probability $1 - \delta/2$, for the function $u'$ defined as $u'_h(s,a) := \min \{ \beta' \|\phi(s,a)\|_{\Lambda_h^{-1}}, H \}$, with $\beta' := c'dH\sqrt{\log(dH\delta^{-1}\epsilon^{-1})}$, we have:*

$$J^*(u'/H) \leqslant c\sqrt{\frac{d^3 H^4 \log \frac{d\tau H}{\delta}}{\tau}},$$

*for some absolute constant $c > 0$.*

## C  Additional Insights on Compatibility

In this appendix, we collect and describe additional insights to the notion of *rewards compatibility* introduced in Section 4. The appendix is organized in the following manner: Appendix C.1 provides a visual explanation to the notion of rewards compatibility, in Appendix C.2 we analyse a multiplicative alternative to the definition of rewards compatibility, and Appendix C.3 discusses the conditions under which a learned reward can be used for "forward" RL, by comparing rewards with small (non)compatibility with rewards learned in previous works.

### C.1  A Visual Explanation for Rewards Compatibility

In this appendix, we aim to provide a visual intuition to the notion of rewards compatibility. For this reason, the reader should keep in mind Figure 3.

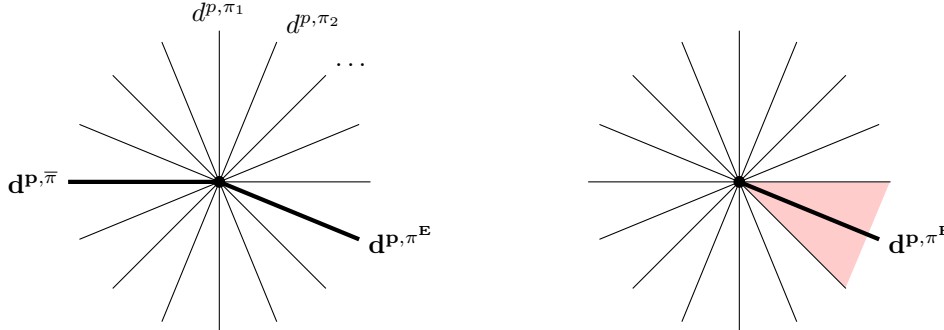

Figure 3: In this figure, the point at the center represents the initial state $s_0 = d_0$ of the environment $\mathcal{M}$, and each ray starting from it represents the occupancy measure $d^{p,\pi}$ of some policy $\pi$. The figure aims to provide the intuition that policies with rays close to each other induce similar visit distributions (e.g., both point towards the same direction in some grid-world), and policies with rays far away from each other point toward very different directions (i.e., they have different occupancy measures). The red area in the right denotes the set of directions (occupancy measures $d^{p,\pi}$ for some $\pi$) that are close in $\|\cdot\|_1$ norm to the direction of the expert $d^{p,\pi^E}$.

As explained in Section 4, even in the limit of infinite samples, i.e., even if we know $\mathcal{M} = (\mathcal{S}, \mathcal{A}, H, d_0, p) \cup \{\pi^E\}$ exactly, and even if we assume that the expert is exactly optimal, i.e., $J^*(r^E; p) - J^{\pi^E}(r^E; p) = 0$ (where $r^E$ is the true reward optimized by the expert), then we still do not have idea of how other policies perform. Expert demonstrations only provide information about the performance of a *single* policy, $\pi^E$, w.r.t. to the reference $J^*(r^E; p)$ under the unknown $r^E$, i.e., demonstrations say that $\pi^E$ in $r^E$ performs as good as $J^*(r^E; p)$. But what about other policies? Demonstrations provide no information.

To see this, consider Figure 3, in which each line exemplifies the visitation distribution induced by some policy $\pi \in \Pi$, and the point in the middle represents the starting state $s_0 = d_0$. Intuitively, observing $d^{p,\pi^E}$ along with knowing that $J^{\pi^E}(r^E; p)$ is good (i.e., because of expert demonstrations), does not tell us anything about the distribution $d^{p,\overline{\pi}}$ induced by some other policy $\overline{\pi}$ potentially arbitrarily different from $d^{p,\pi^E}$. Indeed, it might be the case that $J^{\overline{\pi}}(r^E; p)$ is acceptable, or that it is as good as $J^{\pi^E}(r^E; p)$, or that it is very bad. We cannot know from demonstrations only.

For this reason, if we consider the set of rewards with 0-(non)compatibility, i.e., the feasible reward set, we notice that it contains the rewards $r$ that make $\overline{\pi}$ optimal $J^{\overline{\pi}}(r; p) = J^*(r; p)$, but also the rewards $r'$ that make $\overline{\pi}$ nearly optimal $J^{\overline{\pi}}(r'; p) \approx J^*(r'; p)$, and also the rewards $r''$ that make $\overline{\pi}$ a very bad-performing policy $J^{\overline{\pi}}(r''; p) \ll J^*(r''; p)$. Indeed, as long as both $r, r', r''$ make the direction pointed by $d^{p,\pi^E}$ in Figure 3 a good direction, then they are in accordance with the constraint imposed by the demonstrations. The additional Degrees of Freedom (DoF) provided by policies beyond $\pi^E$ (e.g., $\overline{\pi}, \dots$) permit the ill-posedness of IRL.

We said that expert demonstrations provide information just about the performance of a single policy, $\pi^E$. However, to be precise, in the context of IRL, this is not correct. Indeed, differently from the mere learning from demonstrations setting, in which we just assume that $\pi^E$ is a very good-performing policy, in IRL we assume that the underlying problem is an MDP, i.e., that the expert agent is optimizing a reward function $r^E$.[10] This additional structure (i.e., that the underlying environment is indeed an MDP), makes sure that the performances of various directions $d^{p,\pi}$ in Figure 3 are measured through a dot product with a fixed reward function $r$, i.e.:

$$J^\pi(r; p) = \sum_{h \in [\![H]\!]} \langle d_h^{p,\pi}, r_h \rangle.$$

For this reason, we have the guarantee that the directions in the red area surrounding $d^{p,\pi^E}$ are almost as good as $d^{p,\pi^E}$. Indeed, for all policies $\pi$ such that $\sum_{h \in [\![H]\!]} \|d_h^{p,\pi} - d_h^{p,\pi^E}\|_1 \leqslant \epsilon$, i.e., for all policies

---

[10]When this assumption does not hold, we incur in model misspecification [55, 51].

$\epsilon$-close to $\pi^E$ in 1-norm, we can write:

$$\left| J^{\pi^E}(r^E; p) - J^\pi(r^E; p) \right| = \left| \sum_{h \in [\![H]\!]} \langle d_h^{p,\pi^E} - d_h^{p,\pi}, r_h^E \rangle \right| \leq \sum_{h \in [\![H]\!]} \| d_h^{p,\pi} - d_h^{p,\pi^E} \|_1 \leq \epsilon.$$

In other words, policies $\pi$ and $\pi^E$ have similar performances.

However, it should be remarked that, since we aim to recover the rewards explaining the expert's preferences, then we are guaranteed that policies close in 1-norm perform similarly under any reward function (by definition of 1-norm), and so we do not risk to incur in the error of representing $d^{p,\pi^E}$ and a direction $d^{p,\pi}$ inside the red area of Figure 3 with very different performances.

## C.2 A Multiplicative Compatibility

In Section 4, we have defined an *additive* notion of (non)compatibility, based on the difference of performance between $\pi^E$ and $\pi^*$ (the optimal policy). Here, we analyze a *multiplicative* notion of (non)compatibility, based on the ratio of the performances.[11]

We make the following observation. Any reward $r \in \mathfrak{R}$ induces, in the considered environment $p$, an ordering in the space of policies $\Pi$, based on the performance $J^\pi(r; p)$ of each policy $\pi \in \Pi$. It is easy to notice that for any scaling and translation parameters $\alpha \in \mathbb{R}_{>0}, \beta \in \mathbb{R}$, the reward constructed as $r'(\cdot, \cdot) = \alpha r(\cdot, \cdot) + \beta$ induces the same ordering as $r$ in the space of policies.[12]

For this reason, it seems desirable to use a notion of (non)compatibility such that rewards $r$ and $r'(\cdot, \cdot) = \alpha r(\cdot, \cdot) + \beta$ for some $\alpha, \beta$, suffer from the same (non)compatibility w.r.t. some expert policy $\pi^E$. However, observe that, for the notion of compatibility $\overline{\mathcal{C}}$ in Definition 4.1, we have that, for any $r \in \mathfrak{R}$:

$$\overline{\mathcal{C}}_{p,\pi^E}(r + \beta) = \overline{\mathcal{C}}_{p,\pi^E}(r) \qquad \forall \beta \in \mathbb{R},$$
$$\overline{\mathcal{C}}_{p,\pi^E}(\alpha r) = \alpha \overline{\mathcal{C}}_{p,\pi^E}(r) \neq \overline{\mathcal{C}}_{p,\pi^E}(r) \qquad \forall \alpha \in \mathbb{R}_{>0}.$$

Simply put, for the additive notion of (non)compatibility $\overline{\mathcal{C}}$, the scale ($\alpha$) of a reward matters, and rescaling the reward modifies the (non)compatibility.

To solve this issue, one might introduce a *multiplicative* notion of compatibility $\mathcal{F}$ (defined only for non-negative rewards and setting $\mathcal{F}_{p,\pi^E}(r) = 0$ when the denominator is 0):

$$\mathcal{F}_{p,\pi^E}(r) := \frac{J^{\pi^E}(r; p)}{J^*(r; p)}.$$

Clearly, the larger $\mathcal{F}_{p,\pi^E}(r)$, the closer is the performance of $\pi^E$ to the optimal performance. Observe that, for this definition, we have:

$$\mathcal{F}_{p,\pi^E}(\alpha r) = \mathcal{F}_{p,\pi^E}(r) \qquad \forall \alpha \in \mathbb{R}_{>0}$$
$$\mathcal{F}_{p,\pi^E}(r + \beta) \neq \mathcal{F}_{p,\pi^E}(r) \qquad \forall \beta \in \mathbb{R},$$

i.e., this definition does not care about the scaling $\alpha$ of the reward, but it is sensitive to the actual position $\beta$ of that reward.

Therefore, both $\overline{\mathcal{C}}$ and $\mathcal{F}$ suffer from some "rescaling" issues. Is it possible to devise a notion of compatibility, i.e., a measure of suboptimality, for a policy, that is independent of both the scale $\alpha$ and position $\beta$? Formally, we are looking for a function (notion of distance) $f : \mathbb{R} \times \mathbb{R} \to \mathbb{R}_{\geq 0}$ such that, for any $J_1, J_2 \in \mathbb{R}$:

$$f(\alpha J_1 + \beta, \alpha J_2 + \beta) = f(J_1, J_2), \tag{3}$$

for all $\alpha \in \mathbb{R}_{>0}, \beta \in \mathbb{R}$. Unfortunately, this is not possible, since it is easy to show that all the functions $f$ of this kind are of the following type:

$$\forall J_1, J_2 \in \mathbb{R} \times \mathbb{R} : \quad f(J_1, J_2) = \begin{cases} K_+ & \text{if } J_1 > J_2 \\ K_0 & \text{if } J_1 = J_2 \\ K_- & \text{if } J_1 < J_2 \end{cases},$$

---

[11] E.g., see Theorem 7.2.7 in [45], which is inspired by [43].
[12] Indeed, simply observe that, for any $\pi \in \Pi$: $J^\pi(r'; p) = J^\pi(\alpha r + \beta; p) = \alpha J^\pi(r; p) + \beta$.

for some reals $K_+, K_0, K_-$. In words, any function $f$ that satisfies Equation (3) is able to express just an ordering between inputs $J_1$ and $J_2$, but not an actual measure of sub-optimality/compatibility.

We conclude by stating that we prefer to use $\overline{\mathcal{C}}$ instead of $\mathcal{F}$ for the following reasons:

- First, most RL literature prefers the additive notion of suboptimality towards the multiplicative one.

- The additive notion of suboptimality is simpler to analyze w.r.t. the multiplicative one.

### C.3 When can a learned reward be used for "forward" RL?

In this appendix, we exploit the intuition developed in Appendix C.1 to discuss under which conditions we can exploit demonstrations alone to recover a single reward that *can be used for "forward" RL*, i.e., to recover a single reward $r$ for which we have the guarantee that any $\epsilon$-optimal policy $\pi$ to $r$ in the true environment $p$ has similar performance in the same environment $p$ under the true reward $r^E$, that is, policy $\pi$ is an $f(\epsilon)$-optimal policy to $r^E$ in $p$, for some function $f$.

Applications of IRL range from Apprenticeship Learning (AL), to reward design, to interpretability of expert's preferences. Concerning AL, it is common to "use" the reward $r$ learned through IRL to optimize our learning agent. But what properties $r$ should satisfy in order to obtain performance guarantees on our learning agent w.r.t. the true (unknown) $r^E$? We now list and analyze various plausible requirements.

- First, we might ask that, being $\pi^E$ optimal w.r.t. $r^E$, then $\pi^E \in \arg\max_\pi J^\pi(r)$, i.e., that the expert policy $\pi^E$ is optimal under the learned reward $r$. However, this requirement is not satisfactory for the following reason. Reward $r$ might induce more than one optimal policy (e.g., it might induce both $\overline{\pi}, \pi^E$ as optimal), and optimal policies other than $\pi^E$ (e.g., $\overline{\pi}$) are not guaranteed to perform well under $r^E$ (actually, $\overline{\pi}$ can be any policy in $\Pi$). Clearly, this is not satisfactory. Observe that there are rewards in the feasible set $\mathcal{R}_{p,\pi^E}$ for which multiple policies are optimal (thus, not all the rewards in the feasible set are satisfactory).

- We might additionally ask that $\pi^E$ is the unique optimal policy of reward $r$ (similarly to what happens in entropy-regularized MDPs [70, 15]). However, this is not satisfactory for the following reason. In practice, it is really difficult (almost impossible) to compute the optimal policy of a given reward. Thus, what is usually done in RL, is to settle for an $\epsilon$-optimal policy. Since any policy can be $\epsilon$-optimal under reward $r$, then no guarantee we can have for such policy w.r.t. $r^E$.

- What if we ask that $\pi^E$ is at least $\epsilon$-optimal under $r$ (i.e., the requirement provided by $\epsilon$-(non)compatible rewards)? Well, this is not satisfactory because optimal policies can be any, and because there might be other $\epsilon$-optimal policies that can perform arbitrarily bad under $r^E$.

All the three requirements described above on $r$ do not provide guarantees that optimizing the considered reward $r$ provides a policy with satisfactory performance w.r.t. the true $r^E$. However, as mentioned in Section 4 and in Appendix C.1, expert demonstrations *do not provide any information about the performance of policies other than $\pi^E$ under $r^E$*.

**Remark C.1.** *If we want to be sure that an $\epsilon$-optimal policy $\pi$ for the learned reward $r$ in $p$ is if $f(\epsilon)$-optimal for $r^E$ in $p$ (for some function $f$), then, clearly, we need that* all the (at least) $\epsilon$-optimal policies under the learned $r$ have visitation distribution close to that of $\pi^E$ in 1-norm *(see Appendix C.1).*

We stress that many IRL algorithms for AL, like max-margin [2], learn a reward function just as a mere mathematical tool to compute a policy $\pi$ which is close in 1-norm $\|d^\pi - d^{\pi^E}\|_1$ to $\pi^E$.

**A remark about works on the feasible set.** If we look at recent works about the feasible set [38, 31, 68], it might seem that these works are able to provide guarantees between $r, r^E$ under distance $d^{all}$ (see Section 3.1 of [68]), defined as:

$$d^{all}(r, r^E) := \sup_{\pi \in \Pi} |J^\pi(r) - J^\pi(r^E)|.$$

If $d^{all}(r, r^E)$ is small, then *the performance of any policy in $r$, not just optimal policy or $\epsilon$-optimal policy, is similar also under $r^E$.* In other words, if we use/optimize reward $r$, then we have the guarantee that the performance of the retrieved policy under $r^E$ is more or less the same as its performance in $r$. Therefore, clearly, *rewards $r$ with small distance to $r^E$ w.r.t. $d^{all}$* can *be used for "forward" RL.* However, we have the following result:

**Proposition C.1.** *Let $\mathcal{M} = (\mathcal{S}, \mathcal{A}, H, d_0, p)$ be a known MDP without reward, and let $\pi^E$ be a known expert's policy. Let $r^E$ the true unknown reward optimized by the expert to construct $\pi^E$. Then, there does not exist a learning algorithm that receives in input the pair $(\mathcal{M}, \pi^E)$ and outputs a single reward $r$ such that $d^{all}(r, r^E) \leqslant \epsilon$ w.p. $1 - \delta$.*

*Proof.* The proof is trivial. Indeed, since the feasible set $\mathcal{R}_{p,\pi^E}$ contains an infinite amount of reward functions along with $r^E$, and the learning algorithm cannot discriminate $r^E$ inside $\mathcal{R}_{p,\pi^E}$, then the best it can do is to output an arbitrary reward function $r \in \mathcal{R}_{p,\pi^E}$. However, since $\mathcal{R}_{p,\pi^E}$ contains, for any reward $r \in \mathcal{R}_{p,\pi^E}$, at least another reward $r' \in \mathcal{R}_{p,\pi^E}$ such that $d^{all}(r, r') = c$ is finite and equal to some positive constant $c > 0$,[13] then we can simply construct the problem instance with $r^E := r'$ to make the learning algorithm not able to output rewards that can be used for forward learning. $\square$

Nevertheless, [38, 31, 68] seem to provide sample efficient algorithms w.r.t. $d^{all}$.[14] By looking at Proposition C.1, we realize that this is clearly a *contradiction*. What is the right interpretation?

The trick is that the algorithms proposed in works [68, 38, 31] are *not* able to output a single reward $r$ which is close to $r^E$ w.r.t. $d^{all}$, but, *for any possible reward $r^E = r^E(V, A)$ parametrized*[15] *by some value and advantage functions $V, A$, they are able to output a reward $r$ such that $d^{all}(r, r^E(V, A))$ is small.* In other words, it is like if these works *assume to know* the $V, A$ parametrization of the true reward $r^E$. Simply put, these works are able to output a reward $r$ that can be used for "forward" RL just under such assumption. Otherwise those algorithms do not provide such guarantee.

**Conclusions.**   To sum up, we conclude that, in general, an arbitrary reward function with small (non)compatibility can *not* be used for "forward" learning (see Proposition C.1), because we cannot know given demonstrations alone whether the performances assigned by such reward to policies other than the expert policy are meaningful. In addition, for the same reason, we realize that also an arbitrary reward with zero (non)compatibility, i.e., an arbitrary reward in the feasible set, can *not* be used for "forward" learning.

### C.4   Comparing the (non)compatibility of various rewards

In Section 4, we said that rewards $r$ with smaller values of $\overline{\mathcal{C}}_{p,\pi^E}(r)$ are more compatible with $\pi^E$ in $\mathcal{M} = (\mathcal{S}, \mathcal{A}, H, d_0, p)$. However, one might provide the following "counter-example":

**Example C.1** (Question by Reviewer KyLX). *Let $r^1, r^2$ be two rewards such that $r_h^2(s, a) = 2r_h^1(s, a) \geqslant 0$ for all $(s, a, h) \in \mathcal{S} \times \mathcal{A} \times [\![H]\!]$. Then, clearly, $\overline{\mathcal{C}}_{p,\pi^E}(r^2) = 2\overline{\mathcal{C}}_{p,\pi^E}(r^1)$. Therefore, based on Section 4, we say that reward $r^1$ is more compatible than $r^2$ w.r.t. $\pi^E$ in $\mathcal{M}$. However, since $r^2$ is just $r^1$ re-scaled by a constant, the two MDPs $\mathcal{M} \cup \{r^1\}$ and $\mathcal{M} \cup \{r^2\}$ should be "equivalent", thus, $r^1$ and $r^2$ should be, intuitively, equally compatible with $\pi^E$.*

However, Example C.1 misleads the correct *interpretation* of the notion of reward function in MDPs, and in particular about the *scale* of the rewards. Let us explain better our point.

The MDP is a model, i.e., a simplified representation of reality, which is commonly applied to 2 different kinds of real-world scenarios: $(i)$ problems in which the agent (learner in RL or expert in IRL) actually receives some kind of scalar feedback from the environment, which can be modelled as a reward function; $(ii)$ problems in which the agent does not receive a feedback from the environment,

---

[13]This is immediate from the considerations in Appendix C.1.

[14]Actually, [38, 31] use different notions of distance, like $d_\infty(r, r') := \|r - r'\|_\infty$. However, we can write $\|r - r'\|_\infty \geqslant \|r - r'\|_1/(SAH)$, and by dual norms we have that $d^{all}(r, r') = \sup_{\pi \in \Pi} |\langle d^{p,\pi}, r - r'\rangle| \leqslant \sup_{\overline{d}: \|\overline{d}\|_\infty \leqslant 1} |\langle \overline{d}, r - r'\rangle| = \|r - r'\|_1$. Therefore, the guarantees of [38, 31] can be converted too $d^{all}$ guarantees too.

[15]While [68] makes this parametrization explicit, [38, 31] keep the parametrization implicit, but everything is analogous.

but its objective, i.e., its structure of preferences among state-action trajectories (which trajectories are better than others), satisfies some axioms that permit to represent it through a scalar reward [52, 9] (this is referred to as the Reward Hypothesis in literature [58]).

There is an enormous difference between scenario $(i)$ and scenario $(ii)$. In $(i)$ the notion of $\epsilon$-optimal policy is well-defined for any fixed $\epsilon > 0$, because the reward function is given and, thus, fixed. Instead, in $(ii)$, the notion of reward function is a mere mathematical artifact used to represent preferences among trajectories, whose existence is guaranteed by a set of assumptions/axioms [52, 9, 58]. As Example C.1 shows, positive affine transformations of the reward do not affect the structure of preferences represented (see [52] or Section 16.2 of [50] or [30]). Therefore, in $(ii)$, the notion of $\epsilon$-optimal policy is not well-defined, because rescaling a reward function $r$ to $kr$ changes the suboptimality of some policy $\pi$ from $\epsilon$ to $k\epsilon$. In other words, for fixed $\epsilon > 0$, any policy can be made $\epsilon$-optimal by simply rescaling a reward $r$ to $kr$ for some small enough $k > 0$.

In IRL, this issue is even more influential because, although we are in setting $(i)$, we have no idea on the scale of the true reward function. For this reason, our solution is to attach to any reward $r$ a notion of compatibility $\overline{\mathcal{C}}(r)$ which implicitly contains information about the scale of the reward $r$. Compatibilities of different rewards (e.g., $r^1$ and $r^2$ in Example C.1) cannot be compared unless the rewards have the same scale (e.g., $r^1$ and $r^2$ have different scales, thus their compatibilities shall not be compared).

It should be observed that in Appendix C.2 we discuss a notion of compatibility independent of the scale of the reward. However, we show that it suffers from major drawbacks that make the notion of compatibility introduced in the main paper (Definition 4.1) more suitable for the IRL problem.

In conclusion, to settle Example C.1, rewards $r^1$ and $r^2$ should not have the same compatibility, because they have different scales, and the notion of compatibility (i.e., suboptimality) is strictly connected to the scale of the reward. To carry out a fair comparison of compatibilities, one should rescale the compatibility of each reward based on the scale of the reward.

# D   Missing Proofs and Additional Results for Section 5

This appendix is organized as follows. First, we report the full pseudo-code of `CATY-IRL`. Then, we provide the proof of Theorem 5.1 in Appendix D.2.

## D.1   Algorithm

In this section, we provide the extended version of `CATY-IRL` containing the explicit conditions under which we shall instantiate one BPI/RFE algorithm instead of another.

---

**Algorithm 2: `CATY-IRL`- exploration**

**Data:** Failure probability $\delta > 0$, target accuracy $\epsilon > 0$, expert demonstrations $\mathcal{D}^E$, set of rewards to classify $\mathcal{R}$, problem structure $\imath \in \{$tabular, linear rewards, Linear MDP$\}$

1  **if** $\imath \in \{$*tabular, linear rewards*$\}$ **then**
2      **if** $|\mathcal{R}|$ *is a small constant* **then**
3          $\mathcal{D} \leftarrow \{\}$
4          **for** $r' \in \mathcal{R}$ **do**
5              $\mathcal{D} \leftarrow \mathcal{D} \cup$ BPI_Exploration$(\delta, \epsilon/2, r')$       `/* Algorithm BPI-UCBVI [37] */`
6          **end**
7      **else**
8          $\mathcal{D} \leftarrow$ RFE_Exploration$(\delta, \epsilon/2)$             `/* Algorithm RF-Express [37] */`
9      **end**
10 **else**
11     $\mathcal{D} \leftarrow$ RFE_Exploration$(\delta, \epsilon/2)$                  `/* Algorithm RFLin [62] */`
12 **end**
13 Return $\mathcal{D}$

---

---

**Algorithm 3:** `CATY-IRL`- classification

---

**Data:** Failure probability $\delta > 0$, target accuracy $\epsilon > 0$, expert demonstrations $\mathcal{D}^E$, classification threshold $\Delta \in \mathbb{R}$, reward to classify $r \in \mathcal{R}$, problem structure $\imath \in \{$tabular, linear rewards, Linear MDP$\}$, dataset $\mathcal{D}$

    `// Estimate the expert's performance` $\widehat{J}^E(r)$`:`

1 **if** $\imath = $ *tabular* **then**

2     $\widehat{d}^E \leftarrow$ empirical estimate of $d^{p,\pi^E}$ from $\mathcal{D}^E$

3     $\widehat{J}^E(r) \leftarrow \sum_h \langle \widehat{d}_h^E, r_h \rangle$

4 **else**

5     $\widehat{\psi}^E \leftarrow$ empirical estimate of $\psi^{p,\pi^E}$ from $\mathcal{D}^E$

6     $\widehat{J}^E(r) \leftarrow \sum_h \langle \widehat{\psi}_h^E, r_h \rangle$

7 **end**

    `// Estimate the optimal performance` $\widehat{J}^*(r)$`:`

8 **if** $\imath \in \{$*tabular*, *linear rewards*$\}$ **then**

9     **if** $|\mathcal{R}|$ *is a small constant* **then**

10         $\widehat{J}^*(r) \leftarrow$ BPI_Planning$(\mathcal{D}, r)$                 `/* Algorithm BPI-UCBVI [37] */`

11

12     **else**

13         $\widehat{J}^*(r) \leftarrow$ RFE_Planning$(\mathcal{D}, r)$                `/* Algorithm RF-Express [37] */`

14

15     **end**

16 **else**

17     $\widehat{J}^*(r) \leftarrow$ RFE_Planning$(\mathcal{D}, r)$                    `/* Algorithm RFLin [62] */`

18

19 **end**

    `// Classify the reward:`

20 $\widehat{\mathcal{C}}(r) \leftarrow \widehat{J}^*(r) - \widehat{J}^E(r)$

21 class $\leftarrow$ True **if** $\widehat{\mathcal{C}}(r) \leqslant \Delta$ **else** False

22 **return** class

---

### D.2   Proof of Theorem 5.1

Notice that, according to Definition 4.3, an algorithm is $(\epsilon, \delta)$-PAC for IRL if it computes an estimate $\epsilon$-close to the true (non)compatibility w.h.p.. Such definition does not depend on the specific strategy adopted by the algorithm to actually classify the input reward using the computed estimate of (non)compatibility.

Before diving into the proof of Theorem 5.1, we make the following considerations.

In the common tabular MDPs setting without additional structure, we know that the expected utility $J^\pi(r; p)$ of policy $\pi$ under reward $r$ in environment with dynamics $p$ can computed as:

$$J^\pi(r; p) = \sum_{h \in [\![H]\!]} \langle r_h, d_h^{p,\pi} \rangle,$$

where $d_h^{p,\pi}$ is the occupancy measure of policy $\pi$ in $p$. It should be remarked that both $r_h$ and $d_h^{p,\pi}$ have $SA$ components for all $h \in [\![H]\!]$.

In tabular MDPs with linear reward functions and in Linear MDPs, the reward function is linear in some feature map $\phi$, i.e.:

$$r_h(\cdot, \cdot) = \langle \phi(\cdot, \cdot), \theta_h \rangle \qquad \forall h \in [\![H]\!],$$

where $\|\phi(s, a)\|_2 \leqslant 1$ for all $(s, a) \in \mathcal{S} \times \mathcal{A}$ and $\max_h \|\theta_h\|_2 \leqslant \sqrt{d}$. Using this decomposition, we can rewrite the expected utility $J^\pi(r; p)$ as:

$$J^\pi(r; p) = \sum_{h \in [\![H]\!]} \langle r_h, d_h^{p,\pi} \rangle$$

$$= \sum_{h \in [\![H]\!]} \langle \theta_h^\intercal \phi, d_h^{p,\pi} \rangle$$

$$= \sum_{h \in [\![H]\!]} \theta_h^\intercal \mathop{\mathbb{E}}_{(s,a) \sim d_h^{p,\pi}} \phi(s,a)$$

$$= \sum_{h \in [\![H]\!]} \theta_h^\intercal \psi_h^{p,\pi},$$

where we have defined the feature expectations $\{\psi_h^{p,\pi}\}_{h \in [\![H]\!]}$ as $\psi_h^{p,\pi} := \mathbb{E}_{(s,a) \sim d_h^{p,\pi}} \phi(s,a)$. Observe that vector $\psi_h^{p,\pi}$ has $d$ components instead of the $SA$ components of each $d_h^{p,\pi}$ vector.

Since in our setting the IRL algorithm receives in input the reward function (or its parameter $\theta \in \mathbb{R}^d$), to estimate the expected utility $J^\pi(r;p)$ we must estimate the visit distributions $\{d_h^{p,\pi}\}_h$ or the feature expectations $\{\psi_h^{p,\pi}\}_h$. However, because of the different dimensionalities of such quantities ($SA$ versus $d$), the estimates might require different amounts of samples.

**Theorem 5.1** (Sample Complexity of `CATY-IRL`). *Let $\epsilon, \delta \in (0, 1)$. Then `CATY-IRL` is $(\epsilon, \delta)$-PAC for IRL with a sample complexity upper bounded by:*

*Tabular MDPs:*
$$\tau^E \leqslant \tilde{\mathcal{O}}\Big(\frac{H^3 SA}{\epsilon^2} \log \frac{1}{\delta}\Big), \quad \tau \leqslant \tilde{\mathcal{O}}\Big(\frac{H^3 SA}{\epsilon^2}\Big(N + \log\frac{1}{\delta}\Big)\Big),$$

*Tabular MDPs with linear rewards:*
$$\tau^E \leqslant \tilde{\mathcal{O}}\Big(\frac{H^3 d}{\epsilon^2} \log \frac{1}{\delta}\Big), \quad \tau \leqslant \tilde{\mathcal{O}}\Big(\frac{H^3 SA}{\epsilon^2}\Big(N + \log\frac{1}{\delta}\Big)\Big),$$

*Linear MDPs:*
$$\tau^E \leqslant \tilde{\mathcal{O}}\Big(\frac{H^3 d}{\epsilon^2} \log \frac{1}{\delta}\Big), \quad \tau \leqslant \tilde{\mathcal{O}}\Big(\frac{H^5 d}{\epsilon^2}\Big(d + \log\frac{1}{\delta}\Big)\Big),$$

*where $N = 0$ if $|\mathcal{R}| = \Theta(1)$, and $N = S$ otherwise.*

*Proof.* To prove the theorem, we aim to find a bound to the number of samples $\tau^E$ such that the estimate $\widehat{J}^E(r) \approx J^{\pi^E}(r;p)$ is $\epsilon/2$-correct with probability at least $1 - \delta/2$. Next, similarly, we aim to bound $\tau$ so that $\widehat{J}^*(r) \approx J^*(r;p)$ is $\epsilon/2$-correct with probability at least $1 - \delta/2$. Then, the conclusion follows after performing a union bound and observing that, for any $r \in \mathcal{R}$:

$$\left| \overline{\mathcal{C}}_{p,\pi^E}(r) - \widehat{\mathcal{C}}(r) \right| = \left| \Big( J^*(r;p) - J^{\pi^E}(r;p) \Big) - \Big( \widehat{J}^*(r) - \widehat{J}^E(r) \Big) \right|$$

$$\leqslant \left| J^*(r;p) - \widehat{J}^*(r) \right| + \left| J^{\pi^E}(r;p) - \widehat{J}^E(r) \right|$$

$$\leqslant \frac{\epsilon}{2} + \frac{\epsilon}{2} = \epsilon.$$

**Estimating $\widehat{J}^E(r) \approx J^{\pi^E}(r;p)$**

To estimate $J^{\pi^E}(r;p)$, `CATY-IRL` simply computes the empirical estimate of $\{d_h^{p,\pi^E}\}$ in case of tabular MDPs, and the empirical estimate of $\{\psi_h^{p,\pi^E}\}$ in case of tabular MDPs with linear rewards and Linear MDPs. Notice that by empirical estimates we mean:

$$\widehat{d}_h^E(s,a) := \frac{\sum_{i \in [\![\tau^E]\!]} \mathbb{1}\{s_h^i = s \wedge a_h^i = a\}}{\sum_{i \in [\![\tau^E]\!]} \mathbb{1}\{s_h^i = s\}} \qquad \forall (s,a,h) \in \mathcal{S} \times \mathcal{A} \times [\![H]\!],$$

and:

$$\widehat{\psi}_h^E := \frac{\sum_{i \in [\![\tau^E]\!]} \phi(s_h^i, a_h^i)}{\tau^E} \qquad \forall h \in [\![H]\!].$$

Concerning the estimate of the visit distribution $\widehat{d}^E$, we can use the result of Lemma 6 in [53] (we are working with bounded rewards), to obtain that:

$$\sum_{h \in [\![H]\!]} \|d_h^{p,\pi^E} - \widehat{d}_h^E\|_1 \leqslant \sqrt{\frac{SAH^3 \log \frac{8SAH}{\delta}}{2\tau^E}} \leqslant \frac{\epsilon}{2}.$$

Solving w.r.t. $\tau^E$ we get the bound on $\tau^E$.

In a completely analogous manner, we can bound the feature expectations as:

$$\sum_{h \in [\![H]\!]} \|\psi_h^{p,\pi^E} - \widehat{\psi}_h^E\|_1 \leqslant \sqrt{\frac{dH^3 \log \frac{8dH}{\delta}}{2\tau^E}} \leqslant \frac{\epsilon}{2}.$$

Again, solving w.r.t. $\tau^E$ we get the bound on $\tau^E$.

**Estimating $\widehat{J}^*(r) \approx J^*(r; p)$**

Let us begin with the case in which $\mathcal{R}$ is large. As explained for instance in Definition 4 of [66], both algorithms RF-Express [37] and RFLin [62] satisfy the *uniform policy evaluation property*, i.e., they guarantee that, for any $\epsilon, \delta \in (0, 1)$, after having explored for $\tau \leqslant \widetilde{\mathcal{O}}\left(\frac{H^3 SA}{\epsilon^2}\left(S + \log\frac{1}{\delta}\right)\right)$ in case of RF-Express [37], and $\tau \leqslant \widetilde{\mathcal{O}}\left(\frac{H^5 d}{\epsilon^2}\left(d + \log\frac{1}{\delta}\right)\right)$ for the algorithm in [62] (we omit linear terms in $1/\epsilon$), they compute an estimate $\widehat{p} \approx p$ of the true transition model such that:

$$\mathbb{P}\Big(\sup_{r \in \mathfrak{R}, \pi \in \Pi} \big|J^\pi(r; p) - J^\pi(r; \widehat{p})\big| \leqslant \epsilon\Big) \geqslant 1 - \delta.$$

Clearly, if such property holds, then by computing the performance of the policy $\widehat{\pi}$ outputted by the RFE algorithm we are able to obtain an $\epsilon/2$-correct estimate of $J^*(r; p)$.[16]

Concerning the case in which $|\mathcal{R}|$ is a finite small constant, for tabular and tabular with linear rewards MDPs, we can simply use algorithm BPI-UCBVI of [37] as sub-routine, and run it as many times as there are rewards in $\mathcal{R}$. When $|\mathcal{R}|$ is a small constant, we can proceed with a union bound over $\mathcal{R}$:

$$\mathbb{P}\Big(\sup_{r \in \mathfrak{R}, \pi \in \Pi} \big|J^\pi(r; p) - J^\pi(r; \widehat{p})\big| \leqslant \epsilon\Big) \geqslant 1 - \sum_{r \in \mathcal{R}} \mathbb{P}\Big(\sup_{\pi \in \Pi} \big|J^\pi(r; p) - J^\pi(r; \widehat{p})\big| > \epsilon\Big) \geqslant 1 - |\mathcal{R}|\delta.$$

This allows us to formally distinguish between small and large $|\mathcal{R}|$ based on the following inequality:

$$S + \log\frac{1}{\delta} < \log\frac{|\mathcal{R}|}{\delta} \implies S < \log|\mathcal{R}|.$$

$\square$

# E   Missing Proofs and Additional Results for Section 6.1

This appendix is organized as follows. First, in Appendix E.1, we introduce two problems that share similarities with RFE and IRL, and we characterize the main differences among them. In addition, we enunciate a lower bound to the sample complexity that is common to some of these 4 problems. Next, in Appendix E.2, we provide the missing proofs.

## E.1   Four Problems

The 4 problems that we consider here are Reward-Free Exploration (RFE), Inverse Reinforcement Learning (IRL), Matching Performance (MP), and Imitation Learning from Demonstrations alone (ILfO). MP represents a novel generalization of RFE, while ILfO, introduced in [34], represents an exemplification of MP. Before enunciating the minimax lower bound, it is important to formally define each of these problems, as well as what we mean by learning in each problem.

### E.1.1   Definition of the Problems

In all the 4 problems, the learner is placed into an *unknown* MDP without reward $\mathcal{M} = (\mathcal{S}, \mathcal{A}, H, d_0, p)$, i.e., an environment whose dynamics $(d_0, p)$ is unknown to the learner. For simplicity, w.l.o.g., we assume that there is a single initial state $s_0 := d_0$. In each problem, the learner can explore the environment at will to collect samples about the dynamics $p$, whose knowledge improves the performance of the agent at solving the task. However, at exploration phase, the learner does not

---

[16]Actually, for Linear MDPs, instead of evaluating the policy returned by Algorithm 2 of [62], we can simply consider the optimistic estimate of the $V$-function computed by such algorithm, which has the property of being $\epsilon$-close to the true optimal $V$-function.

know which is the specific task it has to solve. It just knows that the specific task belongs to a given set of tasks $\mathfrak{T}$ (e.g., set of reward functions). The agent can use the knowledge of $\mathfrak{T}$ to engage in a more efficient task-driven exploration. For any $\epsilon, \delta \in (0, 1)$, the goal of the agent is to being able to ouputting, for any task in $\mathfrak{T}$ a quantity $\mathfrak{o}$ (e.g., a policy) that solves that specific task in an $\epsilon$-correct manner with probability at least $1 - \delta$. The ultimate goal of exploration is to collect the least number of samples that permits $(\epsilon, \delta)$-correctness for all the tasks in $\mathfrak{T}$.

Now, let us see what the quantities $\mathfrak{T}$ and $\mathfrak{o}$ represent in each of the 4 problems. In Table 1, we provide a sum up of the various definitions.

**Reward-Free Exploration (RFE).** In RFE, the learner receives a set of reward functions $\mathfrak{T} = \mathcal{R} \subseteq \mathfrak{R}$ in input, and the goal is to exploit the information about $p$ collected at exploration phase to output, for any reward $r \in \mathcal{R}$, an $\epsilon$-optimal policy $\mathfrak{o} = \widehat{\pi}_r$ w.p. $1 - \delta$. When $\mathfrak{T} = \{r\}$ is a singleton, the RFE problem is commonly termed the BPI problem. In symbols, any RFE algorithm must guarantee that:

$$\mathbb{P}\left( \sup_{r \in \mathcal{R}} J^*(r; p) - J^{\widehat{\pi}_r}(r; p) \leqslant \epsilon \right) \geqslant 1 - \delta,$$

where $\widehat{\pi}_r$ is the estimate of the algorithm for reward $r$.

**Inverse Reinforcement Learning (IRL).** In IRL, the learner receives in input an occupancy measure[17] $\{d_h^{p,\pi^E}\}_{h \in [\![H]\!]}$ and a set of reward functions $\mathcal{R} \subseteq \mathfrak{R}$: $\mathfrak{T} = (d^{p,\pi^E}, \mathcal{R})$, but it does not know which specific reward it will have to classify. Under the assumption that the occupancy measure $d^{p,\pi^E}$ is known,[18] the problem reduces to exploiting the information about $p$ collected at exploration phase to output, for any reward $r \in \mathcal{R}$, an $\epsilon$-correct estimate $\mathfrak{o} = \widehat{J}(r)$ of the optimal utility $J^*(r)$ w.p. $1 - \delta$. In symbols, under these conditions, any IRL algorithm must guarantee that:

$$\mathbb{P}\left( \sup_{r \in \mathcal{R}} \left| J^*(r; p) - \widehat{J}(r) \right| \leqslant \epsilon \right) \geqslant 1 - \delta,$$

where $\widehat{J}(r)$ is the estimate of the algorithm for reward $r$.

**Matching Performance (MP).** In MP, the learner receives in input a set of reward functions $\mathcal{R} \subseteq \mathfrak{R}$ and a measure of performance for each of them $\overline{J} : \mathcal{R} \to \mathbb{R}$: $\mathfrak{T} = (\overline{J}, \mathcal{R})$. For any $r \in \mathcal{R}$, the utility $\overline{J}(r)$ represents a performance measure for which we aim to find the policy that achieves closest performance. Thus, in MP, the goal is to exploit the information about $p$ collected at exploration phase to output, for any reward $r \in \mathcal{R}$, a policy $\mathfrak{o} = \widehat{\pi}_r$ such that, if we denote the policy with performance closest to $\overline{J}(r)$ by $\overline{\pi}_r \in \arg\min_\pi |J^\pi(r) - \overline{J}(r)|$, then the utility of policy $\widehat{\pi}_r$ is $\epsilon$-close to the utility of policy $\overline{\pi}_r$ w.p. $1 - \delta$. In symbols, any MP algorithm must guarantee that:

$$\mathbb{P}\left( \sup_{r \in \mathcal{R}} \left| J^{\overline{\pi}_r}(r; p) - J^{\widehat{\pi}_r}(r; p) \right| \leqslant \epsilon \right) \geqslant 1 - \delta,$$

where $\overline{\pi}_r \in \arg\min_\pi |J^\pi(r) - \overline{J}(r)|$, and $\widehat{\pi}_r$ is the estimate of the algorithm for reward $r$.

**Imitation Learning from Demonstrations alone (ILfO).** In ILfO, the learner receives in input a set of *state-only* reward functions $\mathcal{R} \subset \mathfrak{R}$ and a *state-only* occupancy measure $\{\overline{d}_h\}_{h \in [\![H]\!]}$: $\mathfrak{T} = (\overline{d}, \mathcal{R})$. Under the assumption that $\overline{d}$ does not leak any information about the true transition model $p$, the goal is to exploit the information about $p$ collected at exploration phase to output, for any reward $r \in \mathcal{R}$, a policy $\mathfrak{o} = \widehat{\pi}_r$ such that, if we denote the policy with performance closest to $\overline{J}(r) := \sum_{h \in [\![H]\!]} \langle r_h, \overline{d}_h \rangle$ by $\overline{\pi}_r \in \arg\min_\pi |J^\pi(r) - \overline{J}(r)|$, then the utility of policy $\widehat{\pi}_r$ is $\epsilon$-close to the utility of policy $\overline{\pi}_r$ w.p. $1 - \delta$. Simply put, ILfO, as defined in this manner, exemplifies the MP setting by providing a functional form to $\overline{J} : \mathcal{R} \to \mathbb{R}$ as an inner product between a certain state-only occupancy measure and the input reward. It should be remarked that the assumption made for ILfO is mild, because it is

---

[17]Actually, as explained in Section 6.1, the knowledge of $d^{p,\pi^E}$ at exploration phase is useless. The visit measure might be provided after the exploration along with the true reward to classify.

[18]The assumption that $d^{p,\pi^E}$ is known is useful to reduce the estimation problem of the (non)compatibility of a reward $\overline{\mathcal{C}}_{p,\pi^E}(r) := J^*(r; p) - J^{\pi^E}(r; p)$ to the problem of estimating the optimal utility $J^*(r; p)$ only. Indeed, if $d^{p,\pi^E}$ is known, then, for any reward $r$, the utility $J^{\pi^E}(r; p)$ is known.

| | BPI | IRL | MP | ILfO |
|---|---|---|---|---|
| Set of Tasks $\mathfrak{T}$ | $\mathcal{R}$ | $(d^{p,\pi^E}, \mathcal{R})$ | $(\overline{J}, \mathcal{R})$ | $(\overline{d}, \mathcal{R})$ |
| Assumptions | / | $d^{p,\pi^E}$ known | $\overline{J}$ can be non-realisable | $r$ state-only, $\overline{d}$ no info |
| Output $\mathfrak{o}$ | $\widehat{\pi}$ | $\widehat{J}$ | $\widehat{\pi}$ | $\widehat{\pi}$ |
| Goal | $J^{\widehat{\pi}}(r;p) \approx J^*(r;p)$ | $\widehat{J} \approx J^*(r;p)$ | $J^{\widehat{\pi}}(r;p) \approx \overline{J}$ | $J^{\widehat{\pi}}(r;p) \approx \sum_h \langle \overline{d}_h, r_h \rangle$ |

Table 1: Summary of the problems.

satisfied by the setting in which the expert and the learner have the same state space but different action spaces (or different dynamics). Indeed, in such case, the visit distribution $\overline{d}$ of the expert would not leak any information about $p$. In symbols, any ILfO algorithm must guarantee that:

$$\mathbb{P}\Big( \sup_{r \in \mathcal{R}} \big| J^{\overline{\pi}_r}(r;p) - J^{\widehat{\pi}_r}(r;p) \big| \leqslant \epsilon \Big) \geqslant 1 - \delta,$$

where $\overline{\pi}_r \in \arg\min_\pi |J^\pi(r) - \overline{J}(r)|$ and $\overline{J}(r) := \sum_{h \in \llbracket H \rrbracket} \langle r_h, \overline{d}_h \rangle$, and $\widehat{\pi}_r$ is the estimate of the algorithm for reward $r$.

### E.1.2 Lower Bound

We now present a minimax lower bound rate that is common to RFE, IRL, and MP. We report here the lower bounds presented in Section 6.1.

**Theorem 6.1** (IRL Classification - Lower Bound). *Let $\mathfrak{A}$ be an $(\epsilon, \delta)$-PAC algorithm for the IRL classification in tabular MDPs. Let $\tau$ be the number of exploration episodes. Then, there exists an IRL classification instance such that:*

$$\text{if } |\mathcal{R}| \geqslant 1 : \ \tau \geqslant \Omega\Big( \frac{H^3 SA}{\epsilon^2} \log \frac{1}{\delta} \Big), \qquad \text{if } \mathcal{R} = \mathfrak{R} : \ \tau \geqslant \Omega\Big( \frac{H^3 SA}{\epsilon^2} \Big( S + \log \frac{1}{\delta} \Big) \Big).$$

*Proof.* The proof is similar to that of [38]. We split the proof in two parts, by considering two classes of difficult problem instances in Lemma E.2 and Lemma E.3. Next, we combine the two bounds through $\max\{a, b\} \geqslant (a + b)/2$ for all $a, b \geqslant 0$. For the proof, we will assume that the expert visit distribution is known. The obtained bound represents a lower bound to the more general setting in which it is unknown. $\square$

**Theorem E.1** (RFE - Refined Lower Bound). *Let $\mathfrak{A}$ be an $(\epsilon, \delta)$-PAC algorithm for RFE in tabular MDPs. Let $\tau$ be the number of exploration episodes. Then, there exists an RFE instance such that:*

$$\tau \geqslant \Omega\Big( \frac{H^3 SA}{\epsilon^2} \Big( S + \log \frac{1}{\delta} \Big) \Big).$$

*Proof.* The proof of this result is analogous to that of Theorem 6.1, and it employs Lemma E.2 and Lemma E.3. $\square$

Some observations are in order. First, since MP is a more general setting than RFE, then this lower bound is a lower bound for MP too. However, this is not guaranteed for ILfO. We observe that, while for RFE and IRL the bound is tight, for MP we cannot say so because we do not have the upper bound. Notice that, in case the expert state-only distribution $\overline{d}$ was unknown at exploration phase, and revealed afterwards, then the lower bound of Theorem 6.1 holds for ILfO too, because we might a posteriori reveal the state-only distribution $\overline{d}$ of the optimal policy, and thus, in such manner, ILfO would be reduced to RFE.

### E.2 Missing proofs

**Lemma E.2.** *Let IRL and RFE be the learning problems defined as in Appendix E.1. Then, for each problem, any $(\epsilon, \delta)$-PAC algorithm must collect at least the following number of exploration episodes:*

$$\tau \geqslant \Omega\Big( \frac{H^3 SA}{\epsilon^2} \log \frac{1}{\delta} \Big).$$

*Proof.* Observe that the proof for RFE is present in [12]. Thus, we have to prove just the result for IRL. For doing so, we will use both the results of [12] and [38]. Notice that for the sake of this proof we consider $\mathcal{R} = \{r\}$, that will reduce our problem to simple RL as, in order to compute the function $\overline{\mathcal{C}}_{p,\pi^E}(r)$, we just need to compute $J^*(r;p)$, being $J^{\pi^E}(r;p)$ known from the availability of $d^{p,\pi^E}$ and $r$.

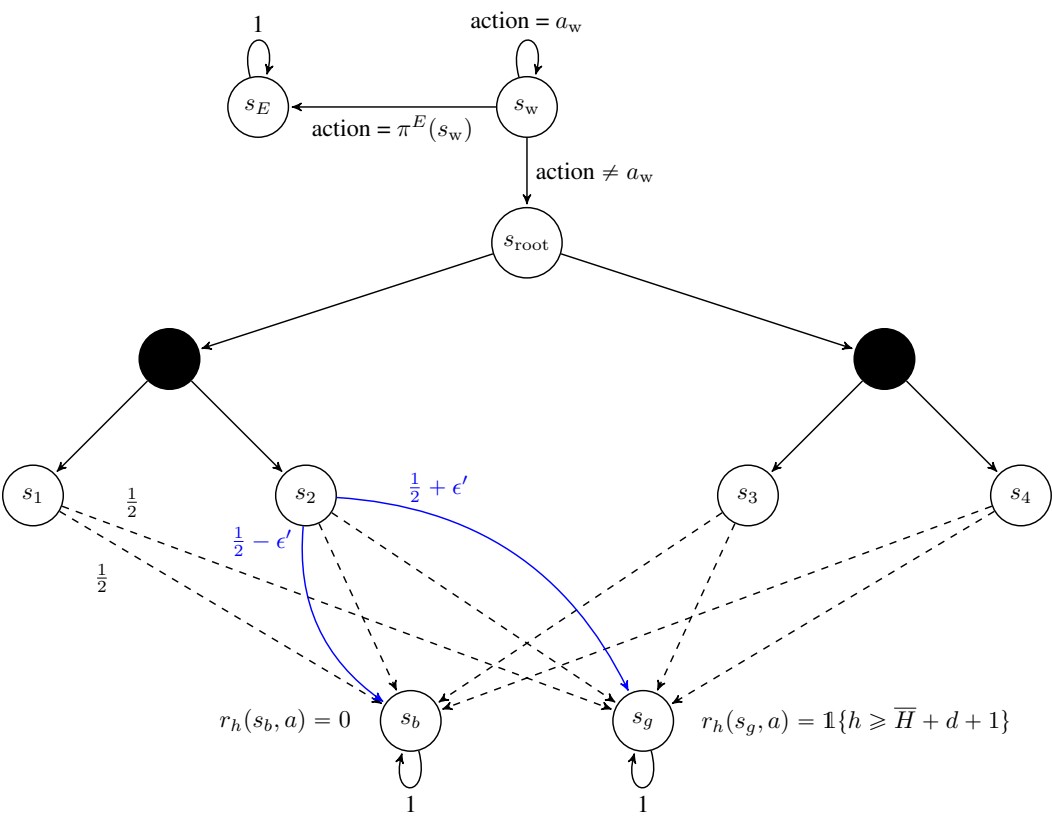

Figure 4: Hard instances.

**Instances Description** The hard instances considered are exactly the same as [12], and are reported in Figure 4 for simplicity. The only difference is the presence of state $s_E$, to which the expert's policy $\pi^E$ brings, which is absorbing. Such state is needed to make the knowledge of the expert's visit distribution $d^{p,\pi^E}$ useless at inferring information about the transition model in other parts of the state-action space. Based on [12], we describe such hard instances. Similarly to [12], we assume that $S \geqslant 7, A \geqslant 2$, and there exists an integer $d$ such that $S = 4 + (A^d - 1)/(A - 1)$, and we assume that $H \geqslant 3d$. Note that [12] show how to relax the assumption on the existence of $d$.

There are the initial state $s_w$, from which the agent starts, and states $s_g, s_b$, respectively, the "good" and "bad" states which are absorbing. Moreover, there is state $s_E$, which is reached by the expert, and is absorbing. The remaining $S - 4$ states are arranged in a full $A$-ary tree of depth $d - 1$ with root $s_{\text{root}}$. We denote by $\overline{H} \leqslant H - d$ a certain integer parameter, and by $\mathcal{L} := \{s_1, s_2, \ldots, s_L\}$ the set of leaves of the tree. We define $\mathcal{I} := \{1 + d, \ldots, \overline{H} + d\} \times \mathcal{L} \times \mathcal{A}$. For any $\imath \in \mathcal{I}$, we define and MDP $\mathcal{M}_\imath$ as follows. In any state of the tree, i.e., in states $\mathcal{S} \backslash \{s_w, s_g, s_b, s_E\}$, the transitions are deterministic, and the $a$-th action of a state brings to the $a$-th child of that node.

The transitions from $s_w$ are given by

$$p_h(s_w|s_w, a) := \mathbb{1}\{a = a_w, h \leqslant \overline{H}\} \quad \text{and} \quad p_h(s_{\text{root}}|s_w, a) := 1 - p_h(s_w|s_w, a).$$

In other words, action $a_\mathrm{w}$ allows the agent to remain in the initial state $s_\mathrm{w}$ up to stage $\overline{H}$. After stage $\overline{H}$, the agent is forced to leave $s_\mathrm{w}$ and to traverse the tree down to the leaves. Action $a_E = \pi_1^E(s_\mathrm{w})$ is the only action that brings to state $s_E$, which is absorbing. The transitions from any leaf $s_i \in \mathcal{L}$ are given, as in [12], by:

$$p_h(s_g|s_i, a) := \frac{1}{2} + \Delta_{(h*,\ell*,a*)}(h, s_i, a) \quad \text{and} \quad p_h(s_b|s_i, a) := \frac{1}{2} - \Delta_{(h*,\ell*,a*)}(h, s_i, a), \quad (4)$$

where $\Delta_{(h*,\ell*,a*)}(h, s_i, a) := \mathbb{1}\{(h, s_i, a) = (h^*, s_{\ell*}, a^*)\} \cdot \epsilon'$, for some $\epsilon' \in [0, 1/2]$. For this reason, there exists a (single) leaf $\ell^*$ where the agent can choose an action $a^*$ at stage $h^*$ to increase its probability of arriving to the good state $s_g$, which provides higher reward. We define states $s_g$ and $s_b$ to be absorbing, i.e., they satisfy $p_h(s_b|s_b, a) := p_h(s_g|s_g, a) := 1$ for any action $a$. The reward function is state-only and is defined as

$$\forall a \in \mathcal{A}, \quad r_h(s, a) := \mathbb{1}\{s = s_g, h \geqslant \overline{H} + d + 1\},$$

so that even though the agent decides to stay at $s_\mathrm{w}$ until stage $\overline{H}$, it does not lose any reward. Observe that state $s_E$ does not provide any reward, so that to estimate the (non)compatibility, any algorithm must provide a good estimate of the optimal performance.

Finally, we define a reference MDP $\mathcal{M}_0$ which is an MDP of the above type but for which $\Delta_0(h, s_i, a) := 0$ for all $(h, s_i, a)$. For certain $\epsilon'$ and $\overline{H}$ to choose, we define the class $\mathbb{M}$ to be the set $\mathbb{M} := \{\mathcal{M}_0\} \cup \{\mathcal{M}_\iota\}_{\iota \in \mathcal{I}}$.

**Distance between problems** We will prove the lower bound for instance $\mathcal{M}_0$. Observe that, in $\mathcal{M}_0$, the optimal utility is:

$$J_0^* = \frac{1}{2}(H - \overline{H} - d),$$

because there is no triple with additional bias towards $s_g$. Instead, for any other $\mathcal{M}_\iota \in \mathbb{M}$, the optimal utility is:

$$J_\iota^* = (H - \overline{H} - d)\left(\frac{1}{2} + \epsilon'\right).$$

Therefore, if we choose $\epsilon' := 2\epsilon/(H - \overline{H} - d)$, we have that, for any $\iota \in \mathcal{I}$:

$$\left| J_0^* - J_\iota^* \right| = 2\epsilon.$$

Thus, in particular, for any estimate $\widehat{J} \in \mathbb{R}$ we necessarily have $|J_0^* - \widehat{J}| \leqslant \epsilon \implies |J_\iota^* - \widehat{J}| > \epsilon$, and vice versa, i.e., we cannot provide an estimate $\widehat{J}$ that is $\epsilon$-close to both $J_0^*$ and $J_\iota^*$.

**Identifying the underlying problem** Following [38], let us consider a generic $(\epsilon, \delta)$-correct algorithm $\mathfrak{A}$ that outputs the estimated optimal utility $\widehat{J}$. Then, for all $\iota \in \mathcal{I}$, we have:

$$\delta \geqslant \sup_{\text{all problem instances } \mathcal{M}} \mathbb{P}_{\mathcal{M}, \mathfrak{A}}\left(\left| J_\mathcal{M}^* - \widehat{J} \right| \geqslant \epsilon\right)$$

$$\geqslant \sup_{\mathcal{M} \in \mathbb{M}} \mathbb{P}_{\mathcal{M}, \mathfrak{A}}\left(\left| J_\mathcal{M}^* - \widehat{J} \right| \geqslant \epsilon\right)$$

$$\geqslant \max_{\ell \in \{0, \iota\}} \mathbb{P}_{\mathcal{M}_\ell, \mathfrak{A}}\left(\left| J_\ell^* - \widehat{J} \right| \geqslant \epsilon\right).$$

For every $\iota \in \mathcal{I}$, we define the *identification function* $\Psi_\iota$ as the index of the problem "recognized" by algorithm $\mathfrak{A}$. In symbols:

$$\Psi_\iota := \arg\min_{\ell \in \{0, \iota\}} \left| J_\ell^* - \widehat{J} \right|.$$

In words, given estimate $\widehat{J}$ returned by algorithm $\mathfrak{A}$, the identification function $\Psi_\iota$ returns the problem between $\mathcal{M}_0$ and $\mathcal{M}_\iota$ whose optimal utility is closest to the estimate $\widehat{J}$. For what we have seen in the previous paragraph, problems $\mathcal{M}_0$ and $\mathcal{M}_\iota$ lie at a distance of at least $2\epsilon$ for all $\iota \in \mathcal{I}$. Therefore, for $\jmath \in \{0, \iota\}$, we have the following inclusion of events:

$$\{\Psi_\iota \neq \jmath\} \subseteq \{|J_\jmath^* - \widehat{J}| > \epsilon\}.$$

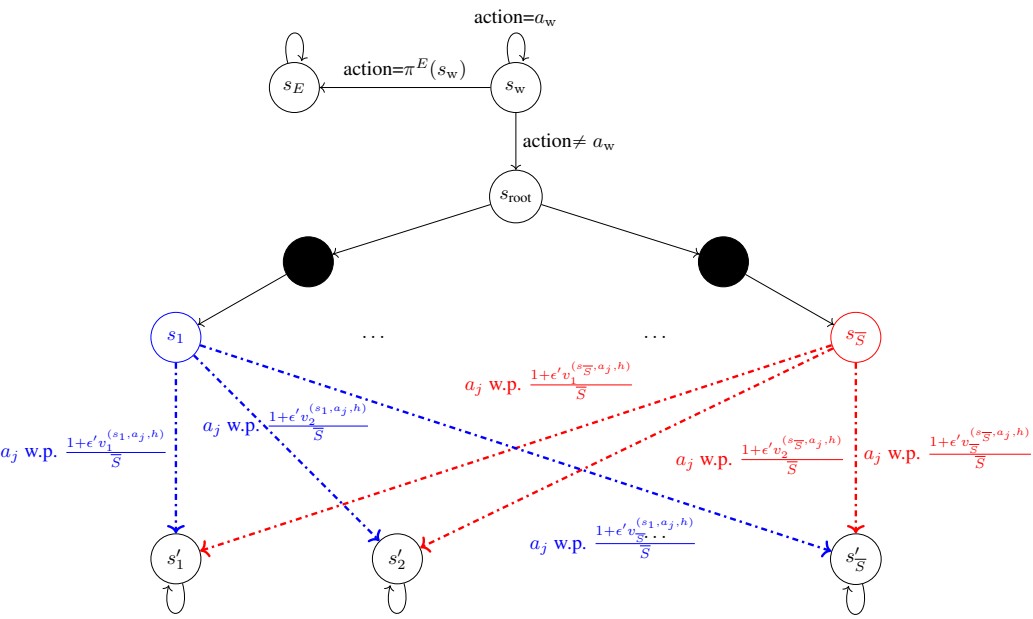

Figure 5: Hard instances.

Thanks to this fact, we can continue lower bounding the probability as:

$$
\begin{aligned}
\max_{\ell \in \{0, \imath\}} \mathbb{P}_{\mathcal{M}_\ell, \mathfrak{A}}\left(\left|J_\ell^* - \widehat{J}\right| \geqslant \epsilon\right) &\geqslant \max_{\ell \in \{0, \imath\}} \mathbb{P}_{\mathcal{M}_\ell, \mathfrak{A}}\{\Psi_\imath \neq \ell\} \\
&\overset{(1)}{\geqslant} \frac{1}{2}\left[\mathbb{P}_{\mathcal{M}_0, \mathfrak{A}}\left(\Psi_\imath \neq 0\right) + \mathbb{P}_{\mathcal{M}_\imath, \mathfrak{A}}\left(\Psi_\imath \neq \imath\right)\right] \\
&= \frac{1}{2}\left[\mathbb{P}_{\mathcal{M}_0, \mathfrak{A}}\left(\Psi_\imath \neq 0\right) + \mathbb{P}_{\mathcal{M}_\imath, \mathfrak{A}}\left(\Psi_\imath = 0\right)\right] \\
&\overset{(2)}{\geqslant} \frac{1}{4}\exp^{-\mathrm{KL}(\mathbb{P}_{\mathcal{M}_0, \mathfrak{A}}, \mathbb{P}_{\mathcal{M}_\imath, \mathfrak{A}})},
\end{aligned}
$$

where at (1) we have lower bounded the maximum with the average, i.e., $\max\{a, b\} \geqslant (a + b)/2$ for all $a, b \geqslant 0$, and at (2) we have applied the Bretagnolle-Huber's inequality [38].

**KL-divergence computation** The proof can be concluded by upper bounding the KL divergence $\mathrm{KL}(\mathbb{P}_{\mathcal{M}_0, \mathfrak{A}}, \mathbb{P}_{\mathcal{M}_\imath, \mathfrak{A}})$ as in the proof of Theorem 7 in [12], and then summing over all the $\Theta(SAH)$ instances to retrieve the result.

$\square$

**Lemma E.3.** *Let IRL and RFE be the learning problems defined as in Appendix E.1. For each problem, if the set of reward functions $\mathcal{R}$ in input is $\mathcal{R} = \mathfrak{R}$, then any $(\epsilon, \delta)$-PAC algorithm must collect at least the following number of exploration episodes:*

$$
\tau \geqslant \Omega\left(\frac{H^3 S^2 A}{\epsilon^2}\right).
$$

*Proof.* **Instances description** The hard instances that we use for the proof of this lemma are obtained by combining the hard instances in Lemma E.2 (i.e., the hard instances of [12]), with those in [38]. Specifically, this construction is based on the intuition described in [21] that, if we want to increase the sample complexity, we have to learn transitions also *to* $\Theta(S)$ states, and not just *from* $\Theta(S)$ states. Observe the presence of state $s_E$ (only for IRL), which plays the same role as in the proof of Lemma E.2. Any action in such state receives always reward $-1$, thus it is meaningless for the estimate of the (non)compatibility, which reduces to the estimation of the optimal performance. In this manner, the expert distribution $d^{p, \pi^E}$ does not provide additional information about the transition model of

other portion of the state-action space. Therefore, in the following, we will present the lower bound construction as if such state did not exist.

The hard instances are reported in Figure 5. Notice that they are exactly the same instances as those presented in the proof of Lemma E.2, with the difference that, from the $\overline{S}$ leaves (differently from earlier, we now denote the number of leaves through $\overline{S}$ instead of $L$), we do not reach just two states $s_g, s_b$, but we reach $\Theta(S)$ absorbing states, i.e., $s'_1, s'_2, \ldots, s'_{\overline{S}}$. The transitions from the leaves to such states is the same as in [38], and we report a description below.

Let us introduce the set $\overline{\mathcal{I}} := \{s_1, \ldots, s_{\overline{S}}\} \times \mathcal{A} \times \{1 + d, \ldots, \overline{H} + d\}$. Let $\overline{\imath} := (s_1, a_1, 1 + d) \in \overline{\mathcal{I}}$ be a specific triple of set $\mathcal{I}$, and denote $\mathcal{I} := \overline{\mathcal{I}} \backslash \{\overline{\imath}\}$. Let us also introduce set $\mathcal{V} := \{v \in \{-1, 1\}^{\overline{S}} : \sum_{j=1}^{\overline{S}} v_j = 0\}$. Thanks to Lemma E.6 of [38] (that we report in Lemma E.4 for simplicity), we know that there exists a subset $\overline{V} \subseteq \mathcal{V}$ (of transition models) with cardinality at least $2^{\overline{S}/5}$ such that, for every pair $v, w \in \overline{V}$ with $v \neq w$, we have that $\|v - w\|_1 \geqslant \overline{S}/16$. In other words, we know that there exists a $\overline{S}/16$-packing of $\mathcal{V}$ with cardinality at least $2^{\overline{S}/5}$.

Following [38], we denote by $\boldsymbol{v} = (v^\imath)_{\imath \in \mathcal{I}} \in \overline{V}^{\mathcal{I}}$ the generic vector of $\overline{V}^{\mathcal{I}}$. Now, for any $\boldsymbol{v} \in \overline{V}^{\mathcal{I}}$, for any triple $\overline{\jmath} \in \mathcal{I}$, and for some parameter $\epsilon' \in [0, 1/2]$ to choose, we construct problem instance $\mathcal{M}_{\boldsymbol{v}, \overline{\jmath}}$ as follows.

First of all, we define the transition model at triple $\overline{\imath}$ as:
$$p_{h_{\overline{\imath}}}(s'_i | s_{\overline{\imath}}, a_{\overline{\imath}}) = \frac{1}{\overline{S}} \quad \forall i \in [\![\overline{S}]\!],$$
where observe that we use notation $\imath = (s_\imath, a_\imath, h_\imath) \in \overline{\mathcal{I}}$ to denote triples in $\overline{\mathcal{I}}$. Instead, for the generic triple $\imath \in \mathcal{I}$ (including triple $\jmath$), the probability distribution of the next state is given by:
$$p_{h_\imath}(s'_i | s_\imath, a_\imath) = \frac{1}{\overline{S}} + \frac{\epsilon'}{\overline{S}} \boldsymbol{v}^\imath_i \quad \forall i \in [\![\overline{S}]\!],$$
where $\boldsymbol{v}^\imath_i$ represents the $i$-th component of the $\imath$-th vector in $\boldsymbol{v}$. In words, the $i$-th component of vector $\boldsymbol{v}^\imath \in \overline{V}$ creates a bias of $\epsilon'/\overline{S}$ towards the next state $s'_i$ for all $i \in [\![\overline{S}]\!]$. Since $\boldsymbol{v}^\imath \in \overline{V}$, then $p_{h_\imath}(\cdot | s_\imath, a_\imath) \in \Delta^{[\![\overline{S}]\!]}$ for all $\imath \in \mathcal{I}$.

We consider non-stationary reward functions. Specifically, all the rewards $r \in \mathfrak{R}$ that we consider assign reward 1 to both triples $\overline{\imath}$ and $\overline{\jmath}$, i.e., $r_{h_{\overline{\imath}}}(s_{\overline{\imath}}, a_{\overline{\imath}}) = 1$ and $r_{h_{\overline{\jmath}}}(s_{\overline{\jmath}}, a_{\overline{\jmath}}) = 1$. Next, for any other triple $(s, a, h) \in \mathcal{S} \times \mathcal{A} \times [\![H]\!]$ with state different from $s'_1, s'_2, \ldots, s'_{\overline{S}}$, we assign reward 0. For states $s'_1, s'_2, \ldots, s'_{\overline{S}}$, we consider state-only rewards whose value is always 0 in stages $[1, \overline{H} + d]$, and whose value is stationary and arbitrary afterwards. Intuitively, as in [12], forcing the reward to be 0 up $h = \overline{H} + d$ guarantees that we cannot obtain a higher expected return $J$ by reaching the leaves states earlier (i.e., by exiting from $s_{\mathrm{w}}$ before $\overline{H}$).

Given the definition above, we construct the class of instances $\mathbb{M} := \{\mathcal{M}_{\boldsymbol{v}, \imath} : \imath \in \mathcal{I}, \boldsymbol{v} \in \overline{V}^{\mathcal{I}}\}$. Moreover, we will use the notation $\mathcal{M}_{\boldsymbol{v} \overset{\imath}{\leftarrow} w, \jmath}$ to denote the instance in which we replace the $\imath$ component of $\boldsymbol{v}$, i.e., $\boldsymbol{v}^\imath$, with $w \in \mathcal{V}$ and $\mathcal{M}_{\boldsymbol{v} \overset{\imath}{\leftarrow} 0, \jmath}$ the instance in which we replace the $\imath$ component of $\boldsymbol{v}$, i.e., $\boldsymbol{v}^\imath$, with the zero vector. Since we will always use this notation when substituting triple $\jmath$, i.e., we always use this notation in situations as $\mathcal{M}_{\boldsymbol{v} \overset{\jmath}{\leftarrow} w, \jmath}$, then we omit the second parameter, and write just $\mathcal{M}_{\boldsymbol{v} \overset{\imath}{\leftarrow} w} := \mathcal{M}_{\boldsymbol{v} \overset{\imath}{\leftarrow} w, \jmath}$.

**Distance between problems** Consider an arbitrary problem instance $\mathcal{M}_{\boldsymbol{v}, \imath} \in \mathbb{M}$, for certain $\imath \in \mathcal{I}$ and $\boldsymbol{v} \in \overline{V}^{\mathcal{I}}$. Let $r \in \mathfrak{R}$ be an arbitrary reward function that satisfies the constraints described earlier. Let $\pi_{\overline{\imath}} \in \Pi$ be the deterministic policy that brings to triple $\overline{\imath}$. Then, its expected return is:
$$J^{\pi_{\overline{\imath}}}(r; \mathcal{M}_{\boldsymbol{v}, \imath}) = 1 + \frac{H - \overline{H} - d}{\overline{S}} \sum_{i=1}^{\overline{S}} r_i,$$

where $r_i := r_{\overline{H}+d+1}(s_i')$ for all $i \in [\![\overline{S}]\!]$. Let policy $\pi_i \in \Pi$ be the deterministic policy that brings to triple $\imath$. Then, its expected return is:

$$J^{\pi_i}(r; \mathcal{M}_{\boldsymbol{v},\imath}) = 1 + \frac{H - \overline{H} - d}{\overline{S}} \sum_{i=1}^{\overline{S}} r_i + \epsilon' \frac{(H - \overline{H} - d)}{\overline{S}} \sum_{i=1}^{\overline{S}} \boldsymbol{v}_i^\imath r_i.$$

Finally, let policy $\pi_\jmath \in \Pi$ be the deterministic policy that brings to any other triple $\jmath \in \mathcal{I}\backslash\{\imath\}$. Then, its expected return is:

$$J^{\pi_\jmath}(r; \mathcal{M}_{\boldsymbol{v},\imath}) = 0 + \frac{H - \overline{H} - d}{\overline{S}} \sum_{i=1}^{\overline{S}} r_i + \epsilon' \frac{(H - \overline{H} - d)}{\overline{S}} \sum_{i=1}^{\overline{S}} \boldsymbol{v}_i^\jmath r_i.$$

It should be remarked that $(v, r) = \sum_{i \in [\![\overline{S}]\!]} v_i r_i \in [-\overline{S}, \overline{S}]$ for any $r \in \mathfrak{R}$ and $v \in \overline{\mathcal{V}}$, therefore, as long as:

$$\epsilon'(H - \overline{H} - d) < 1 - \epsilon'(H - \overline{H} - d) - \epsilon \iff \epsilon' < \frac{1 - \epsilon}{2(H - \overline{H} - d)}, \tag{5}$$

then any policy $\pi_\jmath$ is cannot be $\epsilon$-optimal in problem $\mathcal{M}_{\boldsymbol{v},\imath}$, in which, thus, the optimal policy shall be searched for between $\pi_{\overline{\imath}}$ and $\pi_\imath$.

Now, consider an arbitrary pair $v, w \in \overline{\mathcal{V}}$ such that $v \neq w$, and an arbitrary triple $\imath \in \mathcal{I}$ and vector $\in \overline{\mathcal{V}}^{\mathcal{I}}$. We now compare problem instances $\mathcal{M}_{\boldsymbol{v} \overset{\imath}{\leftarrow} v}$ and $\mathcal{M}_{\boldsymbol{v} \overset{\imath}{\leftarrow} w}$. Among all possible reward functions that satisfy the definition provided in the construction of the hard instances, we find reward $r'$ such that, in every component $i \in [\![\overline{S}]\!]$, satisfies:

$$r_i' = \begin{cases} +1 & \text{if } v_i = +1 \wedge w_i = -1 \\ -1 & \text{if } v_i = -1 \wedge w_i = +1 \\ 0 & \text{if } v_i = w_i \end{cases}.$$

For what we have seen before about class $\overline{\mathcal{V}}$, we know that $\|v - w\|_1 = \sum_{i \in [\![\overline{S}]\!]} |v_i - w_i| \geqslant \overline{S}/16$, thus, since $v, w \in \mathcal{V}$, i.e., their components belong to $\{-1, +1\}$, we know that there are at least $\overline{S}/32$ components of $v, w$ that differ from each other. By using reward $r'$, we have that:

$$\sum_{i=1}^{\overline{S}} v_i r_i' \geqslant \frac{\overline{S}}{32} \geqslant 0,$$

$$\sum_{i=1}^{\overline{S}} w_i r_i' \leqslant -\frac{\overline{S}}{32} \leqslant 0.$$

As a consequence, the expected returns of policies $\pi_{\overline{\imath}}$ and $\pi_\imath$ in problems $\mathcal{M}_{\boldsymbol{v} \overset{\imath}{\leftarrow} v}$ and $\mathcal{M}_{\boldsymbol{v} \overset{\imath}{\leftarrow} w}$ are:

$$J^{\pi_{\overline{\imath}}}(r'; \mathcal{M}_{\boldsymbol{v} \overset{\imath}{\leftarrow} v}) = J^{\pi_{\overline{\imath}}}(r'; \mathcal{M}_{\boldsymbol{v} \overset{\imath}{\leftarrow} w}) = 1 + \frac{H - \overline{H} - d}{\overline{S}} \sum_{i=1}^{\overline{S}} r_i',$$

$$J^{\pi_\imath}(r'; \mathcal{M}_{\boldsymbol{v} \overset{\imath}{\leftarrow} v}) \geqslant 1 + \frac{H - \overline{H} - d}{\overline{S}} \sum_{i=1}^{\overline{S}} r_i' + \epsilon' \frac{(H - \overline{H} - d)}{32},$$

$$J^{\pi_\imath}(r'; \mathcal{M}_{\boldsymbol{v} \overset{\imath}{\leftarrow} w}) \leqslant 1 + \frac{H - \overline{H} - d}{\overline{S}} \sum_{i=1}^{\overline{S}} r_i' - \epsilon' \frac{(H - \overline{H} - d)}{32},$$

from which we infer that:

$$J^{\pi_\imath}(r'; \mathcal{M}_{\boldsymbol{v} \overset{\imath}{\leftarrow} v}) \geqslant J^{\pi_{\overline{\imath}}}(r'; \mathcal{M}_{\boldsymbol{v} \overset{\imath}{\leftarrow} v}) = J^{\pi_{\overline{\imath}}}(r'; \mathcal{M}_{\boldsymbol{v} \overset{\imath}{\leftarrow} w}) \geqslant J^{\pi_\imath}(r'; \mathcal{M}_{\boldsymbol{v} \overset{\imath}{\leftarrow} w}).$$

Now, let us choose $\epsilon' > 64\epsilon/(H - \overline{H} - d)$. To satisfy also the constraint in Equation (5), we can roughly assume $\epsilon < 1/256$ and set $\epsilon' = 65\epsilon/(H - \overline{H} - d)$. Thanks to this choice, observe that:

$$J^{\pi_\imath}(r'; \mathcal{M}_{\boldsymbol{v} \overset{\imath}{\leftarrow} v}) > J^{\pi_{\overline{\imath}}}(r'; \mathcal{M}_{\boldsymbol{v} \overset{\imath}{\leftarrow} v}) + 2\epsilon,$$

$$J^{\pi_{\overline{\imath}}}(r'; \mathcal{M}_{\boldsymbol{v} \overset{\imath}{\leftarrow} w}) > J^{\pi_\imath}(r'; \mathcal{M}_{\boldsymbol{v} \overset{\imath}{\leftarrow} w}) + 2\epsilon.$$

In words, policy $\pi_\imath$ is optimal in problem $\mathcal{M}_{\boldsymbol{v}\xleftarrow{\imath}v}$, and policy $\pi_{\overline{\imath}}$ is worse than $2\epsilon$-suboptimal in such problem. In addition, observe that policy $\pi_{\overline{\imath}}$ is optimal in problem $\mathcal{M}_{\boldsymbol{v}\xleftarrow{\imath}w}$, and policy $\pi_\imath$ is worse than $2\epsilon$-suboptimal in such problem. We stress that any stochastic policy in-between $\pi_\imath$ and $\pi_{\overline{\imath}}$ cannot be $\epsilon$-optimal for both problems.

To sum up, for the choice of $\epsilon'$ made earlier, for arbitrary pairs of problems $\mathcal{M}_{\boldsymbol{v}\xleftarrow{\imath}v}$ and $\mathcal{M}_{\boldsymbol{v}\xleftarrow{\imath}w}$, we have seen that there exist rewards in $\mathfrak{R}$ for which a policy $\epsilon$-optimal for problem $\mathcal{M}_{\imath,v}$ is not $\epsilon$-optimal for problem $\mathcal{M}_{\imath,w}$, and vice versa.

**Identifying the underlying problem: RFE.** We consider first RFE, and then IRL.

Let us consider an $(\epsilon,\delta)$-correct algorithm $\mathfrak{A}$ for RFE, that outputs, for any reward function $r \in \mathfrak{R}$, a policy $\widehat{\pi}_r$. For simplicity, we consider as output of Algorithm $\mathfrak{A}$ a function $\widehat{\pi} : \mathfrak{R} \to \Pi$, that takes in input a reward and outputs a policy.

For any $\imath \in \mathcal{I}$ and $\boldsymbol{v} \in \overline{\mathcal{V}}^{\mathcal{I}}$, we can lower bound the error probability as:

$$
\begin{aligned}
\delta &\geqslant \sup_{\substack{\text{all problem instances } \mathcal{M}}} \mathbb{P}_{\mathcal{M},\mathfrak{A}}\left( \sup_{r\in\mathfrak{R}} J_{\mathcal{M}}^*(r) - J_{\mathcal{M}}^{\widehat{\pi}_r}(r) \geqslant \epsilon \right) \\
&\overset{(1)}{\geqslant} \sup_{\mathcal{M}\in\mathbb{M}} \mathbb{P}_{\mathcal{M},\mathfrak{A}}\left( \sup_{r\in\mathfrak{R}} J_{\mathcal{M}}^*(r) - J_{\mathcal{M}}^{\widehat{\pi}_r}(r) \geqslant \epsilon \right) \\
&\overset{(2)}{\geqslant} \max_{w\in\overline{\mathcal{V}}} \mathbb{P}_{\mathcal{M}_{\boldsymbol{v}\xleftarrow{\imath}w},\mathfrak{A}}\left( \sup_{r\in\mathfrak{R}} J_{\mathcal{M}_{\boldsymbol{v}\xleftarrow{\imath}w}}^*(r) - J_{\mathcal{M}_{\boldsymbol{v}\xleftarrow{\imath}w}}^{\widehat{\pi}_r}(r) \geqslant \epsilon \right),
\end{aligned}
$$

where at (1) we have lower bounded by replacing all possible RFE problem instances with problem instances in $\mathbb{M}$, and at (2) we have lower bounded by replacing all instances in $\mathbb{M}$ with just instances $\{\mathcal{M}_{\boldsymbol{v}\xleftarrow{\imath}w} : w \in \overline{\mathcal{V}}\}$ for the fixed triple $\imath$ and vector $\boldsymbol{v}$.

For every $\imath \in \mathcal{I}$ and $\boldsymbol{v} \in \overline{\mathcal{V}}^{\mathcal{I}}$, we define the *identification function* $\Psi_{\imath,\boldsymbol{v}}$ as the index of the problem $w \in \overline{\mathcal{V}}$ "recognized" by algorithm $\mathfrak{A}$. In symbols:

$$
\Psi_{\imath,\boldsymbol{v}} := \arg\min_{w\in\overline{\mathcal{V}}} \sup_{r\in\mathfrak{R}} J_{\mathcal{M}_{\boldsymbol{v}\xleftarrow{\imath}w}}^*(r) - J_{\mathcal{M}_{\boldsymbol{v}\xleftarrow{\imath}w}}^{\widehat{\pi}_r}(r).
$$

In words, given estimate $\widehat{\pi} : \mathfrak{R} \to \Pi$ returned by algorithm $\mathfrak{A}$, the identification function $\Psi_{\imath,\boldsymbol{v}}$ returns the problem in $\{\mathcal{M}_{\boldsymbol{v}\xleftarrow{\imath}w} : w \in \overline{\mathcal{V}}\}$ whose solution $\pi : \mathfrak{R} \to \Pi$ is closest to the estimate $\widehat{\pi}$. For what we have seen in the previous paragraph, for any $v, w \in \overline{\mathcal{V}}$ with $v \neq w$, for any fixed $\imath \in \mathcal{I}$ and $\boldsymbol{v} \in \overline{\mathcal{V}}^{\mathcal{I}}$, there exists a reward function $r' \in \mathfrak{R}$ such that no policy can have expected utility $\epsilon$-close to the optimal expected utility of both problems $\mathcal{M}_{\boldsymbol{v}\xleftarrow{\imath}v}$ and $\mathcal{M}_{\boldsymbol{v}\xleftarrow{\imath}w}$. Therefore, for $w \in \overline{\mathcal{V}}$, we have the following inclusion of events:

$$
\{\Psi_{\imath,\boldsymbol{v}} \neq w\} \subseteq \left\{ \sup_{r\in\mathfrak{R}} J_{\mathcal{M}_{\boldsymbol{v}\xleftarrow{\imath}w}}^*(r) - J_{\mathcal{M}_{\boldsymbol{v}\xleftarrow{\imath}w}}^{\widehat{\pi}_r}(r) > \epsilon \right\}.
$$

We can continue to lower bound the probability as:

$$
\begin{aligned}
\max_{w\in\overline{\mathcal{V}}} \mathbb{P}_{\mathcal{M}_{\boldsymbol{v}\xleftarrow{\imath}w},\mathfrak{A}}\left( \sup_{r\in\mathfrak{R}} J_{\mathcal{M}_{\boldsymbol{v}\xleftarrow{\imath}w}}^*(r) - J_{\mathcal{M}_{\boldsymbol{v}\xleftarrow{\imath}w}}^{\widehat{\pi}_r}(r) \geqslant \epsilon \right) &\overset{(3)}{\geqslant} \frac{1}{|\overline{\mathcal{V}}|} \sum_{w\in\overline{\mathcal{V}}} \mathbb{P}_{\mathcal{M}_{\boldsymbol{v}\xleftarrow{\imath}w},\mathfrak{A}}\left( \Psi_{\imath,\boldsymbol{v}} \neq w \right) \\
&\overset{(4)}{\geqslant} 1 - \frac{1}{\log|\overline{\mathcal{V}}|}\left( \frac{1}{|\overline{\mathcal{V}}|} \sum_{w\in\overline{\mathcal{V}}} \mathrm{KL}(\mathbb{P}_{\mathcal{M}_{\boldsymbol{v}\xleftarrow{\imath}w},\mathfrak{A}}, \mathbb{P}_{\mathcal{M}_{\boldsymbol{v}\xleftarrow{\imath}0},\mathfrak{A}}) - \log 2 \right),
\end{aligned}
$$

where at (3) we have lower bounded the maximum over $\overline{\mathcal{V}}$ with the average, and at (4) we have applied, similarly to [38], the Fano's inequality, reported in Theorem E.5 for simplicity.

**Identifying the underlying problem: IRL.** For IRL, it is possible to carry out a similar derivation. However, we remark that, now, the error is measured based on the expected utilities, and not on the policies.

Let us consider an $(\epsilon,\delta)$-correct algorithm $\mathfrak{A}$ for IRL, that outputs, for any reward function $r \in \mathfrak{R}$, a utility $\widehat{J}_r$. For simplicity, we consider as output of Algorithm $\mathfrak{A}$ a function $\widehat{J} : \mathfrak{R} \to \mathbb{R}$, that takes in input a reward and outputs a utility.

For any $\imath \in \mathcal{I}$ and $\boldsymbol{v} \in \overline{\mathcal{V}}^{\mathcal{I}}$, we can lower bound the error probability as:

$$\delta \geqslant \sup_{\text{all problem instances } \mathcal{M}} \mathbb{P}_{\mathcal{M},\mathfrak{A}}\left(\sup_{r \in \mathfrak{R}}\left|J_{\mathcal{M}}^*(r) - \widehat{J}_r\right| \geqslant \epsilon\right)$$

$$\geqslant \sup_{\mathcal{M} \in \mathbb{M}} \mathbb{P}_{\mathcal{M},\mathfrak{A}}\left(\sup_{r \in \mathfrak{R}}\left|J_{\mathcal{M}}^*(r) - \widehat{J}_r\right| \geqslant \epsilon\right)$$

$$\geqslant \max_{w \in \overline{\mathcal{V}}} \mathbb{P}_{\mathcal{M}_{\boldsymbol{v} \xleftarrow{\imath} w},\mathfrak{A}}\left(\sup_{r \in \mathfrak{R}}\left|J_{\mathcal{M}_{\boldsymbol{v} \xleftarrow{\imath} w}}^*(r) - \widehat{J}_r\right| \geqslant \epsilon\right).$$

For any $\imath \in \mathcal{I}$ and $\boldsymbol{v} \in \overline{\mathcal{V}}^{\mathcal{I}}$, we define an identification function $\Psi_{\imath,\boldsymbol{v}}$ as:

$$\Psi_{\imath,\boldsymbol{v}} := \arg\min_{w \in \overline{\mathcal{V}}} \sup_{r \in \mathfrak{R}}\left|J_{\mathcal{M}_{\boldsymbol{v} \xleftarrow{\imath} w}}^*(r) - \widehat{J}_r\right|,$$

and by a reasoning analogous to that for RFE, we can continue to lower bounding as:

$$\max_{w \in \overline{\mathcal{V}}} \mathbb{P}_{\mathcal{M}_{\boldsymbol{v} \xleftarrow{\imath} w},\mathfrak{A}}\left(\sup_{r \in \mathfrak{R}}\left|J_{\mathcal{M}_{\boldsymbol{v} \xleftarrow{\imath} w}}^*(r) - \widehat{J}_r\right| \geqslant \epsilon\right) \geqslant \frac{1}{|\overline{\mathcal{V}}|} \sum_{w \in \overline{\mathcal{V}}} \mathbb{P}_{\mathcal{M}_{\boldsymbol{v} \xleftarrow{\imath} w},\mathfrak{A}}\left(\Psi_{\imath,\boldsymbol{v}} \neq w\right)$$

$$\geqslant 1 - \frac{1}{\log|\overline{\mathcal{V}}|}\left(\frac{1}{|\overline{\mathcal{V}}|} \sum_{w \in \overline{\mathcal{V}}} \mathrm{KL}(\mathbb{P}_{\mathcal{M}_{\boldsymbol{v} \xleftarrow{\imath} w},\mathfrak{A}}, \mathbb{P}_{\mathcal{M}_{\boldsymbol{v} \xleftarrow{\imath} 0},\mathfrak{A}}) - \log 2\right), \tag{6}$$

which represents the same lower bound obtained also for RFE.

**KL-divergence computation** The following derivation is analogous to that of [38]. To bound the KL-divergence term, for any $\imath \in \mathcal{I}$, we can write:

$$\mathrm{KL}(\mathbb{P}_{\mathcal{M}_{\boldsymbol{v} \xleftarrow{\imath} w},\mathfrak{A}}, \mathbb{P}_{\mathcal{M}_{\boldsymbol{v} \xleftarrow{\imath} 0},\mathfrak{A}}) \stackrel{(1)}{=} \mathop{\mathbb{E}}_{\mathcal{M}_{\boldsymbol{v} \xleftarrow{\imath} w},\mathfrak{A}}\left[N_{h_\imath}^\tau(s_\imath, a_\imath)\right]\mathrm{KL}(p_{h_\imath}^{\mathcal{M}_{\boldsymbol{v} \xleftarrow{\imath} w}}(\cdot|s_\imath, a_\imath), p_{h_\imath}^{\mathcal{M}_{\boldsymbol{v} \xleftarrow{\imath} 0}}(\cdot|s_\imath, a_\imath))$$

$$\stackrel{(2)}{\leqslant} 2(\epsilon')^2 \mathop{\mathbb{E}}_{\mathcal{M}_{\boldsymbol{v} \xleftarrow{\imath} w},\mathfrak{A}}\left[N_{h_\imath}^\tau(s_\imath, a_\imath)\right],$$

where at (1) we have applied Lemma E.7, and at (2) we have applied Lemma E.6 (having observed that the transition models differ in $\imath$ and defined $N_{h_\imath}^\tau(s_\imath, a_\imath) = \sum_{t=1}^\tau \mathbb{1}\{(s_t, a_t, h_t) = (s_\imath, a_\imath, h_\imath)\}$).

Plugging into Equation (6), we get:

$$\delta \geqslant \frac{1}{|\overline{\mathcal{V}}|} \sum_{w \in \overline{\mathcal{V}}} \mathbb{P}_{\mathcal{M}_{\boldsymbol{v} \xleftarrow{\imath} w},\mathfrak{A}}\left(\Psi_{\imath,\boldsymbol{v}} \neq w\right) \implies \frac{1}{|\overline{\mathcal{V}}|} \sum_{w \in \overline{\mathcal{V}}} \mathop{\mathbb{E}}_{\mathcal{M}_{\boldsymbol{v} \xleftarrow{\imath} w},\mathfrak{A}}\left[N_{h_\imath}^\tau(s_\imath, a_\imath)\right] \geqslant \frac{(1-\delta)\log|\overline{\mathcal{V}}| - \log 2}{2(\epsilon')^2}.$$

Notice that, since $|\overline{\mathcal{V}}| = \Theta(e^S)$ and $\epsilon' = \Theta(\epsilon/H)$, then this bound is in the order of $\Omega(\frac{H^2 S}{\epsilon^2})$. To get the additional $\Omega(SAH)$ dependence, we can make the same observation as in [38], i.e., that ince the derivation is carried out for every $\imath \in \mathcal{I}$ and $\boldsymbol{v} \in \overline{\mathcal{V}}^{\mathcal{I}}$, we can perform the summation over $\imath$ and the average over $\boldsymbol{v}$. By noticing that we get a guarantee on a mean under the uniform distribution of the instances of the sample complexity, we realize that there must exist one $\boldsymbol{v}^{\text{hard}} \in \overline{\mathcal{V}}$ for which it holds the desired $\Omega\left(\frac{H^3 S^2 A}{\epsilon^2}\right)$ dependency.

$$\square$$

### E.2.1 Technical Tools

We report here some results from other works. The notation adopted is the same as the original works.

**Lemma E.4** (Lemma E.6 of [38])**.** *Let $\mathcal{V} = \{v \in \{-1, 1\}^D : \sum_{j=1}^D v_j = 0\}$. Then, the $\frac{D}{16}$-packing number of $\mathcal{V}$ w.r.t. the metric $d(v, v') = \sum_{j=1}^D |v_j - v'_j|$ is lower bounded by $2^{\frac{D}{5}}$.*

**Theorem E.5.** *(Theorem E.2 of [38]) Let $\mathbb{P}_0, \mathbb{P}_1, \dots, \mathbb{P}_M$ be probability measures on the same measurable space $(\Omega, \mathcal{F})$, and let $\mathcal{A}_1, \dots, \mathcal{A}_M \in \mathcal{F}$ be a partition of $\Omega$. Then,*

$$\frac{1}{M} \sum_{i=1}^M \mathbb{P}_i(\mathcal{A}_i^c) \geqslant 1 - \frac{\frac{1}{M}\sum_{i=1}^M D_{KL}(\mathbb{P}_i, \mathbb{P}_0) - \log 2}{\log M},$$

where $\mathcal{A}^c = \Omega \backslash \mathcal{A}$ is the complement of $\mathcal{A}$.

**Lemma E.6** (Lemma E.4 of [38]). *Let $\epsilon \in [0, 1/2]$ and $\mathbf{v} \in \{-\epsilon, \epsilon\}^D$ such that $\sum_{i=1}^d v_i = 0$. Consider the two categorical distributions $\mathbb{P} = \left(\frac{1}{D}, \frac{1}{D}, \ldots, \frac{1}{D}\right)$ and $\mathbb{P} = \left(\frac{1+v_1}{D}, \frac{1+v_2}{D}, \ldots, \frac{1+v_D}{D}\right)$. Then, it holds that:*

$$D_{KL}(\mathbb{P}, \mathbb{Q}) \leqslant 2\epsilon^2 \qquad and \qquad D_{KL}(\mathbb{Q}, \mathbb{P}) \leqslant 2\epsilon^2.$$

**Lemma E.7** (Lemma 5 of [12]). *Let $\mathcal{M}$ and $\mathcal{M}'$ be two MDPs that are identical except for their transition probabilities, denoted by $p_h$ and $p'_h$, respectively. Assume that we have $\forall (sa)$, $p_h(\cdot|s, a) \ll p'_h(\cdot|s, a)$. Then, for any stopping time $\tau$ with respect to $(\mathcal{F}_H^t)_{t \geqslant 1}$ that satisfies $\mathbb{P}_{\mathcal{M}} \tau < \infty = 1$,*

$$KL\left(\mathcal{P}_{\mathcal{M}}^{I_H^\tau}, \mathcal{P}_{\mathcal{M}'}^{I_H^\tau}\right) = \sum_{s \in \mathcal{S}} \sum_{a \in \mathcal{A}} \sum_{h \in \llbracket H-1 \rrbracket} \underset{\mathcal{M}}{\mathbb{E}} \left[N_{h,s,a}^\tau\right] KL\left(p_h(\cdot|s, a), p'_h(\cdot|s, a)\right),$$

*where $N_{h,s,a}^\tau := \sum_{t=1}^\tau \mathbb{1}\{(S_h^t, A_h^t) = (s, a)\}$ and $I_H^\tau : \Omega \to \bigcup_{t \geqslant 1} \mathcal{I}_H^t : \omega \mapsto I_H^{\tau(\omega)}(\omega)$ is the random vector representing the history up to episode $\tau$.*

# F  A Use Case for Objective-Free Exploration (OFE)

Consider the following setting. You are given a certain MDP without reward $\mathcal{M} = (\mathcal{S}, \mathcal{A}, H, d_0, p)$, in which you do not know neither $d_0$ nor $p$. Your job is to explore the environment to collect samples that allow you to construct estimates $\widehat{d}_0 \approx d_0$ and $\widehat{p} \approx p$, that will be subsequently used to perform a task in a given class $\mathscr{F}$ in an $(\epsilon, \delta)$-correct manner. Of course the number of samples should be as small as possible. How do you explore? It depends on which problems are contained in class $\mathscr{F}$.

A use case for OFE is the following.

**Example F.1.** *Assume that we are given a single fixed environment (for instance, a warehouse), in which there are many tasks to do (e.g., labelling objects, putting stuff on the shelves, bringing products from one side to the other), and assume (it is reasonable) that it is desirable to have one robot for each task. To teach these robots how to behave, we decide to use RL. Since all the robots work in the same environment (warehouse), then the (unknown) transition model is the same. For this reason, an efficient exploration (potentially through RFE) is meaningful. However, we realize that some tasks are difficult to design (i.e., the rewards of such tasks). For these tasks, we prefer to use a human expert to exhibit demonstrations, and then use ReL (in particular, IRL), to learn the reward, that will be subsequently used for AL. To perform IRL nicely, the samples collected at the beginning shall be used. To sum up, we might be interested in performing multiple RL and IRL tasks in the same unknown MDP, and, for efficiency reasons, our exploration of the environment has to be performed only once (before) being given the tasks to solve.*

