# OpenReview forum: "How does Inverse RL Scale to Large State Spaces? A Provably Efficient Approach"
_NeurIPS.cc/2024/Conference — NeurIPS 2024 poster_

### Official Review · Reviewer_x3io · 2024-07-10

**Soundness:** 4
**Presentation:** 2
**Contribution:** 3
**Rating:** 7
**Confidence:** 3

**Summary:**

The paper studies IRL in linear MDPs. The authors first demonstrate that the feasible reward set cannot be efficiently learned in large state and action spaces. To address this challenge, they propose a new IRL framework called reward compatibility, where the goal is to learn a classifier that determines whether the expert is approximately optimal for a given reward. Using CATY-IRL, they also introduce a new sample-efficient algorithm for this task, which first explores the MDP via reward-free exploration (RFE) and then evaluates whether a given reward is compatible with the expert's demonstrations. Furthermore, for the tabular setting, they provide a tight minimax lower bound and show that a similar bound also improves on existing lower bounds for RFE. Lastly, the authors propose a novel problem setting of objective-free exploration, which generalizes RFE to arbitrary tasks of interest.

**Strengths:**

The paper provides a good and original contribution to the challenging problem of IRL in large state-action spaces. In my opinion, its strengths are:
1. Technical quality: The authors use elegant notation and carefully introduce all symbols. Moreover, clear proofs for all results are provided in the appendix.
2. Sample complexity bounds: The sample complexity of CATY-IRL is thoroughly analyzed for tabular MDPs, tabular MDPs with linear rewards, and linear MDPs. Additionally, a tight minimax lower bound for the tabular setting is given. This lower bound is particularly appreciated, as lower bounds are still rare in the IRL literature.
3. Originality: After proving that identifying the feasible reward set is intractable, the authors introduce the novel problem setting of checking for reward compatibility and provide a sample efficient algorithm for it.

**Weaknesses:**

1. Motivation: While original, the motivation for learning a reward compatibility classifier is not entirely clear to me. I would expect the authors to better explain its usefulness and potential applications.
2. Writing: The paper is not easy to understand on the first read. There are many propositions and theorems, but more emphasis on motivation and intuition building would be beneficial. For instance, a simple example for Proposition 3.1 could help build geometric intuition.
3. Related work: The authors introduce reward non-compatibility as a novel metric for IRL. However, minimizing the suboptimality of the expert is a core idea behind many imitation learning and IRL algorithms, such as GAIL and others. Of course, in the unregularized setting minimizing the suboptimality would lead to trivial solutions. However, I would expect a discussion about similarities and differences to min-max IL/IRL in the main part of the paper.
4. Computational complexity: The authors claim that their algorithm is also computationally efficient, but I couldn't find a discussion about this.
5. This is a minor point, but in line 103 you use but don't introduce the notation $Y^X$ for the set of functions mapping from $X$ to $Y$. Moreover, slightly inconsistently the set of functions from $X$ to $\Delta^Y$ is denoted as $\Delta_X^Y$. Why don't you change the notation to $\Delta_Y^X$?

**Questions:**

How would you apply your algorithm to a real world IRL problem? What is the computational complexity of the classification step?

**Limitations:**

I think the practical limitations should be discussed more extensively.

---

> ### Author Rebuttal · Authors · 2024-08-03
>
> We thank the Reviewer for praising our analysis as solid and novel, and for noting the significance of the proposed lower bounds. We answer the Reviewer questions and comments below.
> ## Weaknesses
> 1- We divide the answer in two parts. First, we explain why learning a reward compatibility classifier is *useful* from a technical perspective. Next, we describe some *applications*.
>
> Our **ultimate goal** is *to understand how many samples are necessary for inferring as much information as possible about the expert's reward function $r^E$ from demonstrations*. Since the problem is *ill-posed* (underconstrained) [1,2], i.e., $r^E$ is only partially identifiable from demonstrations [3], we resort to understanding how many samples are needed to infer the constraints characterizing $r^E$, i.e., the feasible set (see Definition 3.1). However, as we demonstrate in Section 3, inferring the constraints, i.e., the feasible set, in Linear MDPs cannot be done either in a sample efficient manner (because of the lower bound in Theorem 3.2), or in a computational efficient manner (because it contains a continuum of functions, see lines 184-186). **For these reasons**, instead of learning the feasible set (i.e., the set of rewards with $0$ (non)compatibility), **we propose** to learn the *set* of rewards with $\Delta$ (non)compatibility (for some $\Delta>0$), and we demonstrate that this task is sample efficient (same upper bound as CATY-IRL, see Theorem 5.1) but not computationally efficient, because, again, the set contains a continuum of functions. Thus, we propose to learn a *classifier*, which can be seen as a *trick* for the practical computation of a set of infinite items.
>
> Beyond offering a *significant characterization* of the intrinsic complexity of the IRL problem (see Theorem 5.1 and Theorem 6.1), an algorithm (e.g., CATY-IRL) for learning a reward compatibility classifier **can be applied to most common IRL applications**, **without suffering from the bias induced by some *heuristics*** (e.g., margin maximization [1]) **or *additional assumptions*** (e.g., entropy maximization [4]). For instance, $(i)$ in the context of **designing rewards** for RL agents [5], CATY-IRL permits to combine domain knowledge and expert demonstrations by calculating the degree to which some human-designed rewards are compatible with the given demonstrations; $(ii)$ given some rewards candidate at **modelling the preferences** of the observed agent, obtained for example through human design or some IRL algorithms [1,4], with purposes like imitating or predicting  behaviour [6], CATY-IRL permits to discriminate among such rewards based on their level of compatibility with the demonstrations. $(iii)$ In a **Reward Learning** [3] setting, CATY-IRL allows us to integrate the constraints provided by various feedbacks (e.g., demonstrations and preferences) based on a *soft* notion of constraints satisfaction. Finally, $(iv)$ CATY-IRL also allows us to integrate **demonstrations from different environments** [7] in a *soft* manner.
>
> We will make this point clearer in the paper.
>
> 2- The Reviewer can find two examples for Proposition 3.1 in Appendix B.1. We will leverage the additional page to provide more insights and intuition on the propositions and theorems, and to improve the readability of the paper.
>
> 3- In common **IRL algorithms** [6], the goal is to learn a *single* reward function that minimizes the suboptimality of the expert's policy. In **IL algorithms** (e.g., [8]), the goal is to learn a *policy* whose suboptimality w.r.t. the expert's policy suboptimality is minimized under all rewards, since the true reward is unknown.
>
> Instead, the goal of **our IRL classification algorithm** is to *minimize the error at estimating the suboptimality of the expert's policy under any reward*. In other words, we are not looking for some special reward for which the expert's suboptimality is small, but our objective is to learn the expert's suboptimality under any possible reward function. The insight is that we aim to exploit the *entire expressive power* of demonstrations from an optimal expert to characterize the *whole* range of reward functions.
>
> We will integrate the section on the related works with this discussion.
>
> 4- We thank the Reviewer for having made us notice this issue. Simply put, CATY-IRL is computationally efficient because it exploits a computationally efficient algorithm, RFLin [9], as a sub-routine (see Remark 4.1 in [9] for additional details), and then all other steps require constant time and space to be executed.
>
> We will add this comment to the main paper and a detailed analysis of the computational complexity.
>
> 5- We thank the Reviewer for the suggestion. We will change it.
> ## Questions
> > How would you apply your algorithm to a real world IRL problem?
>
> See the comment to the *Motivation* weakness.
>
> > What is the computational complexity of the classification step?
>
> The classification step consists in executing a RFE algorithm as sub-routine, and then executing some simple operations that require constant time and space, thus, the computational complexity of the classification step is the same as the RFE sub-routine. Specifically, for Linear MDPs, algorithm RFLin [9] is computationally efficient (see Remark 4.1 in [9]).
>
> ## References
>
> [1] Ng and Russell. Algorithms for IRL. ICML 2000.
>
> [2] Metelli et al. Provably efficient learning of transferable rewards. ICML 2021.
>
> [3] Skalse et al. Invariance in policy optimisation and partial identifiability in reward learning. ICML 2023.
>
> [4] Ziebart et al. Maximum entropy IRL. AAAI 2008.
>
> [5] Hadfield-Menell et al. Inverse Reward Design. NeurIPS 2017.
>
> [6] Arora and Doshi. A survey of IRL: Challenges, methods and progress. Artificial Intelligence 2018.
>
> [7] Cao et al. Identifiability in IRL. NeurIPS 2021.
>
> [8] Ho and Ermon. Generative adversarial IL. NeurIPS 2016.
>
> [9] Wagenmaker et al. Reward-free RL is no harder than reward-aware RL in linear MDPs. ICML 2022.

---

> ### Comment · Reviewer_x3io · 2024-08-08
> **Post rebuttal comment**
>
> Thank you for the thorough response. It mostly clarified my questions, so I decided to raise my score to 7. However, I would have the following follow-up questions /remarks:
> 1. I think in the definition of the feasible reward set, it should be clarified that for linear MDPs, we are only considering rewards that are parametrized by $\langle \phi(s,a),\theta_h \rangle$. At the moment, the definition just requires $r\in[-1,1]^{S\times A \times [H]}$.
> 2. What exactly do you mean by "the feasible reward set contains a continuum of rewards"? Since you parametrize the reward using a finite number of features, the set of feasible rewards should be confined within the span of these features (which is a finite-dimensional subspace).
> 3. Just an observation: In Theorem 3.2, the poor scaling $\Omega(S)$ seems to be related to the fact that we need to visit all states to exclude $\lbrace 0 \rbrace$ as the feasible reward set. I think when we additionally assume that the expert is uniquely realizable for some reward, then the problem wouldn't occur. Do you agree?

---

> > ### Author Response · Authors · 2024-08-09
> >
> > Thank you. We address the Reviewer additional questions/remarks below:
> >
> > > I think in the definition of the feasible reward set, it should be clarified that for linear MDPs, we are only considering rewards that are parametrized by $\langle \phi(s,a),\theta_h \rangle$. At the moment, the definition just requires $r\in[-1,1]^{S\times A \times [H]}$
> >
> > Definition 3.1 is general and independent of structural assumptions of the MDP, like Linear MDPs. Nevertheless, we agree that we should remark that, for Linear MDPs, we consider just rewards parametrized through the feature mapping $\phi$. We will clarify this point in the paper.
> >
> > > What exactly do you mean by "the feasible reward set contains a continuum of rewards"? Since you parametrize the reward using a finite number of features, the set of feasible rewards should be confined within the span of these features (which is a finite-dimensional subspace).
> >
> > Yes, the feasible set is confined within the span of the features, which is still a continuous space. For this reason, there are *infinite* rewards inside the feasible set, thus we cannot construct an algorithm that outputs all these rewards. We might construct an algorithm that outputs the *constraints* defining the feasible set, but then we could use such constraints *only* for classifying rewards as inside or outside the feasible set. Therefore, we prefer to explicitly implement a classifier.
> >
> > > Just an observation: In Theorem 3.2, the poor scaling $\Omega(S)$ seems to be related to the fact that we need to visit all states to exclude $\lbrace 0 \rbrace$ as the feasible reward set. I think when we additionally assume that the expert is uniquely realizable for some reward, then the problem wouldn't occur. Do you agree?
> >
> > That is an interesting question. We agree with the Reviewer that the additional assumption that there is a single optimal policy $\pi^E$ for the expert's reward $\theta^E$ would simplify the hard instances in the proof of Theorem 3.2, so that the lower bound $\Omega(S)$ would not hold anymore. The intuition is interesting, and even though we are not sure that it gets rid of the $\Omega(S)$ dependence, we think that it might be analysed in future works.

---

> > > ### Comment · Reviewer_x3io · 2024-08-12
> > >
> > > Thank you very much for the clarifications. I will maintain my positive score.

---

### Official Review · Reviewer_KyLX · 2024-07-12

**Soundness:** 2
**Presentation:** 2
**Contribution:** 3
**Rating:** 5
**Confidence:** 3

**Summary:**

This paper finds that the feasible reward set cannot be efficiently learned even under linear MDPs. Therefore, the paper proposes a new notion called "reward compatibility" that generalizes the notion of "feasible set" and thereby casts IRL as a classification problem. The paper proposes an algorithm to solve this new classification problem and theoretically show that the sample complexity of the proposed algorithms is independent of state cardinality for linear MDPs.

**Strengths:**

1. This paper proposes a new notion called "reward compatibility" and novelly formulates IRL as a classification problem based on the notion.

2. This paper is theoretically solid, which is also the biggest strength of the paper.

**Weaknesses:**

1. The paper title is a little bit exaggerating. The title is "scale IRL to large state spaces", while what the paper does is to scale one specific kind of IRL, i.e., learning the feasible reward set, to large state spaces. In fact, many other kinds of IRL algorithms can work quite efficiently in continuous state spaces without the linear MDP assumption. For example, maximum likelihood IRL [1] is sample efficient where only one sample is needed for each reward update. I think what the authors mean here is that learning the feasible reward set is difficult in large state space, however, learning feasible reward set is only a kind of IRL, but does not represent all IRL methods. I suggest that the authors change the general terminology "IRL" to some more specific terminologies, given that some other kinds of IRL methods can already scale to large state spaces.

2. The paper uses the terminology "online IRL" to represent IRL that needs to interact with the environment. The paper also adds a footnote to explain this notion. I know that this is to contrast offline IRL. However, it is still confusing because online IRL is already defined in literature [2,3,4], i.e., the IRL setting where the demonstrated trajectories are revealed sequentially. Therefore, I highly suggest the authors to use another terminology to avoid confusion.

3. In example 4.1, the authors mention that the reward function with smaller $\bar{C}(r)$ is more compatible. This can be questionable. For example, suppose $r_2=2 r_1$, then $\bar{C}(r_2)>\bar{C}(r_1)$ if they are both positive. However, can we say that $r_1$ is more compatible than $r_2$? The MDPs are equivalent if we multiply the reward by a constant. Then intuitively, $r_1$ and $r_2$ should be same compatible right?

4. Lack of empirical evaluation.

[1] Maximum-likelihood inverse reinforcement learning with finite-time guarantees

[2] First-person activity forecasting with online inverse reinforcement learning

[3] Online inverse reinforcement learning under occlusion

[4] Learning multi-agent behaviors from distributed and streaming demonstrations

**Questions:**

Please see weaknesses.

**Limitations:**

The limitation is discussed in Section 7.

---

> ### Author Rebuttal · Authors · 2024-08-03
>
> We thank the Reviewer for praising our theoretical analysis as solid, and for recognizing the novelty of the formulation of IRL as a classification problem, which we believe is an important finding of our work.
>
> ## Weaknesses
>
> 1- We agree with the Reviewer that the general terminology "*Inverse Reinforcement Learning*" may be confusing, since it does not reveal the specific IRL formulation considered. Even though previous works on the feasible set have adopted the same general notation [5,6,7,8,9], we agree that a more specific terminology, like "*Maximum Likelihood IRL*" [1], "*Bayesian IRL*" [10], or "*Maximum Entropy IRL*" [11], would be clearer. We will change it to "*How does Learning the Feasible Reward Set Scale to Large State Spaces?*".
>
> 2- Again, we agree with the Reviewer that overloading common IRL terminology may create some confusion. Although "*Online IRL*" fits the analysed problem setting, we will resort to "*Active Exploration IRL*", which describes the possibility of exploring the environment, and which is compatible with previous work [7].
>
> 3- Thank you for the interesting question, which allows us to remark a *very important* point about the **interpretation of the notion of reward function** in MDPs, and in particular about the **scale** of the rewards.
>
> The MDP is a model, i.e., a simplified representation of reality, which is commonly applied to 2 different kinds of real-world scenarios: $(i)$ problems in which the agent (learner in RL or expert in IRL) actually **receives** some kind of scalar feedback from the environment, which can be *modelled as a reward function*; $(ii)$ problems in which the agent does **not receive** a feedback from the environment, but its objective, i.e., its structure of preferences among state-action trajectories (which trajectories are better than others), satisfies some axioms that permit to *represent it through a scalar reward* [13,14] (this is referred to as the *Reward Hypothesis* in literature [12]).
>
> There is an enormous difference between scenario $(i)$ and scenario $(ii)$. **In $(i)$ the notion of $\epsilon$-optimal policy is well-defined** for any fixed $\epsilon>0$, because the reward function is given and, thus, *fixed*. Instead, in $(ii)$, the notion of reward function is a *mere* mathematical artifact used to represent preferences among trajectories, whose existence is guaranteed by a set of assumptions/axioms [12,13,14]. As the Reviewer has observed, *positive affine transformations* of the reward do not affect the structure of preferences represented (see [13] or Section 16.2 of [15] or [16]). Therefore, **in $(ii)$, the notion of $\epsilon$-optimal policy is *not* well-defined**, because rescaling a reward function $r$ to $kr$ changes the suboptimality of some policy $\pi$ from $\epsilon$ to $k\epsilon$. In other words, for fixed $\epsilon>0$, any policy can be made $\epsilon$-optimal by simply rescaling a reward $r$ to $kr$ for some *small enough* $k>0$.
>
> In **IRL**, this issue is even more influential because, although we are in setting $(i)$, we have *no* idea on the scale of the true reward function. For this reason, *our solution* is to attach to any reward $r$ a notion of compatibility $\overline{\mathcal{C}}(r)$ which **implicitly** contains information about the *scale* of the reward $r$. Compatibilities of different rewards (e.g., $r_1$ and $r_2$ in the Reviewer example) cannot be compared unless the rewards have the same scale (e.g., $r_1$ and $r_2$ have different scales, thus their compatibilities shall not be compared).
>
> It should be observed that in Appendix C.2 we discuss a **notion of compatibility *independent* of the scale of the reward**. However, we show that it suffers from major drawbacks that make the notion of compatibility introduced in the main paper (Definition 4.1) more suitable for the IRL problem.
>
> In conclusion, the answer to the Reviewer's question is **no, rewards $r_1$ and $r_2$ should not have the same compatibility, because they have different scales, and the notion of compatibility (i.e., suboptimality) is strictly connected to the scale of the reward**. To carry out a fair comparison of compatibilities, one should rescale the compatibility of each reward based on the scale of the reward.
>
> We will make this point clear in the paper.
>
> 4- We stress that the contribution of the paper is theoretical and, given the contributions provided, we believe that an empirical validation of the proposed algorithm is out of the scope of this work.
>
> ## References
>
> [1] Zeng et al. Maximum-likelihood inverse reinforcement learning with finite-time guarantees. NeurIPS, 2022.
>
> [2] Rhinehart and Kitani. First-person activity forecasting with online inverse reinforcement learning. ICCV, 2017.
>
> [3] Arora et al. Online inverse reinforcement learning under occlusion. AAMAS, 2019.
>
> [4] Liu and Zhu. Learning multi-agent behaviors from distributed and streaming demonstrations. NeurIPS, 2023.
>
> [5] Zhao et al. Is inverse reinforcement learning harder than standard reinforcement learning? ICML, 2024.
>
> [6] Lazzati et al. Offline inverse rl: New solution concepts and provably efficient algorithms. ICML, 2024.
>
> [7] Lindner et al. Active exploration for inverse reinforcement learning. NeurIPS, 2022.
>
> [8] Metelli et al. Towards theoretical understanding of inverse reinforcement learning. ICML, 2023.
>
> [9] Metelli et al. Provably efficient learning of transferable rewards. ICML, 2021.
>
> [10] Ramachandran and Amir. Bayesian inverse reinforcement learning. IJCAI, 2007.
>
> [11] Ziebart el al.. Maximum entropy inverse reinforcement learning. AAAI, 2008.
>
> [12] Sutton and Barto. Reinforcement Learning: An Introduction. 2018.
>
> [13] Shakerinava and Ravanbakhsh. Utility theory for sequential decision making. ICML, 2022.
>
> [14] Bowling et al. Settling the reward hypothesis. ICML, 2023.
>
> [15] Russell and Norvig. Artificial Intelligence: A Modern Approach. 2010.
>
> [16] David M. Kreps. Notes on the theory of choice. Westview Press, 1988.

---

> > ### Comment · Reviewer_KyLX · 2024-08-12
> >
> > Thanks for the response. I'll keep my current positive rating.

---

### Official Review · Reviewer_K3ze · 2024-07-12

**Soundness:** 4
**Presentation:** 3
**Contribution:** 4
**Rating:** 8
**Confidence:** 3

**Summary:**

Even under the strong assumption implicit in Linear MDPs, the learning of the set of rewards that make the expert’s policy optimal doesn’t scale well. This improves somewhat, if additionally a notion of compatibility of rewards is introduced, because in this way and under these conditions the IRL problem can be seen as a classification task. In this context a number of theoretical results are possible minimax optimality of a thus formulated IRL algorithm, complexity bounds and some contribution to the complexity theory of reward-free exploration.

**Strengths:**

Outsourcing some of related work to the appendix seems a good idea here, as the proper introduction is still readable and interesting. However, it may have been an alternative option to move the “original contributions” to the appendix as these are already mentioned in the abstract and in --- the paper. The paper aims at much, and although it delivers, but could be more focused and concise.

**Weaknesses:**

“How to” is not really addressed, “How to Scale Inverse RL to Large State Spaces?” should be “How does Inverse RL Scale to Large State Spaces?”

The separation of exploration and classification phases may not appear to be a problem at this level of abstraction, but practically this can be a forbidding features of an algorithm.

The approach uses a restrictive problem setting, while it could be preferable to attempt to discover any compositional structure or any latent low-dimensional manifolds, as would be the present in any practical problem if the problem can be treated at all at larger scales.

No simulation included, although it should be easy provide some illustration, in particular for any worst-case results.

Objective-free exploration need more study to receive the attention that it is claimed to deserve, but the current definition probably needs to be more precise.

Here (as in most contributions to this conference) the use of display equations is dearly missed. Couldn’t the amount of ink be used to measure the length of the papers?

The use of color in the manuscript could be more systematic, if it is encouraged at all.

There is too much material in the paper. It may be tempting to publish a systematic study at a conference to reach some visibility, but it creates an imbalance among the contribution and may bias future submission.

**Questions:**

Would objective-free exploration be independent on the linear MDP assumption? Is it needed at all in the present paper?

Can you compare to other IRL algorithms?

**Limitations:**

Practical application is limited, but this is outside the scope of this very impressive paper.

---

> ### Author Rebuttal · Authors · 2024-08-03
>
> We are glad that the Reviewer found our paper to be impressive and our contribution to be substantial. We provide detailed replies to their questions/comments below.
>
> ## Weaknesses
>
> > “How to” is not really addressed, “How to Scale Inverse RL to Large State Spaces?” should be “How does Inverse RL Scale to Large State Spaces?”
>
> We thank the Reviewer for the observation. We will change it.
>
> > The separation of exploration and classification phases may not appear to be a problem at this level of abstraction, but practically this can be a forbidding features of an algorithm.
>
> From a theoretical perspective, the separation of the exploration phase from the subsequent phase permits to *isolate the challenges of exploration*, as mentioned in [1]. In practical applications, we require our learner to be able to actively explore the environment. Since the results of the classification phase are available *after* the exploration has completed, then any subsequent task has to be postponed.
>
> > The approach uses a restrictive problem setting, while it could be preferable to attempt to discover any compositional structure or any latent low-dimensional manifolds, as would be the present in any practical problem if the problem can be treated at all at larger scales.
>
> We agree with the Reviewer that Linear MDPs suffer from some limitations if we want to apply them to real-world applications, but we believe that they represent an important initial step toward the development of provably efficient IRL algorithms with more general function approximation structures.
>
> > No simulation included, although it should be easy provide some illustration, in particular for any worst-case results.
>
> We agree that an empirical validation would be nice, but, as the Reviewer also noted, we are already providing a lot of contributions, leaving no space for experiments unfortunately.
>
> > Objective-free exploration need more study to receive the attention that it is claimed to deserve, but the current definition probably needs to be more precise.
>
> The formulation of the Objective-Free Exploration (OFE) problem setting (Definition 6.1) is intentionally provided in a general way. The reason is that we aim that this definition will be used as a *template* for analysing exploration problems, and, thus, it should be instantiated more precisely in the specific problem depending on the tasks to be solved (see Example F.1 in Appendix F). In appendix E.1, we provide more insights on OFE, by identifying two additional problems beyond RL and IRL whose exploration phase can be casted in this scheme.
>
> > Here (as in most contributions to this conference) the use of display equations is dearly missed. Couldn’t the amount of ink be used to measure the length of the papers? The use of color in the manuscript could be more systematic, if it is encouraged at all.
>
> We agree with the Reviewer that some choices about the layout and the design of the paper may be improved. We will leverage the additional page to improve the readability of the paper.
>
> > There is too much material in the paper. It may be tempting to publish a systematic study at a conference to reach some visibility, but it creates an imbalance among the contribution and may bias future submission.
>
> We agree with the Reviewer that it may be hard to process all of the contributions we provide. Nevertheless, we believe that they cannot be separated from each other, because it would negatively affect the presentation and the understanding of the paper.
>
> ## Questions
>
> > Would objective-free exploration be independent on the linear MDP assumption? Is it needed at all in the present paper?
>
> Yes, Objective-Free Exploration (OFE) is a problem setting which is independent of specific assumptions on the structure of the MDP (e.g., linear MDP).
>
> Although, at first sight, OFE may seem out of scope in this paper, we believe that its formulation is significant as: $(i)$ it provides a unifying *exploration* framework for RL and IRL problems; $(ii)$ it highlights the efficiency and efficacy of Reward-Free Exploration (RFE) strategies for solving both tasks.
>
> Since part of our contribution consists in showing that RL and IRL enjoy the same sample complexity rate (Theorem 5.1, Theorem 6.1, and Theorem 6.2) in tabular problems and the same upper bound in Linear MDPs, the OFE problem setting permits to interpret these results in a *unifying* and original manner.
>
> > Can you compare to other IRL algorithms?
>
> Unfortunately, we cannot compare with popular IRL algorithms like margin maximization [2] or entropy maximization [3] (and their variants) because they enforce additional assumptions on the reward function to recover. The IRL algorithms that consider a problem setting analogous to ours are those in [4,5,6,7,8], whose objective is the estimation of the feasible reward set. Nevertheless, all algorithms presented in [4,5,6,7,8] focus on the tabular setting, and they exhibit an explicit dependence on the size of the state space. Thus, they cannot be used for problems with large state spaces.
>
> ## References
>
> [1] Chi Jin et al. Reward-free exploration for reinforcement learning. ICML, 2020.
>
> [2] Ng and Russell. Algorithms for inverse reinforcement learning. ICML, 2000.
>
> [3] Ziebart et al. Maximum entropy inverse reinforcement learning. AAAI, 2008.
>
> [4] Metelli et al. Provably efficient learning of transferable rewards. ICML, 2021.
>
> [5] Zhao et al. Is inverse reinforcement learning harder than standard reinforcement learning? ICML, 2024.
>
> [6] Lindner et al. Active exploration for inverse reinforcement learning. NeurIPS, 2022.
>
> [7] Lazzati et al. Offline inverse rl: New solution concepts and provably efficient algorithms. ICML, 2024.
>
> [8] Metelli et al. Towards theoretical understanding of inverse reinforcement learning. ICML, 2023.

---

### Official Review · Reviewer_AxcE · 2024-07-16

**Soundness:** 3
**Presentation:** 3
**Contribution:** 2
**Rating:** 5
**Confidence:** 3

**Summary:**

This paper shows that finding the feasible reward set in IRL needs to sample $\Omega(S)$ number of samples, even when the MDP possesses a linear structure. To enable more efficient scaling with $S$, the authors propose another task in IRL called rewards compatibility: deciding whether $\pi^{E}$ is $\epsilon$-optimal under a given reward $r$. They further use reward-free algorithms to solve such tasks and propose a matching lower bound. As a byproduct, the lower bound also improves the existing lower bounds in tabular reward-free RL.

**Strengths:**

The theoretical analysis is solid. In particular, the lower bounds for feasible reward set learning and reward compatibility are novel and significant as they quantify the hardness of these two tasks in IRL.

**Weaknesses:**

1. The reward compatibility framework is not that interesting in my opinion because it requires you to input a reward function, which indeed makes IRL a standard RL problem given that reward. More specifically, the reward compatibility framework is just policy optimization and evaluation of $\pi^E$ under a given reward $r$, which has been studied sufficiently before.
2. The algorithms that the authors propose are also just the existing algorithms in standard RL, so there is no novelty in the algorithm design.

**Questions:**

The lower bound in Theorem 3.2 characterizes the difficulty of identifying the exact feasible reward set. However, in many cases we just want to learn a feasible reward with some desirable properties instead of the whole set. For this setting will the lower bound still hold?

**Limitations:**

None.

---

> ### Author Rebuttal · Authors · 2024-08-03
>
> We are glad that the Reviewer appreciated the novelty and significance of the lower bound results, and the solidity of the theoretical analysis. Below, we report answers to the Reviewer's comments.
>
> ## Weaknesses
> > The reward compatibility framework is not that interesting in my opinion because it requires you to input a reward function, which indeed makes IRL a standard RL problem given that reward. More specifically, the reward compatibility framework is just policy optimization and evaluation of $\pi^E$ under a given reward $r$, which has been studied sufficiently before.
>
> We agree with the Reviewer that, as we *demonstrate* in the paper (Theorem 5.1 and Theorem 6.2), the rewards compatibility framework turns out to be minimax optimally solvable by a policy optimization and evaluation algorithm in the Reward-Free Exploration (RFE) [1] setting, showing an equivalence of the two problems. *However*, it should be remarked that the significance and the novelty of the scheme lies in the **original formulation and interpretation of the Inverse Reinforcement Learning (IRL) problem**, and *not* in the specific solution technique.
>
> Our ultimate goal is to understand the computational and statistical complexity of inferring as much information (i.e., constraints) as possible about the expert's reward function $r^E$ in an approximate setting. Due to the partial identifiability of the IRL problem [2,3,4], as explained in the introduction, common IRL approaches like *margin* [2] or *entropy* [5] maximization are heuristically "biased" toward a specific reward function somehow close to $r^E$. For this reason, we resort to the *feasible set* formulation [4,6], which does not introduce additional assumptions about $r^E$ beyond the optimality of the observed expert policy $\pi^E$. Nevertheless, since the feasible set is inefficient to learn (being a set) in problems with a large state space (see Theorem 3.2), we introduce the notion of *rewards compatibility*.
>
> **Ideally**, we would like to have an IRL classification algorithm (e.g., some variant of CATY-IRL) that takes in input the problem instance and outputs a binary partition of the space of reward functions into *at most* $\epsilon$-compatible rewards and *at least* $\epsilon$-compatible rewards. **In practice**, due to the impossibility of computing such output (limited computational resources), we develop an algorithm, i.e., CATY-IRL, that *potentially* can classify all possible rewards, but that actually classifies only the input rewards.
>
>  > The algorithms that the authors propose are also just the existing algorithms in standard RL, so there is no novelty in the algorithm design.
>
> We agree with the Reviewer that the proposed algorithm, CATY-IRL, executes existing RFE algorithms as sub-routines. *However*, a major contribution of our work is **demonstrating that such sub-routines actually solve the IRL classification problem** (Definition 4.2) **in a minimax optimal manner** (Theorem 5.1 and Theorem 6.1), and thus, *proving* an equivalence between IRL and RFE from the sample complexity viewpoint which has been *conjectured* in previous works (e.g., Appendix A of [4] or Appendix D of [7]). We will stress this in the final version.
>
> ## Questions
> > The lower bound in Theorem 3.2 characterizes the difficulty of identifying the exact feasible reward set. However, in many cases we just want to learn a feasible reward with some desirable properties instead of the whole set. For this setting will the lower bound still hold?
>
> **Yes**. To see why, as an example, assume you aim to learn only the feasible reward function that satisfies the *margin maximization* criterion in Equation (6) of [2]. By re-using the same hard instance constructed in the proof of Theorem 3.2, we see that no algorithm can discriminate between policies $\pi^E_1$ and $\pi^E_2$ unless it collects a sample from state $\overline{s}$. Without knowing which policy between $\pi^E_1$ and $\pi^E_2$ is the true expert policy, any algorithm will fail with probability at least $0.5$ in the worst case at outputting an accurate estimate of the reward function, since reward $\theta_1=1$ is the margin maximizer for policy $\pi^E_1$, and $\theta_2=0$ is the margin maximizer for policy $\pi^E_2$, and the distance $d$ (see Equation (1)) between these rewards is exactly $1$. The result follows by observing that we need $\Omega(S)$ samples to spot state $\overline{s}$.
>
> To avoid this negative result, we can **learn a single $\epsilon$-compatible reward** (for some $\epsilon>0$) that satisfies the same margin-maximization criterion, instead of learning the feasible reward that satisfies the criterion. Nevertheless, for the reasons presented in the introduction of the paper, we prefer to be *criterion-agnostic*, and to learn *all* the compatible rewards.
>
> We will make this point clear in the paper.
>
> ## References
>
> [1] Chi Jin et al. Reward-free exploration for reinforcement learning. ICML, 2020.
>
> [2] Ng and Russell. Algorithms for inverse reinforcement learning. ICML, 2000.
>
> [3] Skalse et al. Invariance in policy optimisation and partial identifiability in reward learning. ICML, 2023.
>
> [4] Metelli et al. Provably efficient learning of transferable rewards. ICML, 2021.
>
> [5] Ziebart et al.. Maximum entropy inverse reinforcement learning. AAAI, 2008.
>
> [6] Zhao et al. Is inverse reinforcement learning harder than standard reinforcement learning? ICML, 2024.
>
> [7] Lindner et al. Active exploration for inverse reinforcement learning. NeurIPS, 2022.

---

> > ### Comment · Reviewer_AxcE · 2024-08-13
> >
> > Thank you for the rebuttal! I will keep a positive evaluation.

---

### Decision · Program_Chairs · 2024-09-25

**Decision:**

Accept (poster)

**Comment:**

The paper provides a solid and original contribution to the problem of inverse reinforcement learning (IRL) in large state-action spaces, particularly by introducing the reward compatibility framework and the CATY-IRL algorithm. While the reviewers raised concerns about the clarity of motivation for reward compatibility and the need for better intuition and comparisons to related work, they also recognized the technical novelty. All the reviewers agree with acceptance.